# Global population exposure to landscape fire air pollution from 2000 to 2019

Rongbin Xu[1], Tingting Ye[1], Xu Yue[2✉], Zhengyu Yang[1], Wenhua Yu[1], Yiwen Zhang[1], Michelle L. Bell[3], Lidia Morawska[4], Pei Yu[1], Yuxi Zhang[1], Yao Wu[1], Yanming Liu[1], Fay Johnston[5], Yadong Lei[6], Michael J. Abramson[1], Yuming Guo[1✉] & Shanshan Li[1✉]

Wildfires are thought to be increasing in severity and frequency as a result of climate change[1–5]. Air pollution from landscape fires can negatively affect human health[4–6], but human exposure to landscape fire-sourced (LFS) air pollution has not been well characterized at the global scale[7–23]. Here, we estimate global daily LFS outdoor fine particulate matter ($PM_{2.5}$) and surface ozone concentrations at 0.25° × 0.25° resolution during the period 2000–2019 with the help of machine learning and chemical transport models. We found that overall population-weighted average LFS $PM_{2.5}$ and ozone concentrations were 2.5 μg m$^{-3}$ (6.1% of all-source $PM_{2.5}$) and 3.2 μg m$^{-3}$ (3.6% of all-source ozone), respectively, in 2010–2019, with a slight increase for $PM_{2.5}$, but not for ozone, compared with 2000–2009. Central Africa, Southeast Asia, South America and Siberia experienced the highest LFS $PM_{2.5}$ and ozone concentrations. The concentrations of LFS $PM_{2.5}$ and ozone were about four times higher in low-income countries than in high-income countries. During the period 2010–2019, 2.18 billion people were exposed to at least 1 day of substantial LFS air pollution per year, with each person in the world having, on average, 9.9 days of exposure per year. These two metrics increased by 6.8% and 2.1%, respectively, compared with 2000–2009. Overall, we find that the global population is increasingly exposed to LFS air pollution, with socioeconomic disparities.

The term landscape fires refers to any fires burning in natural and cultural landscapes, for example natural and planted forest, shrub, grass, pastures, agricultural lands and peri-urban areas[24]. It includes both planned or controlled fires (for example, prescribed burns, agricultural fires) and wildfires (defined as uncontrolled or unplanned fires burning in wildland vegetation[25]). There is evidence that wildfires are increasingly frequent and severe as a result of climate change[1–5]. Compared with the direct exposure to the flames and heat of landscape fires, the exposure to air pollution caused by landscape fire smoke travelling hundreds, and sometimes even thousands, of kilometres[4] can affect much larger populations, and cause much larger public health risks[6]. Mapping and tracking population exposure to landscape fire-sourced (LFS) air pollution (mainly including particulate matter with a diameter of 2.5 μm or less ($PM_{2.5}$) and ozone ($O_3$)) are essential for monitoring and managing the health impacts of such fires, implementing targeted prevention and interventions, and strengthening arguments for mitigation of climate change.

However, there are a lack of accurate daily fire-sourced air pollution data with complete spatiotemporal coverage across the globe. Wildfires often mainly threaten suburban, rural and remote areas where there are few or no air quality monitoring stations[4]. In many low-income

countries, there are no air quality monitoring stations even in urban areas. Therefore, the data gap cannot be addressed by using air quality monitoring stations alone.

Our previous studies have estimated the daily fire-sourced $PM_{2.5}$ for Brazil[7] and 749 worldwide locations[8] during the period 2000–2016. Many studies also estimated fire-related $PM_{2.5}$ in the USA[9–18] and Europe[19,20] using various approaches (for example, chemical transport models, satellite-based fire smoke plume, machine learning). However, there are still a lack of data in many other regions, particularly sub-Saharan Africa and Southeast Asia where landscape fires are frequent[21]. Two early studies attempted to address the data gap at a global scale using chemical transport models; they estimated global daily fire-sourced $PM_{2.5}$ for 1997–2006[22] and 2016–2019[23]. However, the accuracy of chemical transport model outputs could be problematic without calibration against observations of air quality monitoring stations[16], and these two global studies could not assess the long-term trend of fire-sourced $PM_{2.5}$ given their short study periods. Furthermore, to our knowledge, no previous study has estimated global LFS $O_3$. This important fire-related pollutant has been estimated only for the USA using chemical transport models without calibration against station observations[9,13]. Last but not the least, all these

[1]Climate, Air Quality Research Unit, School of Public Health and Preventive Medicine, Monash University, Melbourne, Victoria, Australia. [2]Jiangsu Key Laboratory of Atmospheric Environment Monitoring and Pollution Control, Collaborative Innovation Center of Atmospheric Environment and Equipment Technology, Joint International Research Laboratory of Climate and Environment Change, School of Environmental Science and Engineering, Nanjing University of Information Science and Technology, Nanjing, China. [3]School of the Environment, Yale University, New Haven, CT, USA. [4]International Laboratory for Air Quality and Health, Queensland University of Technology, Brisbane, Queensland, Australia. [5]Menzies Institute for Medical Research, University of Tasmania, Hobart, Tasmania, Australia. [6]State Key Laboratory of Severe Weather and Key Laboratory of Atmospheric Chemistry of CMA, Chinese Academy of Meteorological Sciences, Beijing, China. ✉e-mail: yuexu@nuist.edu.cn; yuming.guo@monash.edu; shanshan.li@monash.edu

previous studies focused mainly on data generation or health impact assessment; little attention has been paid to population exposure assessment.

This study estimated the daily fire-sourced $PM_{2.5}$ and $O_3$ concentrations at $0.25° \times 0.25°$ (about 28 km × 28 km at the equator) spatial resolution across the globe from 2000 to 2019. Through linking the dataset with global population distribution data, we aimed to perform a comprehensive assessment of global population exposures to fire-sourced $PM_{2.5}$ and $O_3$ during the period 2000–2019.

## Data validation

As detailed in Methods, Extended Data and Supplementary Information, we validated our estimated all-source and fire-sourced $PM_{2.5}$ and $O_3$ in several ways.

The spatial tenfold cross-validation (CV) (that is, by dividing all stations into ten approximately equal subsets, then performing validation of the model estimates on each subset for the model trained in the remaining nine subsets) demonstrated our machine learning models' high level of accuracy in estimating both all-source daily average $PM_{2.5}$ ($R^2 = 0.89$, root mean squared error (RMSE) = 9.24 μg m$^{-3}$) and all-source daily maximum 8 h $O_3$ ($R^2 = 0.80$, RMSE = 19.24 μg m$^{-3}$) in new locations not in the training data. As a further test of our model's ability to generalize to regions far from available training stations, we clustered globally available $PM_{2.5}$ and $O_3$ stations into 75 and 99 contiguous clusters, respectively, and used leave-one-out CV to evaluate model performance on each cluster as it was temporarily excluded from model training. As expected, performance was lower than the spatial tenfold CV. In clusters in which the model was not trained, the model estimates explained 69% and 67% of the overall variations in all-source $PM_{2.5}$ and $O_3$, respectively, and 41% and 52% of local temporal daily variations (that is, after excluding variations across stations and between years) of all-source $PM_{2.5}$ and $O_3$, respectively. This performance, however, was still much higher than the performance of the uncalibrated raw GEOS-Chem outputs, suggesting that our models can predict the daily all-source $PM_{2.5}$ and $O_3$ in large remote areas with no training data with an accuracy much higher than that of the raw GEOS-Chem outputs alone.

Notably, in most regions of the world, we are able to evaluate our model performance in predicting variation only in all-source, but not fire-sourced, $PM_{2.5}$ and $O_3$. We made two further efforts to validate our estimated fire-sourced $PM_{2.5}$ and $O_3$ in some regions.

First, under a straightforward hypothesis that the station-observed $PM_{2.5}$ and $O_3$ during wildfire events are caused mainly by wildfire smoke, we chose ten large wildfire events in Australia, the USA, Chile, Portugal and South Africa to validate our estimated all-source and fire-sourced $PM_{2.5}$ and $O_3$. For each wildfire event, we chose the most affected monitoring station (that is, the nearby station showing the largest increase in observed concentrations during the wildfire event, compared with the pre-wildfire period) as the validation target. During the wildfire event and up to 60 days before and after the event, the observed daily all-source $PM_{2.5}$ or $O_3$ from the most affected station showed good agreement with our estimated daily all-source $PM_{2.5}$ ($R^2 = 0.64$ on average across events) and $O_3$ ($R^2 = 0.78$) based on a model trained in stations excluding all nearby stations, although our estimates tended to substantially understate $PM_{2.5}$ concentrations during some extreme $PM_{2.5}$ periods. Furthermore, we observed an expected increase in the estimated concentrations and proportions (among all sources) of fire-sourced $PM_{2.5}$ and $O_3$ during the selected wildfire events, compared with the pre-wildfire period, suggesting that our models can reasonably capture the wildfires' impacts on the daily $PM_{2.5}$ and $O_3$ concentrations.

Second, we compared our estimated fire-sourced $PM_{2.5}$ with the smoke $PM_{2.5}$ (that is, $PM_{2.5}$ concentrations attributable to fire smoke overhead detected by satellite images) estimated by Childs et al.[17] in the contiguous USA, and found a high agreement (Pearson correlation coefficient $r = 0.88$). When further validated against the smoke $PM_{2.5}$ observed by 2,147 PurpleAir stations that were neither in our training data nor in those of Childs et al.[17], our estimated fire-sourced $PM_{2.5}$ ($R^2 = 0.51$, RMSE = 11.76 μg m$^{-3}$) showed lower accuracy than the estimated smoke $PM_{2.5}$ of Childs et al.[17] ($R^2 = 0.66$, RMSE = 10.46 μg m$^{-3}$), perhaps as a result of our attempts to build a globally generalizable model. However, our performance was still much greater than the accuracy of the fire-sourced $PM_{2.5}$ from raw GEOS-Chem outputs ($R^2 = 0.18$, RMSE = 22.96 μg m$^{-3}$).

On the basis of our validated data, the global population exposures to fire-sourced $PM_{2.5}$ and $O_3$ were described as follows.

## Fire-sourced $PM_{2.5}$ and $O_3$ concentrations

The global spatial distributions of fire-sourced $PM_{2.5}$ and $O_3$ were generally similar in 2000–2009 and 2010–2019 (Fig. 1), with Central Africa exposed to the highest levels of wildfire $PM_{2.5}$ and $O_3$, followed by Southeast Asia, South America and North Asia (Siberia). There were also some other regional hotspots, including north-western Australia, and western USA and Canada. From 2000 to 2019, fire-sourced $PM_{2.5}$ showed statistically significant increasing trends in central and northern Africa, North America, Southeast Asia, Amazon areas in South America, Siberia and northern India, whereas notable decreasing trends were found in southern parts of Africa and South America, northwest China and Japan. Fire-sourced $O_3$ also showed similar statistically significant increasing trends in Central Africa, Siberia, western USA and Canada, Mexico, Southeast Asia and northern India, and similar decreasing trends in northwest China and southern parts of Africa and South America; however, its trends in Amazon areas, central and eastern USA, Northern Africa, Japan and Indonesia were in the opposite direction of the trends of fire-sourced $PM_{2.5}$ in those areas.

The population-weighted average fire-sourced $PM_{2.5}$ and $O_3$ across the globe and six continents fluctuated substantially over the 2000–2019 period (Fig. 2), with different trends and seasonal patterns observed on different continents. The peak months of fire-sourced $PM_{2.5}$ and $O_3$ were June to September and December to January for Africa, March to April for Asia, July to August for Europe, April to May for North America, November to January for Oceania and August to October for South America.

Globally, the annual population-weighted average fire-sourced $PM_{2.5}$ and $O_3$ were 2.5 μg m$^{-3}$ and 3.2 μg m$^{-3}$ in 2010–2019, accounting for 6.1% and 3.6% of all-source $PM_{2.5}$ and $O_3$, respectively (Extended Data Table 1a). The annual population-weighted average wildfire $PM_{2.5}$ from 2000 to 2019 showed increasing trends over the globe (0.11 μg m$^{-3}$ increase per decade, $P = 0.072$ for trend) and in North America (0.27 μg m$^{-3}$ increase per decade, $P = 0.001$ for trend), but decreasing trends in Africa (−0.27 μg m$^{-3}$ per decade, $P = 0.020$ for trend) and South America (−0.61 μg m$^{-3}$ per decade, $P = 0.012$ for trend). The annual population-weighted average wildfire $O_3$ also showed decreasing trends in Africa (−0.45 μg m$^{-3}$ per decade, $P = 0.043$ for trend) and South America (0.60 μg m$^{-3}$ per decade, $P = 0.012$ for trend), but the trend was not significant for the globe or other continents (all $P > 0.37$ for trend).

The proportions of fire-sourced $PM_{2.5}$ and $O_3$ among all sources showed similar spatial distributions for 2000–2009 and 2010–2019 (Extended Data Fig. 1a). The highest landscape fire contribution to $PM_{2.5}$ was observed in Central Africa (up to 70%), followed by South America (approximately 40%), northern Australia (approximately 40%), Southeast Asia (approximately 30%), western USA and Canada (approximately 20% in 2000–2009, increased to approximately 30% in 2010–2019) and Northeast Asia (approximately 20%). The highest landscape fire contribution to $O_3$ was also observed in Central Africa (up to 46%), followed by South America (approximately 30%), northern Australia (up to 20%) and Southeast Asia (up to 20%).

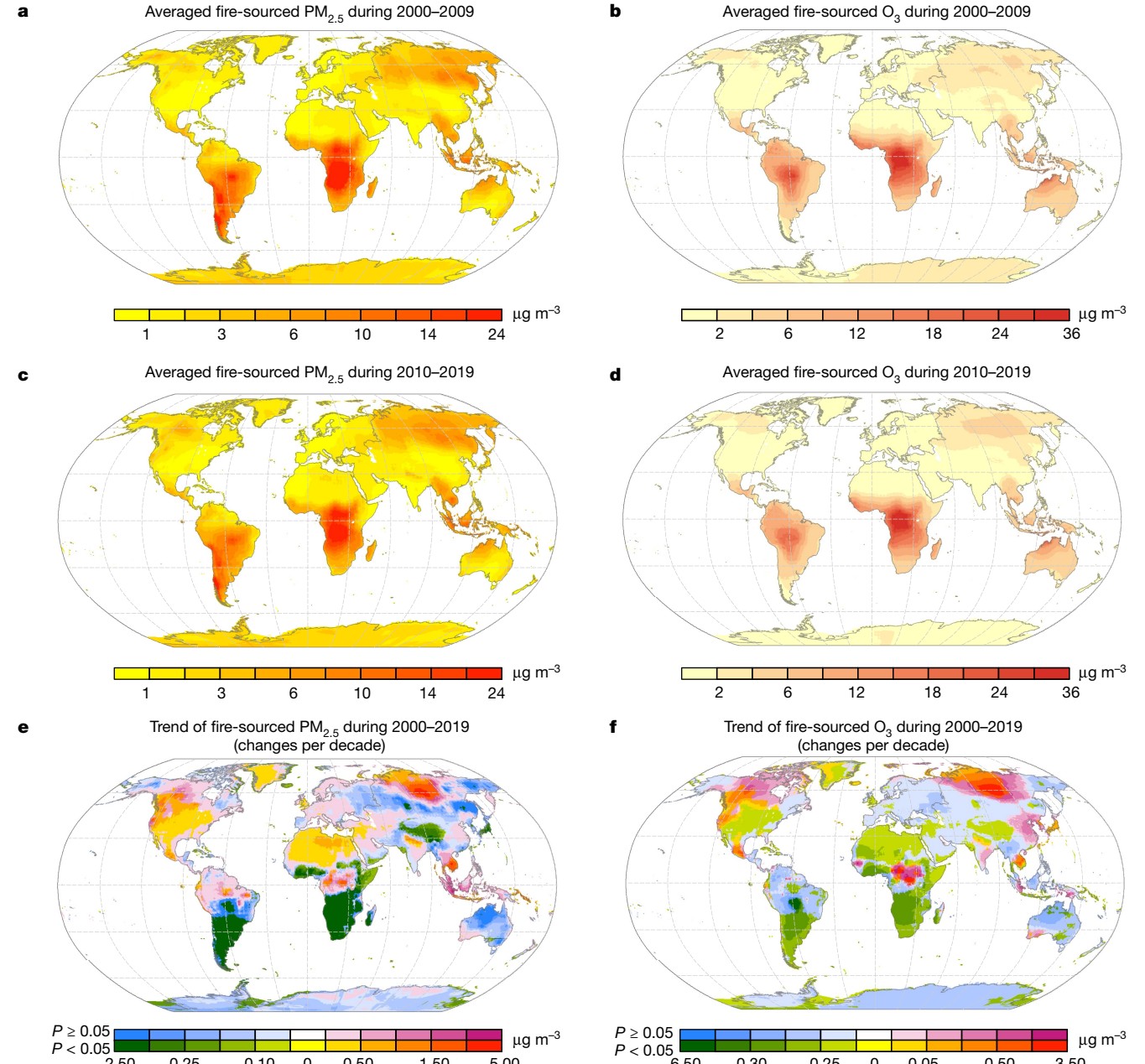

**Fig. 1 | Global maps of estimated concentrations. a–f,** Maps of LFS PM$_{2.5}$ (**a,c,e**) and O$_3$ (**b,d,f**) concentration in the first (**a,b**) and second (**c,d**) decades of 2000–2019, and the estimated trend (**e,f**) during the period. For each 0.25° × 0.25° grid, the trend from 2000 to 2019 was fitted using all annual concentrations during the period (not just 2000 and 2019 data) with a linear regression. $P$ (**e,f**) indicates the $P$ values for long-term trends, with $P < 0.05$ indicating a statistically significant trend.

## Socioeconomic disparities in concentrations

There were consistent socioeconomic disparities in the annual average fire-sourced PM$_{2.5}$ and O$_3$ concentrations (Fig. 3 and Extended Data Table 1a). Countries with a low Human Development Index (HDI) score and low income had the greatest exposure to fire-sourced air pollution, whereas countries with a very high HDI score and high income had the least exposure. The annual population-weighted average fire-sourced PM$_{2.5}$ concentrations in countries with low HDI scores were 2.9- to 4.2-fold (varied in different years) those of countries with very high HDI scores during the period 2000–2019. These ratios for annual fire-sourced O$_3$ (low HDI score versus very high HDI score) were 4.1 to 7.8. Similarly, annual fire-sourced PM$_{2.5}$ and O$_3$ concentrations in low-income

countries were 4.5- to 6.2-fold, and 3.9- to 8.1-fold, respectively, those in high-income countries.

## Global population exposure to SFAP

We defined a substantial fire-sourced air pollution (SFAP) day as at least one of the following scenarios: (1) the daily average PM$_{2.5}$ (all-source PM$_{2.5}$) exceeded the 2021 daily guideline value (15 µg m$^{-3}$) of the World Health Organization (WHO), and fire-sourced PM$_{2.5}$ accounted for at least 50% of the daily PM$_{2.5}$; (2) the daily maximum 8 h O$_3$ (all-source O$_3$) exceeded the WHO's 2021 daily guideline value (100 µg m$^{-3}$), and fire-sourced O$_3$ accounted for at least 50% of the daily O$_3$. The population exposures to SFAP were represented by three metrics, comprising annual total person-days, annual average days per person and annual

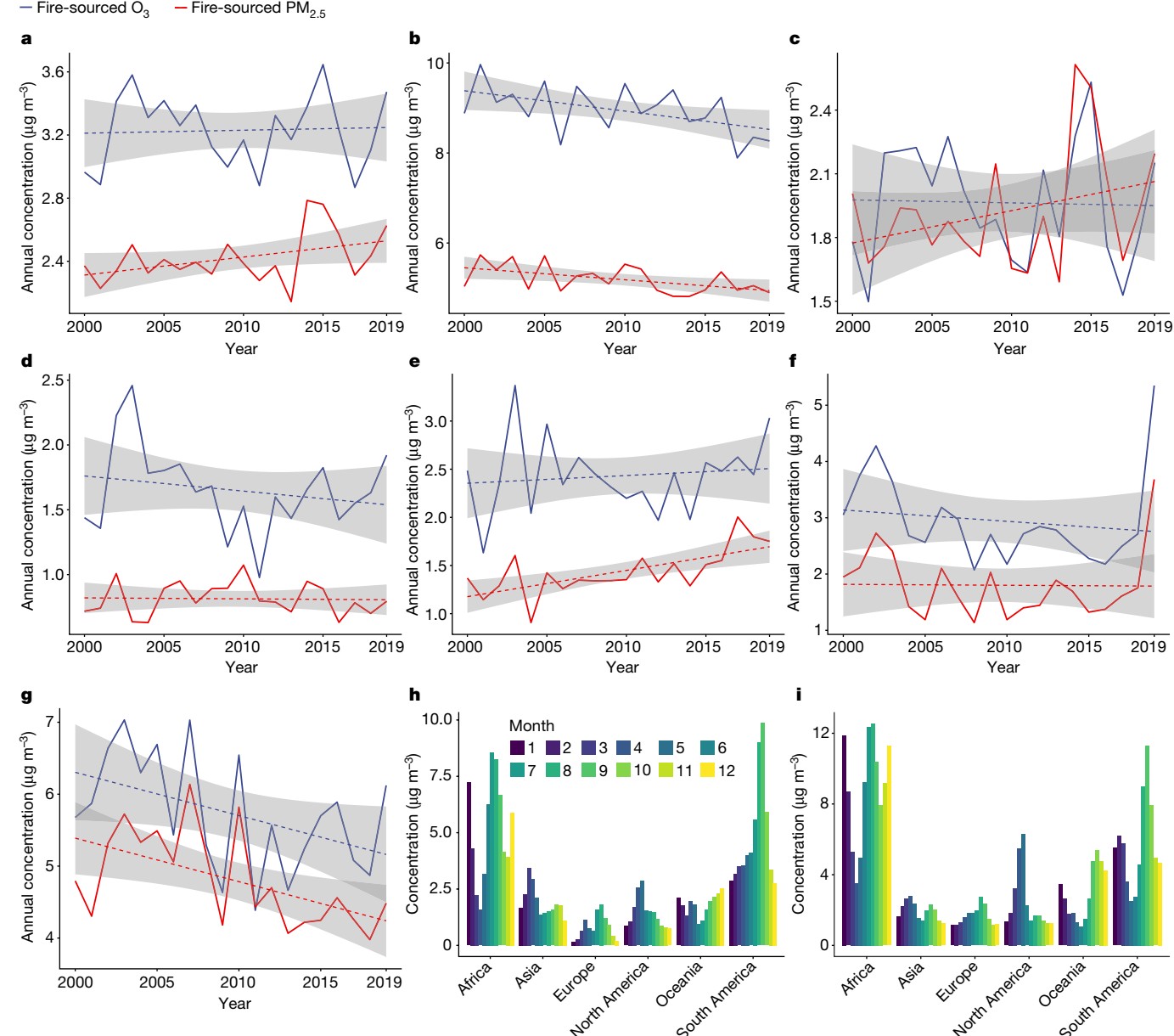

**Fig. 2 | Global and continent-specific trends and seasonal patterns. a–i**, The long-term trend (global (**a**), Africa (**b**), Asia (**c**), Europe (**d**), North America (**e**), Oceania (**f**) and South America (**g**)) and seasonal pattern of population-weighted average fire-sourced $PM_{2.5}$ (**h**) and $O_3$ (**i**) from 2000 to 2019 for the globe and six continents. The dashed lines in **a–g** denote point-estimates of fitted trend by linear regression and the shaded areas denote the corresponding 95% confidence intervals.

total number of people exposed to SFAP. One person-day refers to one person exposed to 1 day of the SFAP; thus the total exposed person-days can be viewed as the total population exposure level to SFAP.

The global total number of exposed person-days increased significantly from 63.2 billion per year during the period 2000–2009 to 72.8 billion per year during 2010–2019 ($P = 0.010$ for trend, an increase of 8.6 billion person-days per decade) (Extended Data Table 1b and Extended Data Fig. 2a). This increase was mainly due to population growth, as the average exposed days per person per year increased only slightly from 9.7 days during 2000–2009 to 9.9 days during 2010–2019. In each year during 2000–2009, 2.04 billion people, on average, were exposed to at least 1 day of SFAP across the globe, and this number rose to 2.18 billion people per year during 2010–2019 ($P = 0.007$ for trend, a 190.1 million-person increase per decade).

There were notable disparities in the population exposures to SFAP between different continents. Africa experienced the largest

proportion of exposed person-days (approximately 50% of global total) over the period 2000–2019, followed by Asia (more than 25%) (Extended Data Table 1b and Extended Data Fig. 2a). Africa experienced the fastest increase in exposed person-days (an increase of 6.0 billion person-days per decade, $P < 0.001$ for trend) from 2000 to 2019. North America also saw a significant increasing trend (an increase of 1.5 billion person-days per decade, $P = 0.042$ for trend).

Africa had the highest average number of days exposed to SFAP per person per year (32.5 days per person per year during 2010–2019), despite a significant decrease (−2.5 days per decade, $P = 0.029$ for trend) since 2000–2009. South America had the second highest average number of exposed days (23.1 days per person per year during 2010–2019), whereas other continents were generally exposed to less than 10 days per person per year, except for a few outliers (for example, 23 days in 2019 for Oceania), and Europe had the lowest average number of

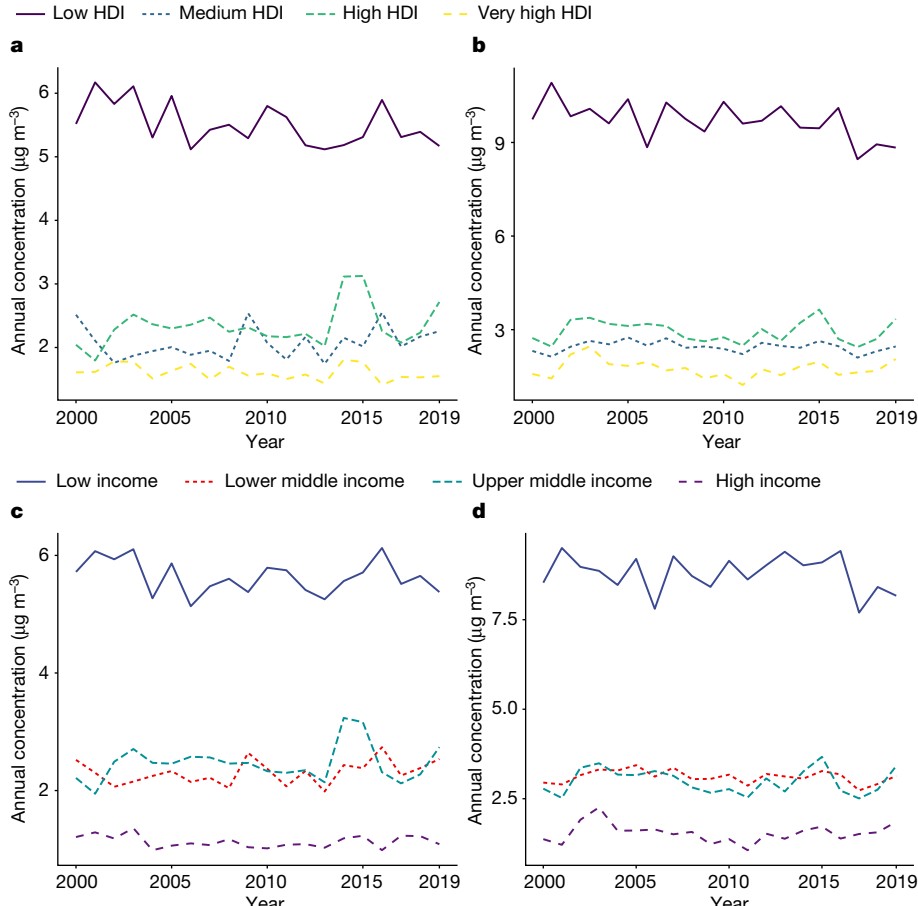

**Fig. 3 | Socioeconomic disparities in exposure between countries. a–d,** Annual population-weighted average fire-sourced PM$_{2.5}$ (**a,c**) and O$_3$ (**b,d**) from 2000 to 2019 by country HDI score (**a,b**) and income level (**c,d**).

exposed days (approximately 1 day per person per year) (Extended Data Table 1b and Extended Data Fig. 2a).

Asia had the largest annual population size exposed to at least 1 day of SFAP (803.1 million people per year during the period 2000–2019, 36.8% of the global total), followed by Africa (596.4 million, 27.4%), South America (342.5 million, 15.7%) and North America (319.2 million, 14.7%) (Extended Data Table 1b and Extended Data Fig. 2a). The fastest increase in exposed population size was seen in North America (a 109.1 million-person increase per decade, $P = 0.001$ for trend), then Africa (an 83.5 million-person increase per decade, $P < 0.001$ for trend) and South America (a 30.4 million-person increase per decade, $P = 0.096$ for trend).

Most of the person-days exposed to SFAP were characterized by substantial fire-sourced PM$_{2.5}$ pollution only (approximately 50% globally) and substantial fire-sourced PM$_{2.5}$ and O$_3$ simultaneously (approximately 45% globally). Fire-sourced PM$_{2.5}$ contributed to SFAP much more than fire-sourced O$_3$ in all continents except North America and Oceania, where around or more than 25% of total exposed person-days were due to substantial fire-sourced O$_3$ only in some years (Extended Data Fig. 2b).

## Socioeconomic disparities in SFAP exposure

Overall, low- and middle-income countries shared more than 96% of global total exposed person-days and over 86% of global total exposed people (Extended Data Table 1b, Extended Data Fig. 2a). The annual average number of days exposed to SFAP was three times greater for countries with a low HDI score and low income (30–45 days per person

per year) than for countries with other HDI scores and income groups (generally <10 days per person per year).

Despite a decreasing trend of the annual exposed days per person, the countries with low HDI scores saw the largest increasing trends for both exposed person-days and exposed people ($P < 0.001$ for trends), whereas the countries with very high HDI scores had the smallest increasing trends in these two metrics (Extended Data Table 1b). This pattern was similar when comparing different income groups.

## Leading countries in exposure

All leading countries (top ten) for five different exposure metrics were low- and middle-income countries, except for the USA, Japan and Chile.

In 2010–2019, the top five countries in population-weighted average fire-sourced PM$_{2.5}$ concentrations were the Democratic Republic of the Congo (DR Congo), the Central African Republic, Angola, Congo and Zambia (all greater than 12 µg m$^{-3}$); the top five countries in population-weighted average fire-sourced O$_3$ concentrations were Congo, DR Congo, the Central African Republic, Burundi and Rwanda (all greater than 23 µg m$^{-3}$) (Fig. 4a,b). The list of countries with the highest annual average number of days exposed to SFAP per person was similar, with Angola, DR Congo, Zambia, Congo and Gabon as the top five countries (all greater than 115 days per year during the period 2010–2019) (Fig. 4d). All top ten countries in these three exposure metrics were sub-Saharan African countries (mostly Central African countries), with three exceptions (Chile, Bolivia and Paraguay, in South America).

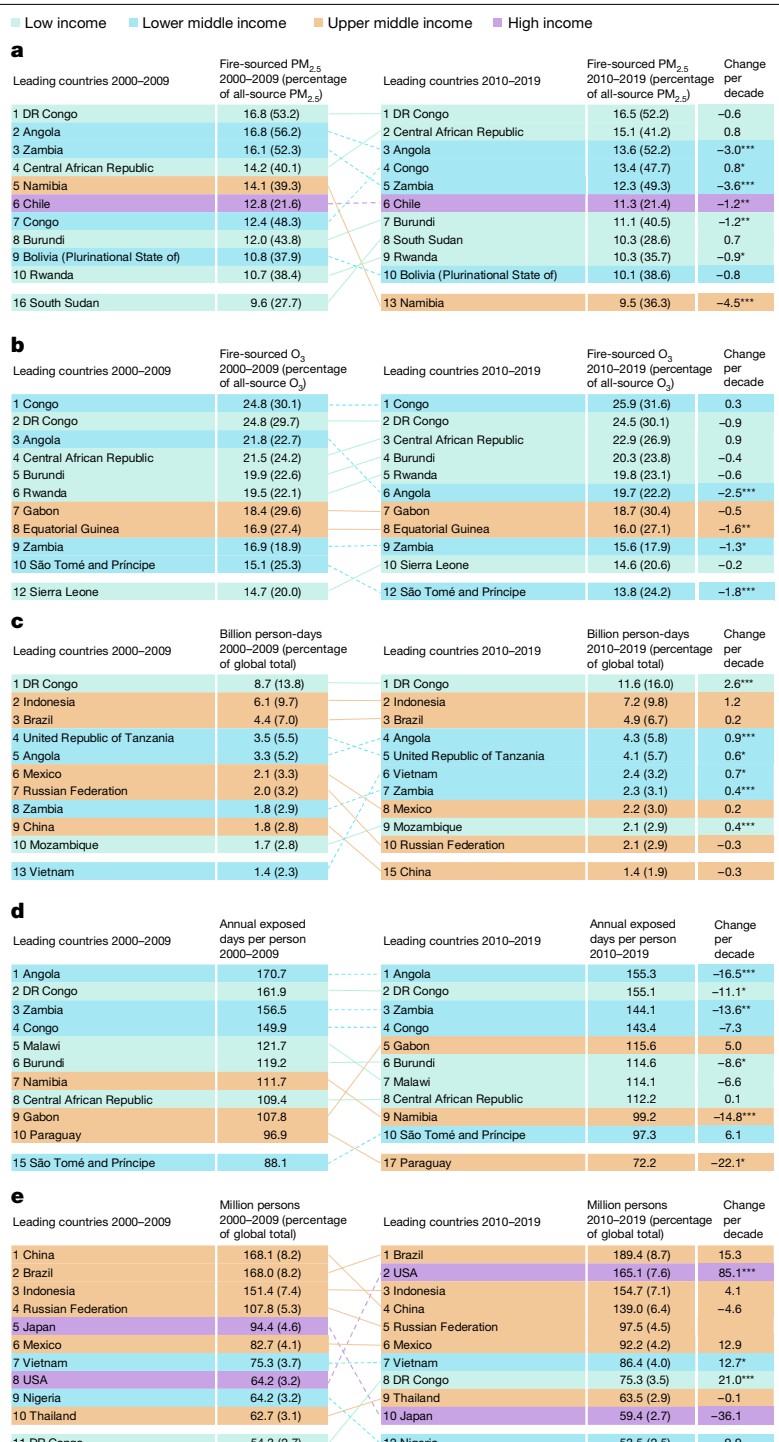

**Fig. 4 | Leading countries with greatest exposures. a–e,** Top ten countries with greatest annual population exposure levels to fire-sourced air pollution in 2000–2009 and 2010–2019, using five different exposure metrics: annual population-weighted average fire-sourced $PM_{2.5}$ concentration ($\mu g\, m^{-3}$) (**a**); annual population-weighted average fire-sourced $O_3$ concentration ($\mu g\, m^{-3}$) (**b**); annual person-days exposed to SFAP (**c**); annual population average number of days exposed to SFAP (**d**); and annual total persons exposed to at least one day of SFAP (**e**). *$P < 0.05$ for long-term trend; **$P < 0.01$; ***$P < 0.001$.

By contrast, the leading countries in total person-days and people exposed to SFAP were more dominated by several populous countries (Fig. 4c,e). In 2010–2019, the top five countries in terms of total exposed person-days were DR Congo (11.6 billion person-days per year), Indonesia (7.2 billion), Brazil (4.9 billion), Angola (4.3 billion) and Tanzania (4.1 billion); the top five countries in terms of total exposed people were Brazil (189.4 million people per year), the USA (165.1 million), Indonesia (154.7 million people), China (139.0 million) and the Russian Federation

(97.5 million). DR Congo had consistently been the country with the largest total exposed person-days in 2000–2009 and 2010–2019, and it showed notable increasing trends in both exposed person-days (an increase of 2.6 billion person-days per decade, $P < 0.001$ for trend) and exposed people (a 21.0 million-person increase per decade, $P < 0.001$ for trend).

The rankings of these exposure metrics changed over time. A notable change was the USA, which ranked only eighth in the total number

of exposed people in 2000–2009, but rose to second in 2010–2019 (an 85.1 million-person increase per decade, $P < 0.001$ for trend).

## Discussion

Through a validated machine learning approach with inputs from chemical transport models, ground-based monitoring stations and gridded weather data[7,8,26], we estimated and mapped the global daily LFS $PM_{2.5}$ and $O_3$ at a 0.25° × 0.25° spatial resolution between 2000 and 2019. This filled a critical data gap, particularly for areas without monitoring stations. With these data and high-resolution global population distribution data, we made by far the most comprehensive assessment of global population exposure to LFS air pollution in the world, to the best of our knowledge.

Our assessment highlighted the severity and scale of the fire-sourced air pollution and a notable increasing trend in the population exposure. Short-term exposure to fire-sourced air pollution has many adverse health impacts, including increased mortality and exacerbations of cardiorespiratory conditions[6,7,27]. The large quantity and increasing trend of the population exposure to SFAP suggests that landscape fire air pollution is an increasing public health concern. Addressing this concern needs multisectoral efforts to reduce landscape fires and prevent adverse health impacts of landscape fire air pollution. Landscape fires can be partially reduced through effective evidence-based fire management, as well as appropriate planning and design of natural and urban landscapes[4]. Policy change may help to reduce some landscape fires caused directly by humans, such as agricultural waste burning in Europe, India, eastern China and the USA (Extended Data Fig. 1b), and the fires deliberately set by humans to convert wildlands to agricultural or commercial lands (common in South America and South and Southeast Asia[24,28]).

However, unplanned wildfires are more difficult to control, as evidenced by the fact that aggressive fire suppression actually contributed to the extreme wildfires in western USA in recent decades because of fuel accumulation[29]. Wildfires are also an essential component of Earth's ecosystem and cannot be totally prevented[4]. Therefore, a considerable proportion of human exposure to LFS air pollution seems to be unavoidable. This highlights the importance of health protection measures against exposure. Unfortunately, existing measures that individuals can take to protect themselves from landscape air pollution, such as relocation, staying indoors, using air purifiers with effective filters and wearing N95 or P100 face masks, all have limitations and are not feasible for people with limited resources[6]; thus it is urgent to develop more cost-effective health protection measures.

The observed increasing global trend of fire-sourced $PM_{2.5}$, although only marginally significant, seems to be inconsistent with the previously reported decline in global burned areas in previous decades[30,31]. However, the decreased global burned areas were mainly in savannas and grasslands because of cropland and pasture expansion, whereas burned areas in forests increased[30,31]. Forests provide much more fuels per unit of burned area than savannas and grasslands[31], and also have a much larger quantity of emissions per unit of dry biomass burned[32]. Therefore, the increased $PM_{2.5}$ emissions from forest fires tends to exceed the decline in $PM_{2.5}$ emissions from savannas and grassland fires. This could explain our observed increasing trend of global fire-sourced $PM_{2.5}$ despite the decline in global burned areas.

It was expected that the temporal trend of fire-sourced $O_3$ was not perfectly consistent with the trend of fire-sourced $PM_{2.5}$. Ground-level or tropospheric $O_3$ is a secondary pollutant generated from photochemical reactions between volatile organic compounds (VOCs) and nitrogen oxides ($NO_x$) under sunlight[33,34]. The generation of fire-sourced $O_3$ can thus be affected by many non-fire factors, such as VOCs and $NO_x$ from industrial and traffic sources, and weather conditions (for example, reduced sunlight during smoky days)[33,34]. In particular, the impacts of VOCs and $NO_x$ emissions on $O_3$ formation are nonlinear[34]; thus whether the $NO_x$ and VOCs emitted from landscape fires can increase the ground-level $O_3$ level is often uncertain. This uncertainty was supported by our results showing that the estimated fire-sourced $O_3$ could even decrease during wildfire periods, compared with pre-wildfire periods, in two out of the ten selected wildfire events (Extended Data Fig. 6b). The relatively uncertain impacts of fires on surface $O_3$ could explain why the global fire-sourced $O_3$ did not show a significant increasing trend like the wildfire $PM_{2.5}$.

Our assessment highlighted the substantial geographical disparities in the population exposures to fire-sourced air pollution. There were several hotspots, including Central Africa, Southeast Asia, South America and Siberia, which experienced the most severe fire-sourced air pollution during the years 2000–2019. North America saw the most significant increases in fire-sourced $PM_{2.5}$ concentrations and the population size exposed to SFAP. The geographical distributions of fire-sourced $PM_{2.5}$ and $O_3$ in our study were generally consistent with a previous map of global landscape fire density[35], but were very different from the global map of meteorological fire danger, that is, the fire weather index (FWI)[36]. For example, the FWI value was very high in North Africa, but low in Central Africa and Siberia. This suggests that the FWI may not be able to capture the actual landscape fire density and the related air pollution, and thus should be used with caution in monitoring and managing landscape fire impacts.

Our assessment also highlighted the socioeconomic disparities in population exposures to fire-sourced air pollution. The disparity could be partly explained by the fact that many low- and middle-income countries are located in hot and dry areas that are prone to landscape fires[4]. The disparity could also be partly due to some other factors, such as that less industrialized countries have more agricultural waste burning and deliberate burning of forests for agricultural or other purposes, and poorer management or control of wildfires[4,37]. More studies are warranted to understand the underlying causes of the disparity, which will help to narrow the gaps. However, our finding does not mean that LFS air pollution is not serious or not important in high-income countries. In fact, we also identified regional hotspots of high levels of fire-sourced air pollution in Australia, the USA and Canada, which were caused by their catastrophic wildfire events in recent years[6]. The value of our study is in highlighting that many low- and middle-income countries have more serious fire-sourced air pollution than that of the high-income countries (for example, the USA, Australia, Canada and western and northern Europe) that attracted the most media and research attention. More attention is needed for those neglected countries to mitigate their fire-sourced air pollution and the related health consequences.

Because the increasing severity and frequency of wildfires are related to anthropogenic climate change[1-4], our finding about the socioeconomic disparities provides further evidence of climate injustice, that is, those least responsible for climate change suffer the most from its consequences[38,39]. A vivid example in our study is the DR Congo, a low-income country with the world's highest fire-sourced $PM_{2.5}$ concentrations. Its anthropogenic carbon dioxide emission per capita was among the lowest in the world (0.03 tons versus the world average of 4.76 tons in 2019[40]). The global socioeconomic disparities in population exposure to fire-sourced air pollution are likely to lead to even larger disparities in health consequences related to the exposure, as poorer countries have more limited resources to protect health against this hazard. This exemplifies how climate change is exacerbating global health inequality. To address this climate injustice, more resources should be allocated to low- and middle-income countries to prevent the health risks from exposure to landscape fire air pollution.

Robust projections suggest that climate change will increase wildfire frequency and intensity in future[4,5,41-43]. Therefore, global fire-sourced air pollution is likely to continue to be an increasingly important public health concern in the next decades. Immediate actions to limit the magnitude of climate change are needed. A projection suggests that wildfire frequency will substantially increase across 74% of the global lands by

2100 under a scenario of high greenhouse gas emissions[43]. However, if the global mean temperature increase could be limited to 2.0 °C or 1.5 °C above pre-industrial levels, over 60% or 80%, respectively, of the increase in wildfire exposure could be avoided[43]. The 1.5 °C target remains reachable, if the world can reduce annual carbon emissions by an extra 28 gigatons of carbon dioxide equivalent (approximately 50% of current emission levels) by 2030[44].

The main strength of our study, compared with previous studies of population exposure assessment of landscape fires, is that we evaluated the population exposure to fire-sourced air pollution, rather than just direct exposure to the flames and heat of landscape fires[21,45]. Fire-sourced air pollution can often travel hundreds (sometimes even thousands) of kilometres and affect much larger populations, causing greater health consequences[4,6]. For example, previous data found that 260,000 people suffered from direct exposure to landscape fires in 2018[45], but this number was only about 0.01% of the population (2.15 billion in 2018) exposed to SFAP. The other data source estimated the annual number of person-days exposed to landscape fires (direct exposure) for each country in the world. Consistent with our study to some extent, it found that DR Congo experienced the largest number of person-days of direct exposure to landscape fires (15,300 person-days per year during 2017–2020)[21]. Again, this number was only about 0.001% of this country's person-days exposed to SFAP (12.0 billion in 2019).

Our study generated a database that can be used for evaluating and tracking the population exposure to LFS air pollution (both $PM_{2.5}$ and $O_3$) across the globe, which is superior to previous studies focusing on fire-related $PM_{2.5}$ in specific regions (the USA[9–18], Europe[19,20] and Brazil[7]). Our estimated fire-sourced $PM_{2.5}$ showed a high level of agreement (Pearson correlation coefficient $r = 0.88$) with the estimated smoke $PM_{2.5}$ by Childs et al.[17]. The high level of agreement with Childs et al.[17] is supported by another study, which found that the summer wildfire smoke $PM_{2.5}$ estimated by the satellite-based smoke plume approach and the GEOS-Chem approach showed generally similar spatial and temporal distribution in the USA over the period 2006–2016[18]. However, the smoke $PM_{2.5}$ estimates of Childs et al.[17] covered only the contiguous USA because it relied on a satellite-based smoke plume polygon product (available only in the contiguous USA and Alaska[46]) to define days and locations covered by landscape fire smoke[17]. The smoke $PM_{2.5}$ tends to be a conservative measure of fire-sourced $PM_{2.5}$ because of the limitations of the satellite-based smoke polygon product, for example, undetected plumes during night time and under cloud cover and in the scenarios when the smoke is dilute and difficult to detect[17]. GEOS-Chem also has some limitations, as we discuss later, so it is still not conclusive which approach is better in terms of accuracy, but the GEOS-Chem approach definitely has the advantage of global coverage.

Two previous studies also used chemical transport model simulations to assess global exposure to fire-sourced $PM_{2.5}$ during 1997–2006 and 2016–2019[22,23]. These two studies observed global spatial distribution patterns of fire-sourced $PM_{2.5}$ that were similar to those observed by us, but our study has the advantage of further calibrating chemical transport model outputs against air quality stations with a machine learning approach. According to our spatial CV and validation against the smoke $PM_{2.5}$ by Childs et al.[17], the calibration approach substantially improved the accuracy of the estimated all-source $PM_{2.5}$ and $O_3$, as well as fire-sourced $PM_{2.5}$ (Extended Data Figs. 4 and 7). With this approach, we also estimated the world's first daily fire-sourced $O_3$ data with global coverage. Moreover, we have a much longer study period and have conducted more comprehensive analysis of the population exposure levels using various metrics for both fire-sourced $PM_{2.5}$ and $O_3$, at several spatial–temporal levels (global/regional/national, yearly/ monthly/daily). Overall, our study provides the most accurate and comprehensive data at present for policymakers and the public to manage and mitigate LFS air pollution at global scale. The generated database also forms a critical basis for many future applications, such

as evaluating various health impacts of this environmental hazard[6], and estimating corresponding attributable mortality, morbidity and health-care costs[19,22].

Several limitations of our study should be acknowledged. $PM_{2.5}$, $O_3$ and carbon monoxide (CO) are the main pollutants of public health concern during wildfire events[47], but we did not quantify CO from landscape fires because of data unavailability. Previous studies suggest that the impacts of wildfires on CO are generally confined to the immediate fire areas[9,48], which can be explained by the photochemical loss of CO (that is, photochemical oxidation of CO and hydrocarbons in the presence of nitrogen oxides produces $O_3$) during long-distance transport of biomass plumes[49]. Therefore, the unavailability of CO would be expected to have minimal impact on the estimation of the population exposure to fire-related air pollution. Other limitations, including the uncertainties of the fire emission inventory, the GEOS-Chem simulations, and machine learning models, are discussed in detail in Methods.

In conclusion, we conducted a comprehensive assessment of global population exposure to LFS air pollution. We found that billions of people worldwide were exposed to substantial LFS air pollution, and the exposure levels were particularly high in several hotspots (Central Africa, Southeast Asia, South America and Siberia) and in the least developed countries.

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

## Methods

### Data collection

**Monitoring station data.** We collected global air quality monitoring station data from several sources. Monitoring data for the USA were downloaded from the US Environmental Protection Agency (US EPA)[50]. Data for China were downloaded from the China National Environmental Monitoring Centre (http://www.cnemc.cn/en/). Data for member countries of the European Economic Area were downloaded from the European Environment Agency[51]. Data for Australia were sourced from the National Air Pollution Monitoring Database, which integrated all available monitoring data from Australian state-specific governmental agencies[52,53]. Data for New Zealand were downloaded from Environment Canterbury (http://data.ecan.govt.nz/Catalogue/Method?MethodId=98). Data for Chile were downloaded from its National Air Quality Information System (https://sinca.mma.gob.cl/index.php/region/index/id/II). Data for South Africa were downloaded from the South African Air Quality Information System (https://saaqis.environment.gov.za/). Data for two African countries (Algeria and Nigeria) were downloaded from AirQo (https://www.airqo.net/, only PM$_{2.5}$ data available).

Data for other countries and territories were downloaded from OpenAQ (https://openaq.org/). To ensure data quality, we used data from reference-grade monitoring stations only.

After a data cleaning and quality control process (Supplementary Information), we kept 9,528,179 valid daily average PM$_{2.5}$ observations of 5,661 stations from 73 countries and territories and 21,097,834 valid daily 8 h maximum O$_3$ observations of 6,851 stations from 58 countries and territories (Supplementary Tables 1 and 2 and Extended Data Fig. 3). Both PM$_{2.5}$ and O$_3$ station data covered the whole period between 2000 and 2019, although the data period varied by country and stations. We unified all units of PM$_{2.5}$ and O$_3$ as µg m$^{-3}$, consistent with the latest WHO air quality guidelines 2021[54]. For O$_3$, 1 part per billion (ppb) was approximated as 1.96 µg m$^{-3}$, assuming a standard air pressure and temperature (25.5 °C and 101.325 kPa)[55].

**Chemical transport model simulations.** As described previously[7,8,26], we used the three-dimensional chemical transport model GEOS-Chem (v.12.0.0) based on O$_3$–NO$_x$–hydrocarbon–aerosol chemical mechanisms to estimate daily total (that is, all-source) and fire-sourced PM$_{2.5}$ and O$_3$ concentrations at 2.0° latitude by 2.5° longitude horizontal resolution (about 220 km × 280 km) during years 2000–2019 across the globe. Daily fire-sourced PM$_{2.5}$ and O$_3$ concentrations were estimated as the differences between GEOS-Chem simulations with and those without fire emissions. The fire emission data came from the Global Fire Emissions Database (v.4.1 with small fires, GFED4.1s)[56], which captured aerosol emissions from six fire sources (boreal forest fires; tropical forest fires; savanna, grassland and shrubland fires; temperate forest fires; peatland fires; and agricultural waste burning) according to satellite retrieval of burned areas and active fire information[26]. On the basis of the GFED4.1s data, the relative contributions of different fire types to the fire-emitted PM$_{2.5}$ varied by continent (for example, North America and Asia are characterized by high proportions of boreal forest fires; Oceania and Africa by savanna, grassland and shrubland fires; South America by tropical forest fires; and Europe by agricultural fires) (Extended Data Fig. 1c). We also provide the dominant landscape fire type burned during the period 2000–2019 at 0.25° × 0.25° spatial resolution across the globe in Extended Data Fig. 1b, which suggests that the peatland fires burned mainly in Southeast Asia.

**Meteorological data.** We derived hourly meteorological data at 0.25° × 0.25° spatial resolution from the fifth-generation European Centre for Medium-Range Weather Forecasts Reanalysis (ERA5)[57]. ERA5 combines model results with worldwide weather observations into a globally complete and consistent dataset using the laws of physics. Hourly records were used to calculate daily metrological parameters according to the local time zone of each grid. These daily metrological parameters included daily mean/minimum/maximum 2 m (that is, at 2 m above the surface of the earth) ambient temperature ($T_{mean}$, $T_{min}$ and $T_{max}$, all calculated from 24-hourly records of 2 m ambient temperature), daily temperature variability (TV, standard deviations of 24-hourly 2 m temperatures), daily mean 2 m dew point temperature ($T_{dew\_mean}$), daily mean eastward component of 10 m wind (Wind_u, 10 m refers to 10 m above the surface of the earth), daily mean northward component of 10 m wind (Wind_v), daily total precipitation (Precip), daily mean surface air pressure (Pressure) and daily mean downward ultraviolet radiation at the surface (UV). Daily mean relative humidity (RH) was calculated from $T_{mean}$ and $T_{dew\_mean}$ using the humidity R package[58].

**Population data.** We collected annual population count data at 30 arcseconds (about 1 km$^2$) spatial resolution across the globe during the years 2000–2019 from the WorldPop project[59]. Specifically, we downloaded the unconstrained global mosaics data (approximately 1 km × 1 km spatial resolution). This dataset was generated using the top-down unconstrained approach to disaggregate administrative unit-based census and projection counts for each year to grid cell-based population counts, by using a set of detailed geospatial predictors and a random forest machine learning model[60]. We aggregated the gridded population counts to 0.25° × 0.25° spatial resolution to match the air pollution data. For each country or territory in each year, all grid-specific population counts within its boundary were further multiplied by an adjustment coefficient (that is, the population size of that country or territory reported by the United Nations/sum of all grid-specific population counts within the boundary). This adjustment ensured that the country-specific population counts were consistent with data from the United Nations[61].

**Socioeconomic data.** Countries were classified as low-income countries (gross national income (GNI) per capita ≤ US$1,035), lower-middle-income countries (US$1,035 < GNI per capita ≤ US$4,045), upper-middle-income countries (US$4,045 < GNI per capita ≤ US$12,535) and high-income countries (GNI per capita > US$12,535) according to the World Bank's 2019 criteria[62]. Country-level HDI data in 2019 were downloaded from the United Nations Development Programme (UNDP). HDI is a unified measure of average achievement in key dimensions of human development, including a long and healthy life, being knowledgeable (educated) and having a decent standard of living. HDI scores range from 0 to 1, and can be divided into four tiers: very high (0.8 to 1.0), high (0.70 to 0.79), medium (0.55 to 0.69) and low (less than 0.55)[63].

### Estimating fire-sourced PM$_{2.5}$ and O$_3$

We estimated global fire-sourced PM$_{2.5}$ and O$_3$ at 0.25° × 0.25° spatial resolution with three steps. In step one, we downscaled daily total and fire-sourced PM$_{2.5}$ and O$_3$ derived from GEOS-Chem to 0.25° × 0.25° spatial resolution using the inverse distance weighted spatial interpolation[8,64].

In step two, downscaled GEOS-Chem outputs were further calibrated to match ground monitoring station observations based on a random forest machine learning algorithm. Briefly, the downscaled GEOS-Chem outputs and gridded meteorological data were linked to ground monitoring stations based on longitude and latitude, which generated the model training datasets. Then we trained two random forest models to predict station-observed total PM$_{2.5}$ (PM$_{2.5\_station}$) and O$_3$ (O$_{3\_station}$) separately, with the following equations:

$$PM_{2.5\_station} = f(PM_{2.5\_chem\_total}, T_{mean}, T_{max}, T_{min}, TV, RH, Wind\_u, Wind\_v, Precip, Pressure, UV, Year, Month, DOW, DOY, Lon, Lat) \quad (1)$$

$$O_{3\_station} = f(O_{3\_chem\_total}, T_{mean}, T_{max}, T_{min}, TV, RH, Wind\_u,$$
$$Wind\_v, Precip, Pressure, UV, Year, Month, DOW, \quad (2)$$
$$DOY, Lon, Lat)$$

$PM_{2.5\_chem\_total}$ and $O_{3\_chem\_total}$ were downscaled daily total (all-source) $PM_{2.5}$ and $O_3$ derived from GEOS-Chem. $T_{mean}$ to UV were ERA5 meteorological variables, as mentioned above. DOW was day of week (Monday to Sunday). DOY was day of year (1 to 366). Lon and Lat were longitude and latitude, respectively. $f$ referred to the random forest algorithm fitted with the ranger R package[65].

In step three, the daily total (all-source) $PM_{2.5}$ ($PM_{2.5\_est\_total}$) and $O_3$ ($O_{3\_est\_total}$) for each $0.25° \times 0.25°$ grid (regardless of whether close to or far away from the training stations) across global lands were estimated using the trained random forest models (that is, machine learning calibration or bias correction algorithms found where training stations existed) and global seamless predictor data. Then the final estimated fire-sourced $PM_{2.5}$ ($PM_{2.5\_est\_fire}$) and $O_3$ ($O_{2.5\_est\_fire}$) were calculated as follows[7,8,64]:

$$PM_{2.5\_est\_fire} = PM_{2.5\_est\_total} \times (PM_{2.5\_chem\_fire} / PM_{2.5\_chem\_total}) \quad (3)$$

$$O_{2.5\_est\_fire} = O_{3\_est\_total} \times (O_{3\_chem\_fire} / O_{3\_chem\_total}) \quad (4)$$

The $PM_{2.5chem\_fire}$ and $O_{3\_chem\_fire}$ refer to the downscaled fire-sourced $PM_{2.5}$ and $O_3$ from GEOS-Chem.

## Model performance evaluation

We used tenfold CV to test the performance of the random forest models and to find the optimal model parameters. Specifically, the whole model training dataset was randomly divided into ten approximately equal subsets. Each subset was then treated as a validation set to test the performance of the model trained in the remaining nine subsets (this was repeated ten times)[66]. We also used a spatial tenfold CV (that is, dividing all stations, rather than the dataset, into ten approximate equal subsets, then performing CV in a manner similar to that described above) to test the model's prediction ability in new locations not in the training data (that is, spatial generalization ability of the model).

We tested the spatial generalization ability of the models further using a spatial cluster-based CV approach. Specifically, we conducted a $k$-means cluster analysis[67] based on the Euclidean distances between stations based on their longitude and latitude, and the optimal number of spatial clusters was determined by selecting the minimum sum-of-squares distances within groups. As a result, we identified 75 spatial clusters for $PM_{2.5}$ stations and 99 spatial clusters for $O_3$ stations across the globe. We then used each cluster as a testing dataset and the remaining clusters as the training dataset to train and test our random forest model 75 and 99 times for $PM_{2.5}$ and $O_3$, respectively. Compared with spatial tenfold CV, in which the nearby stations could be allocated to training and testing datasets simultaneously, the spatial cluster-based CV increases the difficulty of the prediction task[68] but is a more realistic test of the models' prediction abilities in large remote areas with essentially no training stations (for example, many areas in Africa and South America; Extended Data Fig. 3a,b).

The model reached a high level of accuracy in estimating both daily average $PM_{2.5}$ (tenfold CV, $R^2 = 0.91$, RMSE = 8.47 µg m$^{-3}$) and daily maximum 8 h $O_3$ (tenfold CV, $R^2 = 0.82$, RMSE = 18.96 µg m$^{-3}$) (Extended Data Fig. 3e). The model also showed a similarly high level of accuracy in the spatial tenfold CV for both $PM_{2.5}$ ($R^2 = 0.89$, RMSE = 9.24 µg m$^{-3}$) and $O_3$ ($R^2 = 0.80$, RMSE = 19.64 µg m$^{-3}$) (Extended Data Fig. 3f), suggesting good spatial generalization abilities of the trained random forest models.

We calculated station-specific $R^2$ based on the spatial tenfold CV. The median station-specific model performance among stations was comparable to overall model performance (median station-specific $R^2$,

0.80 for $PM_{2.5}$ and 0.72 for $O_3$), with 90% of station-specific $R^2$ values above 0.38 for $PM_{2.5}$ and above 0.53 for $O_3$. There were notable spatial variations of the station-specific model performance (Extended Data Fig. 3c,d). Although the model estimates showed a high level of agreement with station observations in most stations, a low level of agreement between model estimates and station observations was found in some $PM_{2.5}$ stations in the middle and southwestern USA, Hawaiian islands, southern Europe, Africa, and western and inland Australia, and some $O_3$ stations in Chile, South Africa and New Zealand.

We also estimated the within-$R^2$ value of the spatial tenfold and cluster-based CV. The within-$R^2$ value was calculated by regressing station observations on model estimates while controlling for the station and year fixed effects. As calculated by the *fixest* R package[69], the within-$R^2$ value of spatial tenfold CV was 0.81 and 0.74 for $PM_{2.5}$ and $O_3$, respectively (Extended Data Fig. 4a). This suggests that our random forest models can predict, on average, 81% and 74% of local temporal daily variations of all-source $PM_{2.5}$ and $O_3$, respectively, within a year, not just variations in average $PM_{2.5}$ and $O_3$ across locations and years.

As expected, the model performance of spatial cluster-based CV ($PM_{2.5}$, $R^2 = 0.69$, RMSE = 14.79 µg m$^{-3}$; $O_3$, $R^2 = 0.67$, RMSE = 18.14 µg m$^{-3}$) was lower than spatial tenfold CV, but still much higher than the performance of the raw GEOS-Chem outputs ($PM_{2.5}$, $R^2 = 0.48$, RMSE = 31.00 µg m$^{-3}$; $O_3$, $R^2 = 0.47$, RMSE = 46.81 µg m$^{-3}$) across the globe and in all continents. This suggests that our models can predict the daily all-source $PM_{2.5}$ and $O_3$ in large remote areas with no training data with an accuracy that is much higher than that of the raw GEOS-Chem outputs alone. Similarly, the within-$R^2$ values for spatial cluster-based CV suggest that the model estimates can explain 41% and 52% of local temporal daily variations of all-source $PM_{2.5}$ and $O_3$, respectively, in spatial clusters not in the training data.

## Validation against smoke PM$_{2.5}$

Childs et al.[17] trained a machine learning model to predict the station-based smoke $PM_{2.5}$ using meteorological factors, fire variables, aerosol measurements, and land use and elevation data, and the model was used to estimate daily smoke $PM_{2.5}$ across the USA at $0.1° \times 0.1°$ spatial resolution during the period 2006–2020. In their study, the station-based observed smoke $PM_{2.5}$ was calculated through two steps: (1) days when smoke was overhead were defined as 'smoke days' based on satellite imagery-based plume classification (or simulated air trajectories originating at fires when clouds may be obscuring plumes), and days without smoke overhead were 'non-smoke days'; (2) for each station on each smoke day, the observed smoke $PM_{2.5}$ concentration was calculated as the station-observed all-source $PM_{2.5}$ on the smoke day minus the background $PM_{2.5}$, which was defined as the 3-year (previous year, current year and the next year) station- and month-specific median $PM_{2.5}$ on non-smoke days. For example, if a smoke day was 10 January 2018 for station A, then its corresponding background $PM_{2.5}$ was the median value of all daily $PM_{2.5}$ observations of station A on non-smoke days in the January of each year during 2017–2019. The smoke $PM_{2.5}$ on non-smoke days was assumed to be 0.

Because most of the training stations used by Childs et al.[17] were also our training stations, directly validating our estimated fire-sourced $PM_{2.5}$ against the observed smoke $PM_{2.5}$ by the training stations of Childs et al.[17] may have overfitting issues (that is, may overestimate our accuracy). To avoid this problem, we chose the observed smoke $PM_{2.5}$ of PurpleAir stations (a kind of low-cost sensor) as our validation target. The PurpleAir stations were not included in our training stations nor in those of Childs et al.[17], and thus could give a fair comparison of the accuracy of our estimated fire-sourced $PM_{2.5}$ and the estimated smoke $PM_{2.5}$ of Childs et al.[17]. The PurpleAir station data were collected and cleaned as detailed previously[17], and its measured daily $PM_{2.5}$ had been calibrated against US EPA reference-grade stations by Childs et al.[17] before calculating its station-observed smoke $PM_{2.5}$.

When validated against the PurpleAir station-observed smoke $PM_{2.5}$, our estimates' accuracy ($R^2 = 0.51$, RMSE = 11.76 μg m$^{-3}$) was lower than the accuracy of the estimates of Childs et al.[17] ($R^2 = 0.66$, RMSE = 10.46 μg m$^{-3}$), but much higher than the accuracy of the fire-sourced $PM_{2.5}$ from raw GEOS-Chem outputs ($R^2 = 0.18$, RMSE = 22.96 μg m$^{-3}$) (Extended Data Fig. 4b). Our estimated fire-sourced $PM_{2.5}$ values were highly correlated with the estimated smoke $PM_{2.5}$ by Childs et al.[17] (Pearson correlation coefficient $r = 0.88$) (Extended Data Fig. 4c).

We also calculated with-block $R^2$ by regressing station observations on model estimates while controlling for the block (that is, the $2.0° \times 2.5°$ grid box of GEOS-Chem simulations) and date fixed effects. We found that our estimated fire-sourced $PM_{2.5}$ could account for 10% (within-block $R^2 = 0.10$) of the spatial variations of the PurpleAir station-observed smoke $PM_{2.5}$ within the $2.0° \times 2.5°$ grid box for each day. Although this is lower than the within-block spatial variations accounted for by the estimates of Childs et al.[17] (within-block $R^2 = 0.32$), it suggests that our model can explain some spatial variations of fire-sourced $PM_{2.5}$ at a resolution higher than the resolution of GEOS-Chem simulations, after downscaling of the GEOS-Chem outputs, machine learning calibration and including meteorological data inputs at $0.25° \times 0.25°$ spatial resolution.

## Validation against wildfire events

As detailed in the Supplementary Information, we chose ten large wildfire events in Australia, the USA, Chile, Portugal and South Africa to validate our estimated all-source and fire-sourced $PM_{2.5}$ and $O_3$ (Supplementary Table 3). According to the results (Extended Data Figs. 5 and 6), during the wildfire event and up to 60 days before and after the event, the observed daily all-source $PM_{2.5}$ or $O_3$ from the most affected monitoring stations (that is, a nearby station showing the largest increase in observed concentrations during the wildfire event, compared with the pre-wildfire period, for each event) showed moderate to strong correlation with our estimated daily all-source $PM_{2.5}$ ($r$, 0.44–0.85; pooled $R^2$ across wildfire events = 0.64) and $O_3$ ($r$, 0.54–0.92; pooled $R^2 = 0.78$), based on the model trained in the data excluding nearby stations. Furthermore, these was an increase in the estimated concentrations and proportions (among all sources) of fire-sourced $PM_{2.5}$ during all the selected wildfire events, compared with the pre-wildfire period (Extended Data Fig. 5b). There was also an increase in the estimated concentrations and proportions (among all sources) of fire-sourced $O_3$ during eight of the ten selected wildfire events (Extended Data Fig. 6b); the two exceptions in which there was decreased fire-sourced $O_3$ during the wildfire period could be explained by the uncertain impacts of wildfires on ambient $O_3$ (Discussion). Overall, the results indicate that our models can reasonably capture the wildfires' contribution to the all-source and fire-sourced $PM_{2.5}$ and $O_3$.

## Mapping population exposure

The estimated global fire-sourced $PM_{2.5}$ and $O_3$ during the period 2000–2019 were linked with global population distribution data to map the global population exposure to daily LFS air pollution. The population exposure was measured by four metrics: (1) population-weighted average fire-sourced $PM_{2.5}$ and $O_3$ concentrations (that is, average of all grids weighted by population count of each grid); (2) annual number of person-days exposed to SFAP, with 1 person-day referring to one person exposed to 1 day of SFAP; (3) annual average number of days per person exposed to SFAP, equal to the metric 2 divided by total population size; and (4) annual total number of people exposed to at least 1 day of SFAP.

A day with SFAP should consist of at least one of the following scenarios: (1) the daily average $PM_{2.5}$ (all-source $PM_{2.5}$) exceeded the WHO's 2021 daily guideline value (15 μg m$^{-3}$), and fire-sourced $PM_{2.5}$ accounted for at least 50% of the daily all-source $PM_{2.5}$, and (2) the daily maximum 8 h $O_3$ (all-source $O_3$) exceeded the WHO's 2021 daily guideline value (100 μg m$^{-3}$), and fire-sourced $O_3$ accounted for at least 50% of the daily all-source $O_3$.

All descriptive analyses were at global scale, and by continent (Africa, Asia, Europe, North America, South America and Oceania), country or territory, HDI group and income group, for each year from 2000 to 2019. Our analyses included 206 countries or territories covered by the ERA5 land grids. We tested the long-term trend of each metric using linear regressions, with the annual metrics during the period 2000–2019 as the dependent variable and year (numeric) as the only predictor.

## Sensitivity analyses

In our primary analyses, we used the GFED4.1s as the fire emission inventory of the GEOS-Chem simulations. However, previous studies in North America found that chemical transport model simulations based on different fire emission inventories generated very different estimates of fire-sourced $PM_{2.5}$ and $O_3$[70,71]. Therefore, apart from the GFED4.1s[56], we also collected data from three other widely used global fire emission inventories: the Fire INventory from the National Center for Atmospheric Research (NCAR) v.1.6 (FINN1.6)[72], the Quick Fire Emission Dataset v.2.5 (QFED2.5)[73] and the Global Fire Assimilation System v.1.2 (GFAS1.2)[74]. Each inventory has its own advantages and disadvantages; thus we cannot decide which one is best without validation against real-world observations, although the GFED4.1s is the one with the best data availability (Supplementary Table 4).

Because a previous study suggested that the largest difference of population-weighted fire-sourced $PM_{2.5}$ estimates in North America between four different fire emission inventories was observed in 2012[70], we ran GEOS-Chem simulations for 2012 using GFED, FINN, QFED and GFAS separately, and performed the aforementioned machine learning calibrations against air quality station data. To ensure comparability, we used the same station and linked predictor data that were available in 2012 in model training and tenfold spatial CV for all four fire emission inventories. We also validated the estimated fire-sourced $PM_{2.5}$ based on different inventories against the station-observed smoke $PM_{2.5}$ in 2012 provided by Childs et al.[17].

According to the validation results, the estimated all-source $PM_{2.5}$ and $O_3$ based on different fire emission inventories were highly consistent with each other ($r$, 0.99 or above), and they showed very similar accuracy in validation against station observations (spatial tenfold CV $R^2$ for different inventories, 0.75–0.76 for $PM_{2.5}$ and all 0.82 for $O_3$; RMSE, 5.65–5.72 μg m$^{-3}$ for $PM_{2.5}$ and all 12.52 μg m$^{-3}$ for $O_3$) (Extended Data Fig. 7). When validated against the station-observed smoke $PM_{2.5}$, the GFED-, GFAS- and QFED-based fire-sourced $PM_{2.5}$ values showed similar accuracy (spatial tenfold CV $R^2$, 0.27–0.30; RMSE, 8.35–8.75 μg m$^{-3}$), whereas the FINN-based estimates showed the least accuracy ($R^2 = 0.19$, RMSE = 9.83 μg m$^{-3}$). The GFED-based fire-sourced $PM_{2.5}$ showed good agreement with FINN-, GFAS- and QFED-based estimates ($r$, 0.73, 0.81 and 0.83, respectively). The GFED-based fire-sourced $O_3$ showed moderate agreement with FINN-based estimates ($r$, 0.57), good agreement with GFAS-based estimates ($r$, 0.72) and poor agreement with QFED-based estimates ($r$, 0.30); the QFED-based fire-sourced $O_3$ showed even poorer agreement with FINN- and GFAS-based estimates ($r$, 0.18 and 0.16, respectively).

Because FINN showed least agreement with the GFED-based estimates of fire-sourced $PM_{2.5}$ (the main contributor to SFAP), we performed sensitivity analyses by running GEOS-Chem simulations based on FINN for all its available years (2002–2017), and generated the daily FINN-based estimates of fire-sourced $PM_{2.5}$ and $O_3$ at $0.25° \times 0.25°$ spatial resolution using the same machine learning calibration procedures as in our primary analyses. Compared with GFED-based estimates during the period 2002–2017, the FINN-based estimates of fire-sourced $PM_{2.5}$ and $O_3$ showed very similar spatial distribution (GFED versus FINN agreement in grid-specific 16-year average concentrations, $r = 0.93$ for fire-sourced $PM_{2.5}$, $r = 0.92$ for fire-sourced $O_3$), temporal trends (GFED versus FINN agreement in grid-specific change in concentrations per year, $r = 0.80$ for both fire-sourced $PM_{2.5}$ and $O_3$), continent-specific long-term trends and seasonal patterns (Extended Data Figs. 8 and 9).

## Uncertainties of our estimates

There were some uncertainties or potential errors in the processes of estimating fire-sourced $PM_{2.5}$ and $O_3$. First, the GFED4.1s used for GEOS-Chem simulations has some uncertainties and limitations, such as uncertainties in the emission factor and the estimation of burned areas based on satellite images[56]. Studies suggest that GEOS-Chem simulations based on different fire emission inventories may generate very different estimates of fire-sourced $PM_{2.5}$ in North America[70,71]. However, according to our validation results (Extended Data Fig. 7 and Supplementary Table 4), the GFED4.1s was the best inventory of the four widely used inventories considering both accuracy (that is, agreement with ground station observations and the smoke $PM_{2.5}$ of Childs et al.[17]) and data availability, and it is also the most widely used one at present[70]. Our results also suggest that the estimates of all-source and fire-sourced $PM_{2.5}$ and $O_3$ based on three alternative inventories were mostly highly consistent with GFED-based estimates, and the consistency improved after machine learning calibrations (Extended Data Fig. 7). Furthermore, even based on FINN (the inventory that showed the least agreement with GFED-based estimates of fire-sourced $PM_{2.5}$), the generated estimates of fire-sourced $PM_{2.5}$ and $O_3$ showed spatial distribution, temporal trends and seasonal patterns that were very similar to GFED-based estimates (Extended Data Figs. 8 and 9). Therefore, our assessment of population exposure to fire-sourced air pollution was robust against the choice of fire emission inventory.

Second, our GEOS-Chem simulations did not account for plume rise and assumed that all fire emissions were emitted at the surface, because there are large uncertainties in the fire plume height data[75,76], and a recent study found that including the fire plume rise did not always improve the accuracy of simulated $PM_{2.5}$ and $O_3$[77]. GEOS-Chem simulations without considering plume rise can overestimate the contribution of fire emissions to surface $PM_{2.5}$ and $O_3$ in fire source regions while underestimating the impacts of fire emissions in regions downwind from the fire source[75,77]. Given that fire source regions (for example, wildlands or agricultural lands) tend to have smaller population densities than other regions, our GEOS-Chem approach is likely to cause an underestimation of global population exposure to fire-sourced air pollution. Further studies are warranted to quantify and correct the bias caused by omitting plume rise.

Third, the GEOS-Chem was run at a coarse spatial resolution ($2.0° × 2.5°$), which may cause errors in population exposure assessment at high spatial resolution. However, we have performed downscaling of the GEOS-Chem and added higher-resolution meteorological data as extra predictors in the machine learning model. The validation against observed smoke $PM_{2.5}$ in PurpleAir stations suggested that our estimated fire-sourced $PM_{2.5}$ can explain about 10% of spatial variations of the observed smoke $PM_{2.5}$ within the large $2.0° × 2.5°$ grid box (Extended Data Fig. 4b), which was a big improvement compared with the raw GEOS-Chem outputs. Moreover, there was almost no correlation between grid-specific population counts and the annual fire-sourced $PM_{2.5}$ ($r = -0.02$) and $O_3$ ($r = 0.001$) concentrations in our data, suggesting that the bias in concentration caused by coarse-resolution of GEOS-Chem tend to be distributed to $0.25° × 0.25°$ grid boxes with high or low population counts randomly and cause random errors, rather than systematic errors of population exposure assessment. Nevertheless, cautions should be taken if our data are used to perform individual-level exposure assessment in epidemiological studies.

Finally, the machine learning models were trained against station observations dominated by several regions (Europe, the USA and China), which may not apply to regions with few or no stations. However, according to our spatial-cluster CV that mimics this situation, our models showed good accuracy in predicting observations far away from the training stations (overall $R^2$, 0.69 for $PM_{2.5}$ and 0.67 for $O_3$), and the accuracy was still much higher than the raw GEOS-Chem outputs even in continents (Africa, South America and Oceania) with a small number of stations (Extended Data Fig. 4a), suggesting that our trained machine learning model can also add accuracy to the GEOS-Chem in regions with limited or no training stations.

We performed the downscaling of GEOS-Chem outputs using ArcGIS desktop (v.10.1); all other data analyses were performed using R software (v.4.0.2).

## Data availability

Data of air quality stations are available for free or with certain conditions from the US Environmental Protection Agency (https://aqs.epa.gov/aqsweb/airdata/download_files.html), the China National Environmental Monitoring Centre (http://www.cnemc.cn/en/), the European Environment Agency (https://www.eea.europa.eu/data-and-maps/data/aqereporting-9), the Australian National Air Pollution Monitor Database (http://cardat.github.io/), New Zealand's Environment Canterbury (http://data.ecan.govt.nz/Catalogue/Method?MethodId=98), the Chilean National Air Quality Information System (https://sinca.mma.gob.cl/index.php/region/index/id/II), the South African Air Quality Information System (https://saaqis.environment.gov.za/), AirQo (https://www.airqo.net/) and OpenAQ (https://openaq.org/). The cleaned air quality station data used for this study were deposited at https://doi.org/10.17605/OSF.IO/DN7YA. Data of weather predictors are open access and are available from https://cds.climate.copernicus.eu/cdsapp#!/dataset/reanalysis-era5-single-levels?tab=overview. Population exposure estimates globally, for different continents, HDI and income groups, and for 206 countries and territories were shared on https://github.com/Rongbin553/wildfire_population. The GEOS-Chem simulation outputs and estimated all-source and fire-sourced air pollution data are available from the corresponding authors on request, and will be made open access at https://doi.org/10.17605/OSF.IO/DN7YA after the paper is published. Source data are provided with this paper.

## Code availability

Analysis codes are available from the corresponding authors on request, and will be shared on https://github.com/Rongbin553/wildfire_population.

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

**Acknowledgements** This study was supported by the Australian Research Council (grant no. DP210102076) and the Australian National Health and Medical Research Council (grant no. GNT2000581). Y.G. was supported by a Career Development Fellowship (grant no. GNT1163693) and a Leader Fellowship (grant no. GNT2008813) from the Australian National Health and Medical Research Council. S.L. was supported by an Emerging Leader Fellowship from the Australian National Health and Medical Research Council (grant no. GNT2009866). R.X. was supported by Monash Faculty of Medicine Nursing and Health Science (FMNHS) Bridging Postdoctoral Fellowships 2022. X.Y. was supported by Jiangsu Science Fund for Distinguished Young Scholars (grant no. BK20200040). T.Y., P.Y. and Y.W. were supported by the China Scholarship Council (grant nos 201906320051, 201906210065 and 202006010044, respectively). Z.Y. and W.Y. were supported by the Monash Graduate Scholarship and the Monash International Tuition Scholarship. We thank G. Chen for assistance in GEOS-Chem data preparation. We thank A. Haines, A. Woodward and K. L. Ebi for their comments and suggestions. The funding bodies did not play any role in study design, data collection, data analyses, results interpretation or writing of this manuscript.

**Author contributions** R.X., Y.G. and S.L. conceived and conceptualized the study. Y.G. and S.L. designed the methodology. X.Y. and Y. Lei performed the GEOS-Chem simulations. R.X., Z.Y., Yuxi Z. and Y.W. collected and cleaned the air quality station data. R.X. performed most of the statistical analyses and wrote the first manuscript draft. T.Y., Z.Y. and Y. Liu contributed to visualization of the results. W.Y. and Yiwen Z. contributed to machine learning model validation work. T.Y., P.Y., Z.Y. and Y.W. assisted with the collection and cleaning of ERA5 weather data and gridded population data. M.L.B, L.M, P.Y., F.J and M.J.A provided important comments and made critical editing of the manuscript. Y.G. and S.L. oversaw and coordinated the whole study.

**Competing interests** Michael J. Abramson holds investigator-initiated grants from Pfizer, Boehringer-Ingelheim, Sanofi and GSK for unrelated research. He has undertaken an unrelated consultancy for Sanofi. He also received a speaker's fee from GSK. The other authors declare no competing interests.

**Additional information**
**Correspondence and requests for materials** should be addressed to Xu Yue, Yuming Guo or Shanshan Li.

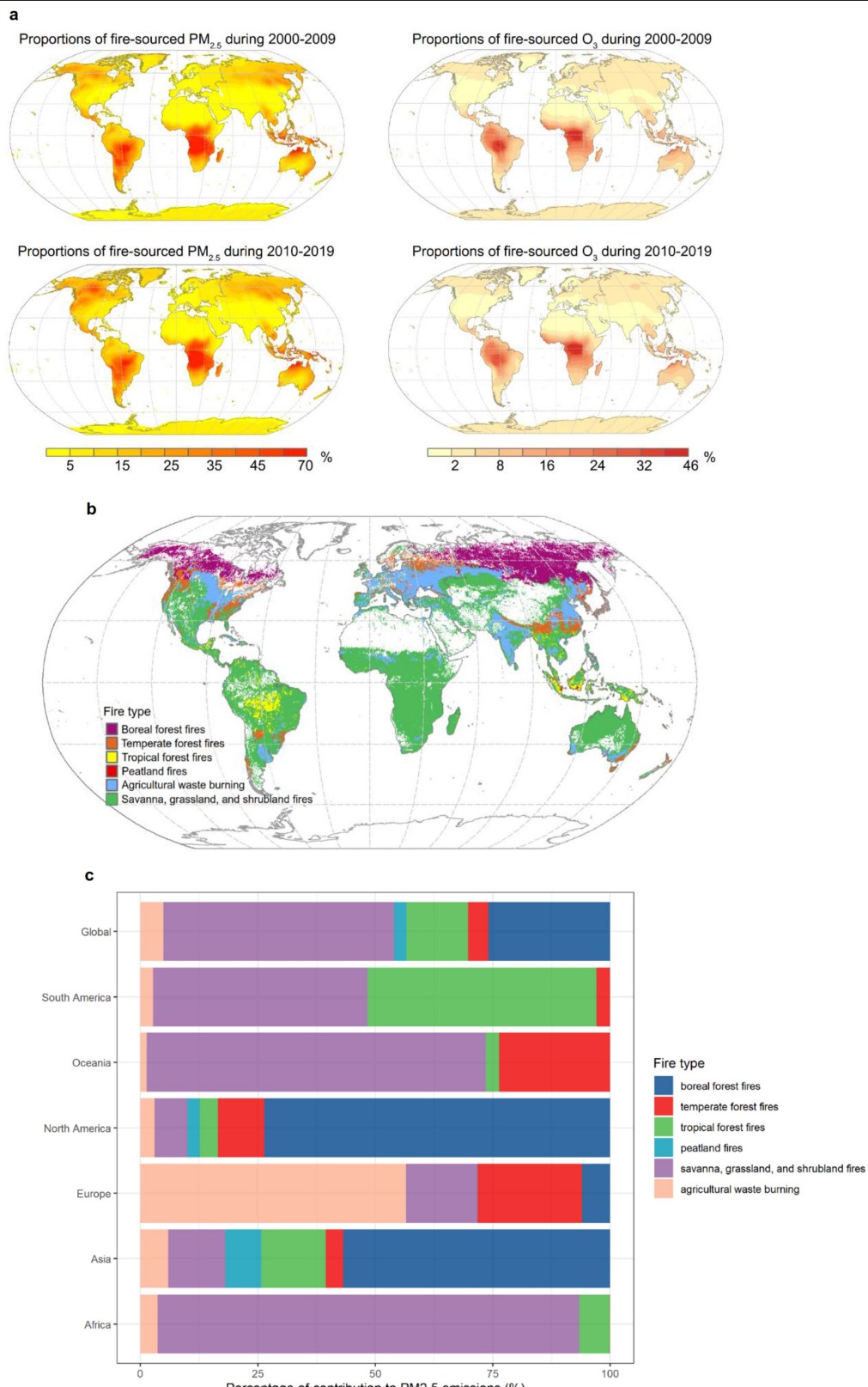

**Extended Data Fig. 1 | Landscape fires' relative contributions to air pollution and the global distribution and relative contributions of different fire types.** **a**, Maps showing the proportions of fire-sourced $PM_{2.5}$ and $O_3$ (among all-source $PM_{2.5}$ and $O_3$, respectively) during 2000–2009 and 2010–2019 across the globe. **b**, Map showing the dominant landscape fire type burned during 2000–2019 at 0.25° × 0.25° spatial resolution across the globe according to Global Fire Emissions Database (version 4.1 with small fires, GFED4.1s). The dominant fire type in each grid was determined based on highest proportion of dry matter burned for this grid during the period 2000–2019. Those white grids were grids without any landscape fire burned during the period. **c**, Stacked bar chart showing the relative contributions of different fire types to the fire-emitted $PM_{2.5}$ across the globe and in each continent according to GFED4.1s. Here we only calculated the relative contribution to primary $PM_{2.5}$ emitted from fires, without considering the secondary $PM_{2.5}$ generated from fire emissions.

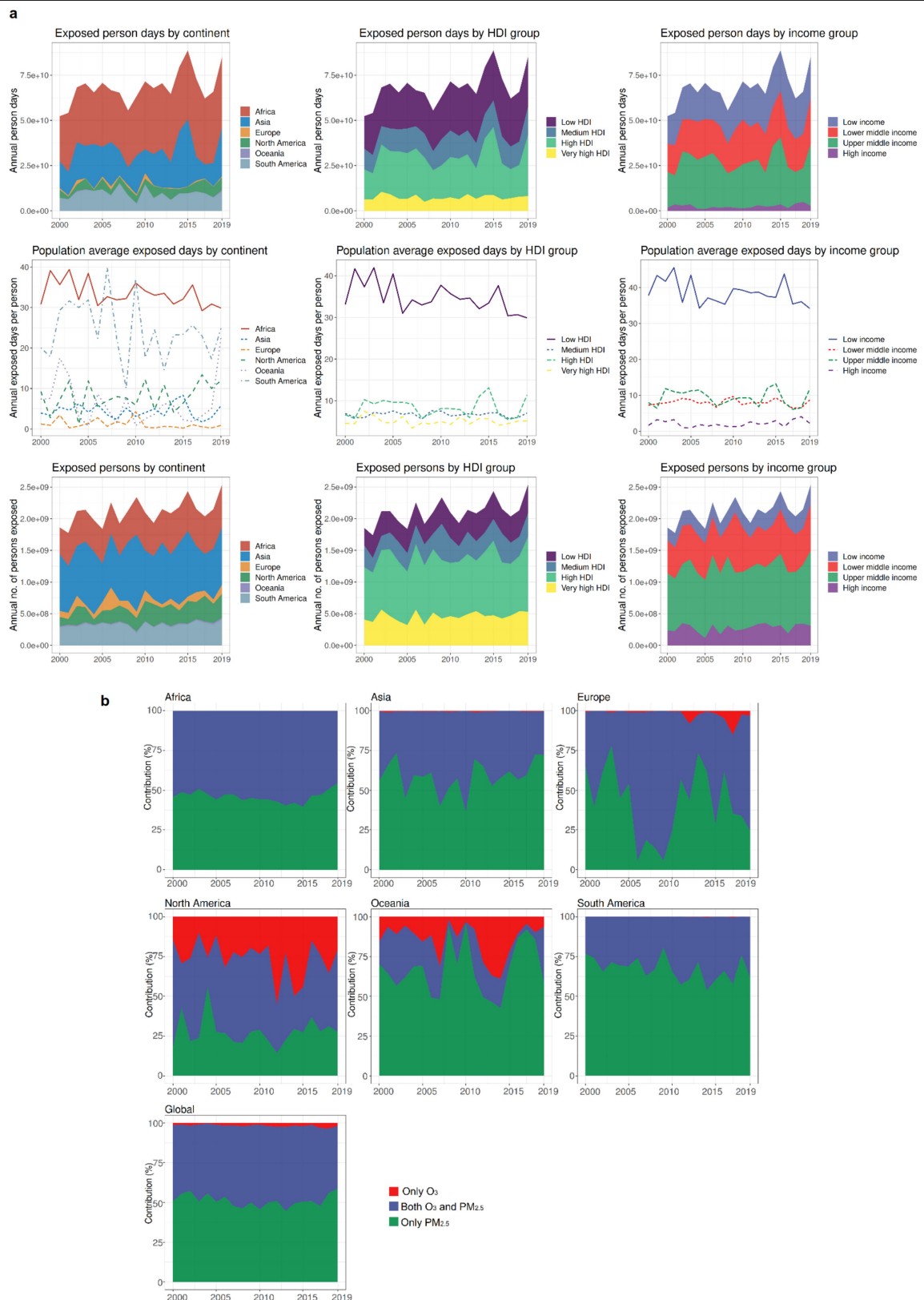

**Extended Data Fig. 2 | Annual exposures for main subgroups and the relative contribution of fire-sourced PM$_{2.5}$ and O$_3$ to the exposure. a**, Stacked area charts and line plots showing the annual population exposure to substantial fire-sourced air pollution (SFAP) from 2000 to 2019 by continent, countries' human development index (HDI) and income level. **b**, Stacked area charts showing the relative contribution of fire-sourced PM$_{2.5}$ and O$_3$ to the person days exposed to substantial fire-sourced pollution (SFAP) from 2000 to 2019 by continent.

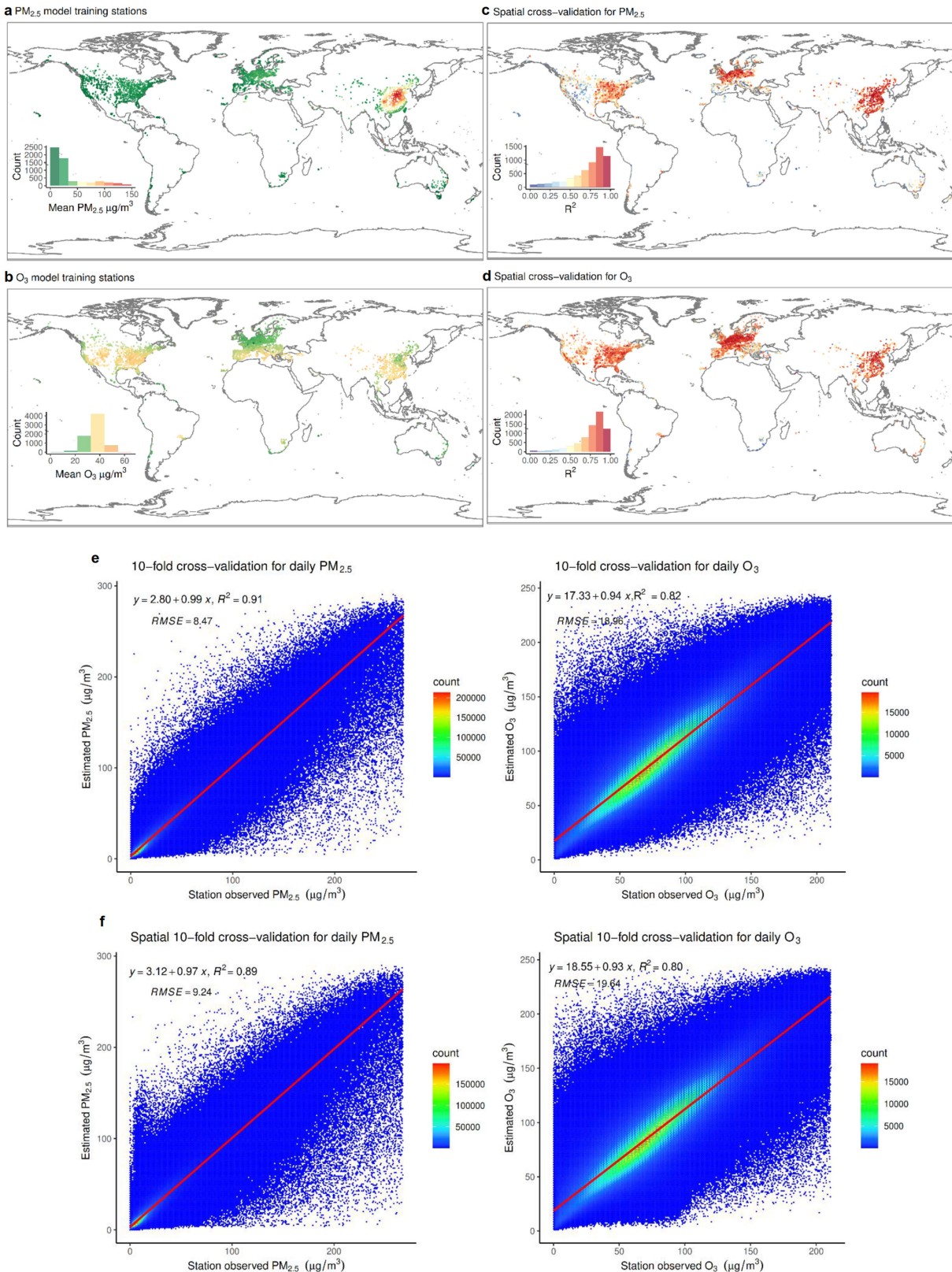

**Extended Data Fig. 3 | Locations of model training stations and model performance of cross-validations. a** and **b**, Maps showing the geographical distribution of air quality monitoring stations of $PM_{2.5}$ (**a**) and $O_3$ (**b**) used for machine learning model training, the mean $PM_{2.5}$ or $O_3$ refers to the average value of all available valid observations for each station. **c** and **d**, Maps showing the station-specific $R^2$ in the spatial 10-fold cross-validation of the random forest models for estimating all-source daily average $PM_{2.5}$ (**c**) and daily maximum 8-hour average $O_3$ (**d**). **e**, Density scatter plots showing the overall performance of the machine learning models for estimating all-source daily average $PM_{2.5}$ (left) and daily maximum 8-hour average $O_3$ (right) based on general 10-fold cross-validation. RMSE, root mean squared error. **f**, Density scatter plots showing the overall performance of the machine learning models for estimating all-source daily average $PM_{2.5}$ (left) and daily maximum 8-hour average $O_3$ (right) based on spatial 10-fold cross-validation.

**a**

| | Continent | No. of stations | No. of observations | Spatial 10-fold CV | | | Cluster-based CV | | | | Raw GEOS-Chem output performance | | |
|---|---|---|---|---|---|---|---|---|---|---|---|---|---|
| | | | | $R^2$ | RMSE | within-$R^2$ | $R^2$ | RMSE | within-$R^2$ | No. of clusters | $R^2$ | RMSE | within-$R^2$ |
| PM$_{2.5}$ | Africa | 208 | 259855 | 0.53 | 12.22 | 0.36 | 0.34 | 15.78 | 0.14 | 11 | 0.09 | 22.68 | 0.09 |
| | Asia | 1823 | 2857198 | 0.87 | 13.68 | 0.85 | 0.56 | 23.34 | 0.42 | 17 | 0.31 | 51.93 | 0.23 |
| | Europe | 1129 | 2446285 | 0.77 | 6.11 | 0.74 | 0.53 | 8.33 | 0.48 | 13 | 0.32 | 14.12 | 0.30 |
| | North America | 2203 | 3456399 | 0.72 | 4.06 | 0.67 | 0.39 | 5.68 | 0.38 | 16 | 0.22 | 9.03 | 0.23 |
| | Oceania | 193 | 384091 | 0.51 | 4.97 | 0.45 | 0.19 | 7.07 | 0.12 | 11 | 0.11 | 13.72 | 0.03 |
| | South America | 105 | 124352 | 0.60 | 25.50 | 0.48 | 0.38 | 36.96 | 0.15 | 11 | 0.0004 | 60.80 | 0.02 |
| | **Global** | **5661** | **9528180** | **0.89** | **9.24** | **0.81** | **0.69** | **14.79** | **0.41** | **75** | **0.48** | **31.00** | **0.22** |
| O$_3$ | Africa | 165 | 315284 | 0.43 | 25.29 | 0.43 | 0.31 | 23.11 | 0.31 | 18 | 0.28 | 46.39 | 0.28 |
| | Asia | 1831 | 2865348 | 0.77 | 23.35 | 0.71 | 0.53 | 26.40 | 0.32 | 19 | 0.31 | 49.04 | 0.16 |
| | Europe | 2393 | 10043651 | 0.79 | 18.71 | 0.74 | 0.70 | 16.37 | 0.58 | 19 | 0.53 | 45.38 | 0.36 |
| | North America | 2247 | 7459422 | 0.82 | 19.20 | 0.76 | 0.68 | 16.15 | 0.57 | 19 | 0.47 | 48.36 | 0.34 |
| | Oceania | 98 | 263885 | 0.70 | 14.19 | 0.70 | 0.49 | 12.80 | 0.51 | 11 | 0.29 | 34.87 | 0.32 |
| | South America | 117 | 150245 | 0.63 | 20.14 | 0.51 | 0.32 | 25.36 | 0.16 | 13 | 0.07 | 38.42 | 0.02 |
| | **Global** | **6851** | **21097835** | **0.80** | **19.64** | **0.74** | **0.67** | **18.14** | **0.52** | **99** | **0.47** | **46.81** | **0.30** |

**b**

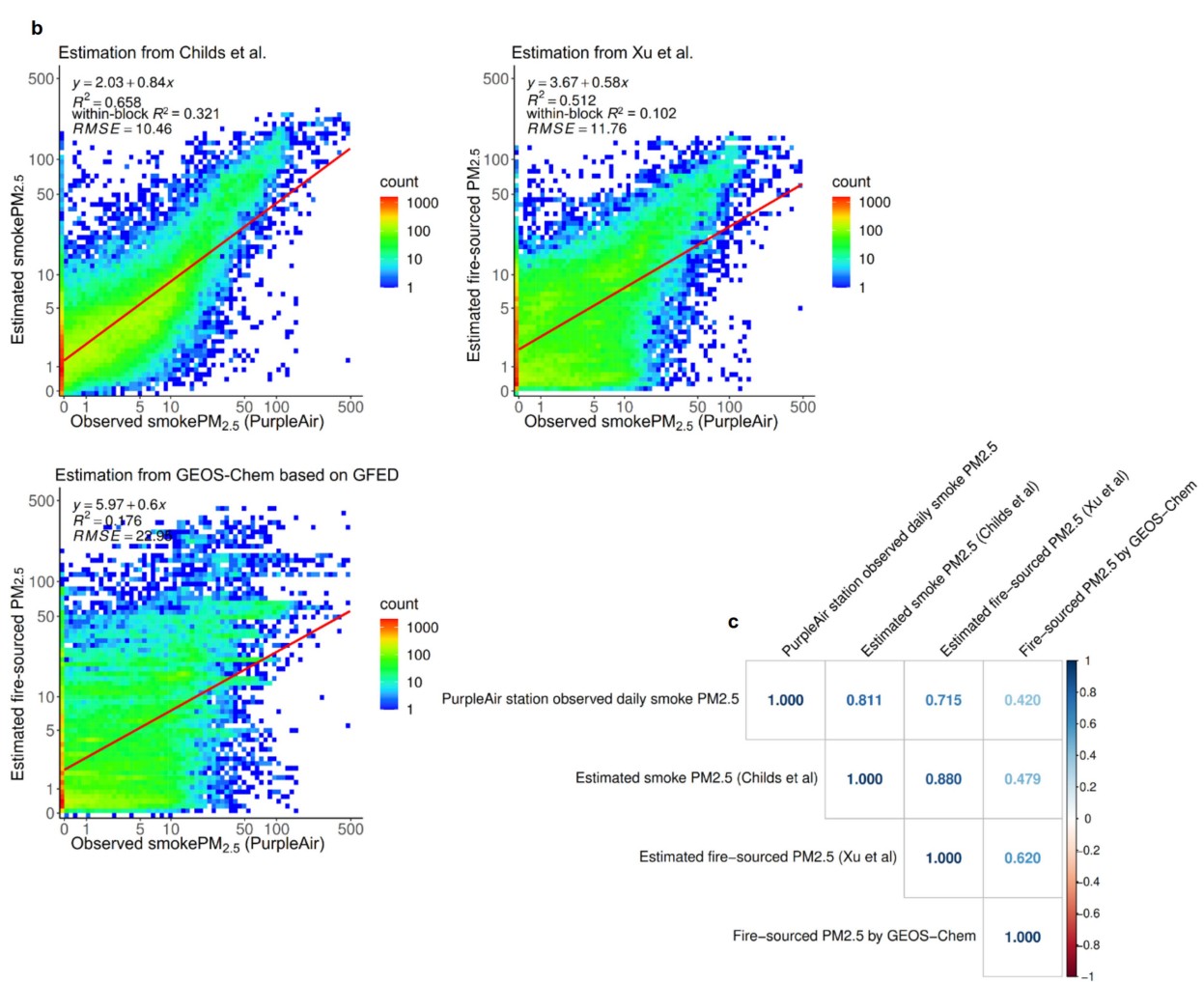

**Extended Data Fig. 4 |** See next page for caption.

**Extended Data Fig. 4 | Overall cross-validation performance for estimating all-source PM$_{2.5}$ and O$_3$ and the accuracy of fire-sourced PM$_{2.5}$ in validation against Childs et al. a**, Table showing the global and continent-specific performance of the machine learning models estimating all-source daily PM$_{2.5}$ and O$_3$ based on spatial 10-fold and cluster-based cross-validation (CV), in comparison with the performance of the raw GEOS-Chem outputs. Some spatial cluster could be in two continents; this is why the global total cluster number is smaller than the sum of continent-specific cluster number. The unit of the root mean squared error (RMSE) is μg/m$^3$ for both PM$_{2.5}$ and O$_3$. The continent-specific CV results was calculated by extracting each continent's CV data from the global spatial 10-fold CV or global cluster-based CV. **b**, Density scatter plots showing the performance of our estimated fire-sourced PM$_{2.5}$, GEOS-Chem simulated fire-sourced PM$_{2.5}$, Childs et al's estimated smoke PM$_{2.5}$ in validation against smoke PM$_{2.5}$ observed by PurpleAir stations. In this analysis, we included 68041 observations during 2016–2019 linked to 2147 PurpleAir stations in the contiguous US based on date, longitude and latitude. The smoke PM$_{2.5}$ data and PurpleAir data were sourced from Childs et al.[17]. The within-block R$^2$ could be interpreted as how much spatial variations of smoke PM$_{2.5}$ within the 2.0° × 2.5° grid box can the model estimates (our estimated fire-sourced PM$_{2.5}$ or Childs et al's estimated smoke PM$_{2.5}$) account for. **c**, Correlation matrix showing the Pearson correlations between our estimated fire-sourced PM$_{2.5}$, GEOS-Chem simulated fire-sourced PM$_{2.5}$, Childs et al's estimated smoke PM$_{2.5}$, and the smoke PM$_{2.5}$ observed by PurpleAir station. The data used for panel c is the same as the data used for panel b.

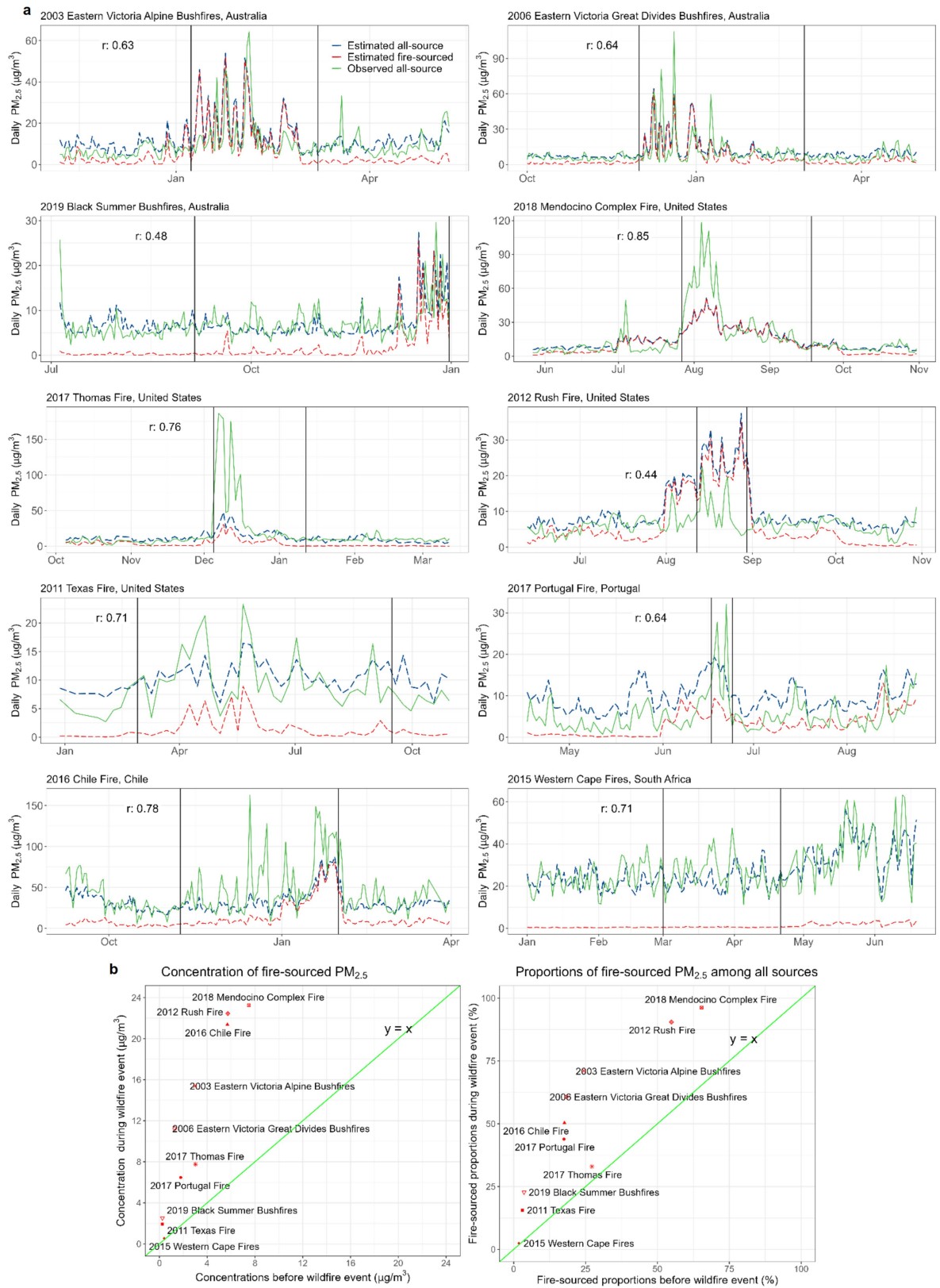

**Extended Data Fig. 5 | Validation of all-source and fire-sourced PM₂.₅ against large wildfire events. a**, Line plots showing the time-series trend of daily all-sourced and fire-sourced PM₂.₅ concentrations before, during, and after ten large wildfire events. The two vertical solid lines represent the start date and the end date of the wildfire event. The end date of the 2019 Black Summer Bushfires was not the real end date of the bushfire event, but the last date that

our model can cover. **b**, Scatter plots comparing the average daily fire-sourced PM₂.₅ concentrations (left) and the proportions of fire-sourced PM₂.₅ among all-source PM₂.₅ (right) during and before the ten selected large wildfire events. The period before each wildfire event was defined as 60 days before the wildfire event start date.

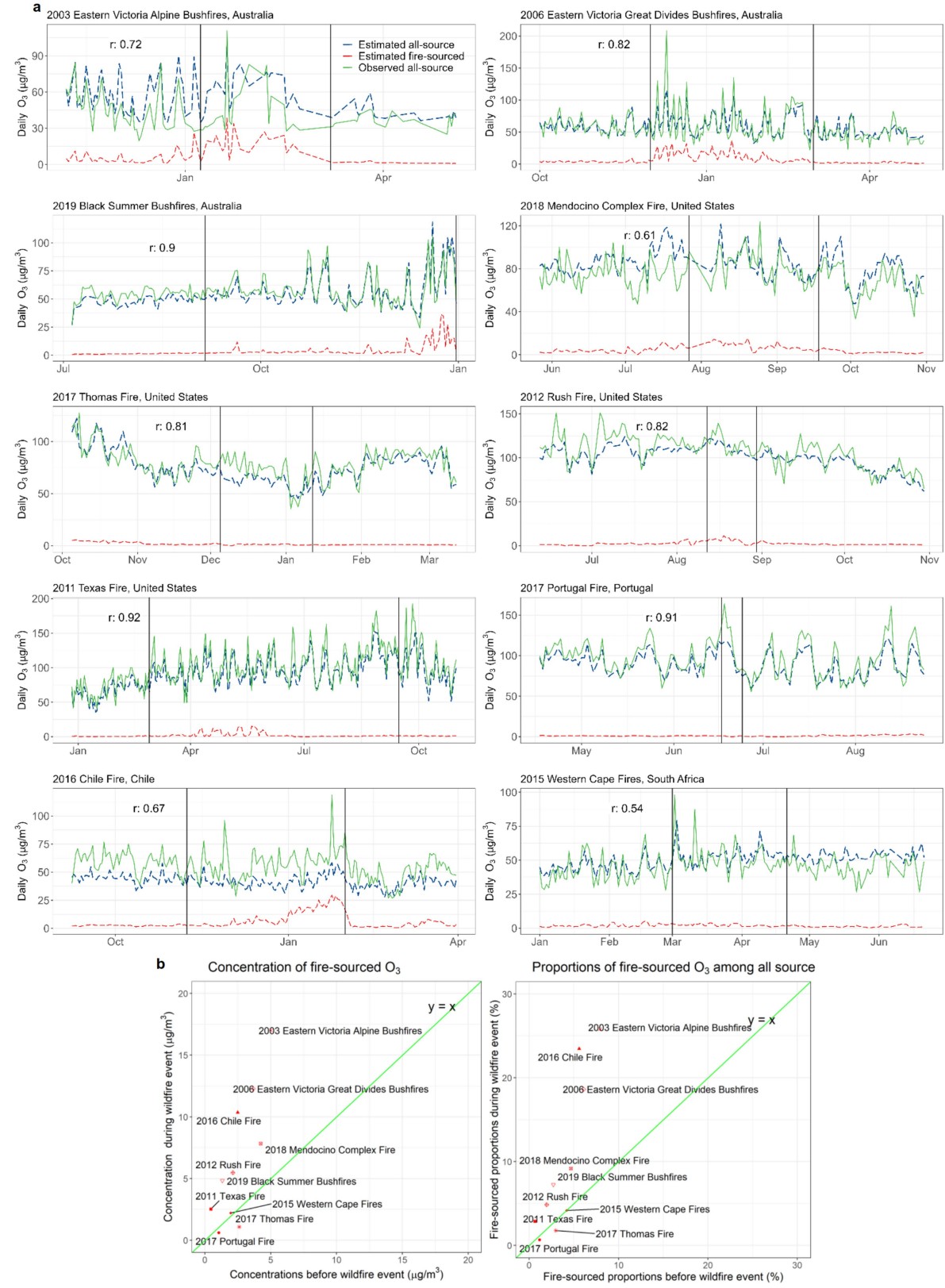

**Extended Data Fig. 6 | Validation of all-source and fire-sourced O₃ against large wildfire events. a**, Line plots showing the time-series trend of daily all-sourced and fire-sourced O₃ concentrations before, during, and after ten large wildfire events. The two vertical solid lines represent the start date and the end date of the wildfire event. The end date of the 2019 Black Summer Bushfires was not the real end date of the bushfire event, but the last date that our model can cover. **b**, Scatter plots comparing the average daily fire-sourced O₃ concentrations (left) and the proportions of fire-sourced O₃ among all-source O₃ (right) during and before the ten selected large wildfire events. The period before each wildfire event was defined as 60 days before the wildfire event start date.

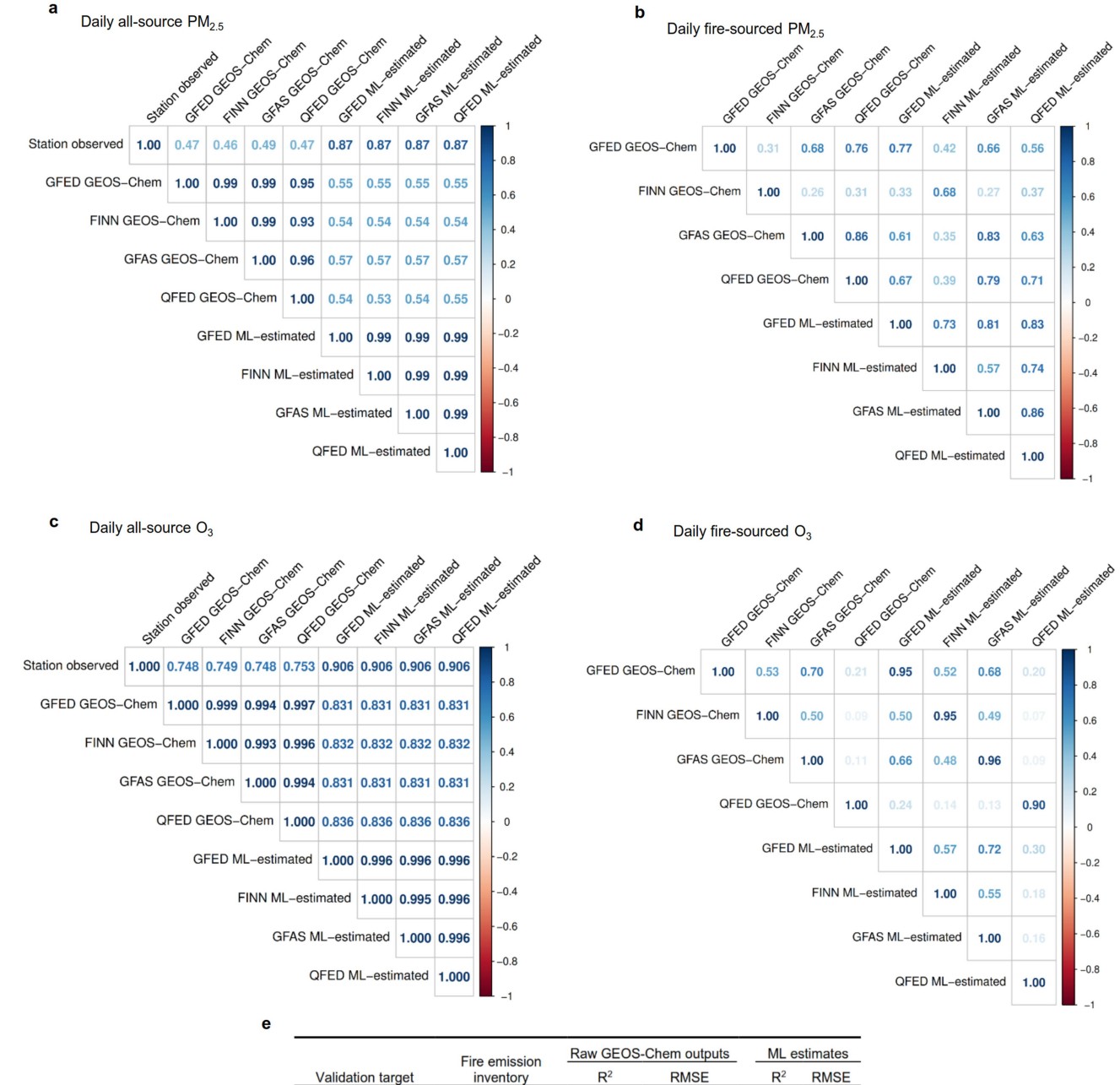

**Extended Data Fig. 7 |** See next page for caption.

**Extended Data Fig. 7 | Comparisons between four fire emission inventories.**
**a**, Correlation matrix showing the Pearson correlations between station observed all-source daily $PM_{2.5}$, and GEOS-Chem simulated and machine learning (ML) estimated all-source $PM_{2.5}$ based on different fire emission inventories in 2012. **b**, Correlation matrix showing the Pearson correlations between GEOS-Chem simulated and ML estimated fire-sourced $PM_{2.5}$ based on different fire emission inventories in 2012. **c**, Correlation matrix showing the Pearson correlations between station observed all-source daily $O_3$, and GEOS-Chem simulated and ML estimated all-source $O_3$ based on different fire emission inventories in 2012. **d**, Correlation matrix showing the Pearson correlations between GEOS-Chem simulated and ML estimated fire-sourced $O_3$ based on different fire emission inventories in 2012. **e**, Table showing the performance of GEOS-Chem outputs and machine learning estimates based on different fire emission inventories in 2012 in validation against station observed daily all-source $PM_{2.5}$, all-source $O_3$ and smoke $PM_{2.5}$. RMSE, root mean squared error, in $\mu g/m^3$. For all the panels in this figure, the ML estimates were based on the ML model trained in out-of-sample stations (according to the 10-fold spatial cross-validation). For each specific fire emission inventory (e.g., FINN), the ML predictors include corresponding GEOS-Chem simulated all-source $PM_{2.5}$ or $O_3$ (e.g., the FINN GEOS-Chem all-source $PM_{2.5}$) and other predictors mentioned in equation 1 (or 2) in the Methods section.

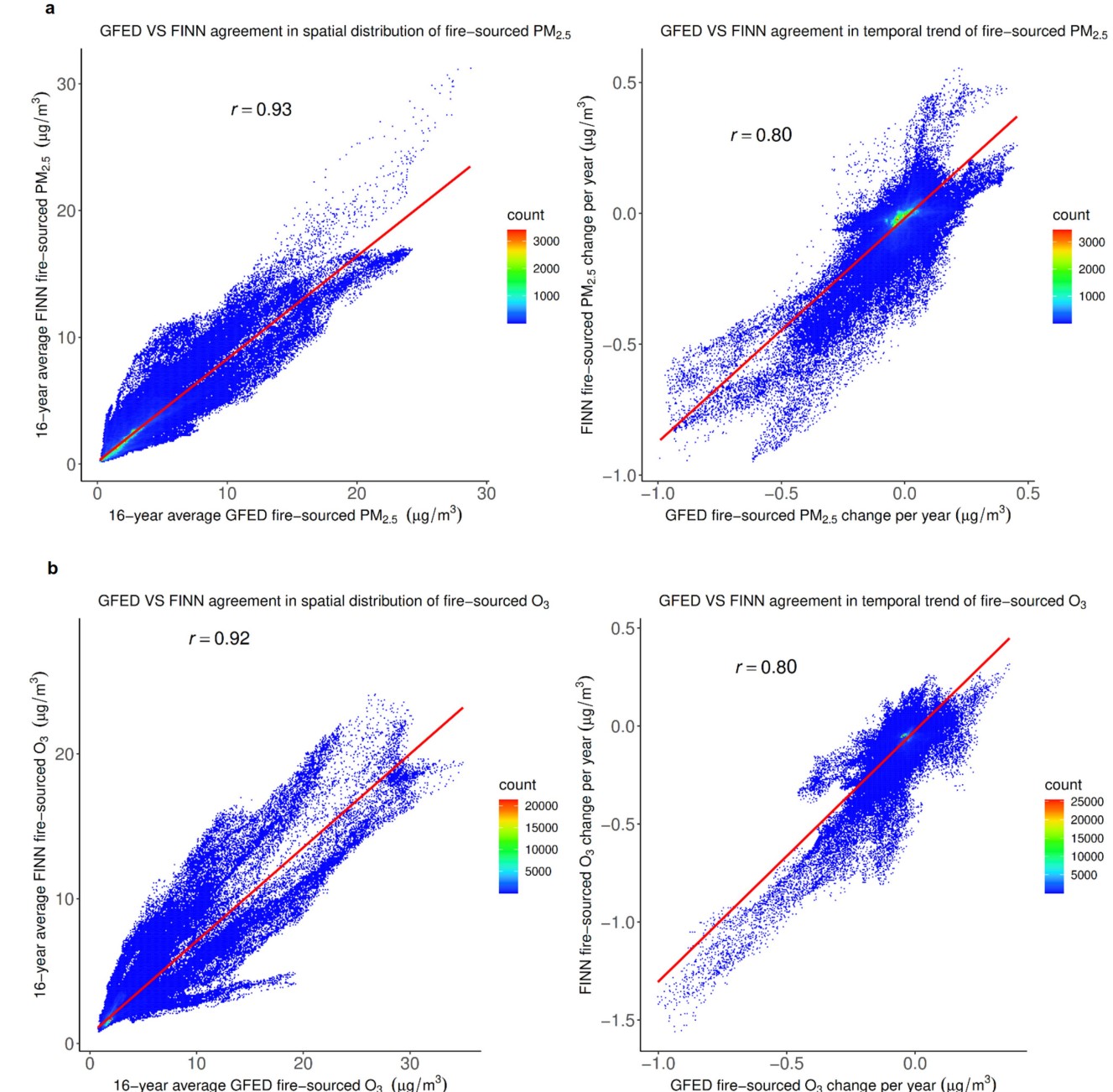

**Extended Data Fig. 8 | Overall agreements between GFED- and FINN-based estimates in global spatial distribution and temporal trends. a**, Density scatter plots showing the agreements between GFED and FINN in the overall spatial distribution (left) and temporal trend (right) of estimated fire-sourced $PM_{2.5}$ during 2002–2017. **b**, Density scatter plots showing the agreements between GFED and FINN in the overall spatial distribution (left) and temporal trend (right) of estimated fire-sourced $O_3$ during 2002–2017. In panel a and b, the daily fire-sourced $PM_{2.5}$ or $O_3$ were estimated using the GEOS-Chem simulations plus machine learning model calibration approach, as detailed in the Methods section. FINN data were only available from 2002 to 2017, so here the 16-year average and the change per year were estimated for this period. The count refers to the count of 0.25° × 0.25° land grids, and all the 394,899 grids across the global land were included in the analyses. For each grid, the temporal trend from 2002 to 2017 was fitted using all annual concentrations during the period with a linear regression.

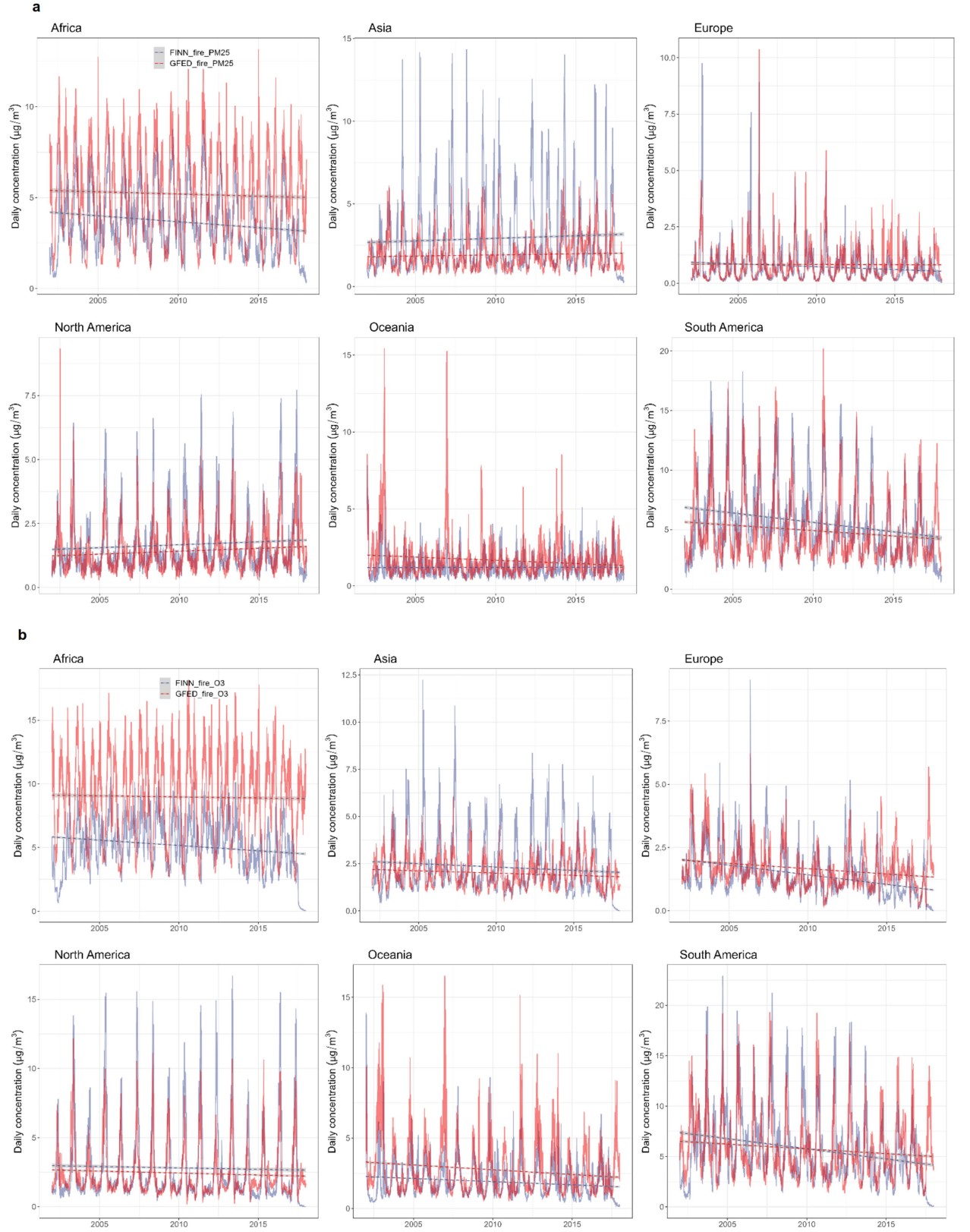

**Extended Data Fig. 9 | Continent-specific agreements between GFED- and FINN-based estimates of daily fire-sourced PM$_{2.5}$ and O$_3$. a**, Line plots showing the agreements between GFED and FINN in the continent-specific daily population-weighted average fire-sourced PM$_{2.5}$ estimates during 2002–2017. **b**, Line plots showing the agreements between GFED and FINN in the continent-specific daily population-weighted average fire-sourced O$_3$ estimates during 2002–2017. In both panels, the fire-sourced PM$_{2.5}$ or O$_3$ were estimated using the GEOS-Chem simulations plus machine learning model calibration approach, as detailed in the Methods section. The dashed lines refer to trend lines fitted using linear regressions.

**Extended Data Table 1 | Tables of decadal average exposures for the globe and main subgroups**

**a**

| | Fire-sourced PM$_{2.5}$ (% of all-source PM$_{2.5}$) | | Trend of fire-sourced PM$_{2.5}$ | | Fire-sourced O$_3$ (% of all-source O$_3$) | | Trend of fire-sourced O$_3$ | |
|---|---|---|---|---|---|---|---|---|
| | 2000-09 | 2010-19 | Change per decade | P for trend | 2000-09 | 2010-19 | Change per decade | P for trend |
| **Continent** | | | | | | | | |
| Africa | 5.3 (17.7%) | 5.1 (16.5%) | -0.27 | 0.020 | 9.1 (10.4%) | 8.8 (10.3%) | -0.45 | 0.024 |
| Asia | 1.9 (3.9%) | 2.0 (3.8%) | 0.15 | 0.163 | 2.0 (2.1%) | 1.9 (2.1%) | -0.01 | 0.901 |
| Europe | 0.8 (3.6%) | 0.8 (4.1%) | -0.01 | 0.881 | 1.7 (2.2%) | 1.6 (1.9%) | -0.12 | 0.376 |
| North America | 1.3 (9.8%) | 1.6 (13.6%) | 0.27 | 0.001 | 2.5 (2.8%) | 2.4 (2.8%) | 0.08 | 0.613 |
| Oceania | 1.9 (20.5%) | 1.7 (20.2%) | -0.02 | 0.947 | 3.1 (5.2%) | 2.8 (4.7%) | -0.20 | 0.527 |
| South America | 5.2 (20.5%) | 4.5 (19.4%) | -0.61 | 0.012 | 6.1 (9.3%) | 5.4 (8.4%) | -0.60 | 0.050 |
| **HDI group** | | | | | | | | |
| Low HDI | 5.6 (17.4%) | 5.4 (16.1%) | -0.28 | 0.026 | 9.9 (11.5%) | 9.5 (11.3%) | -0.56 | 0.013 |
| Medium HDI | 2.0 (4.0%) | 2.1 (3.6%) | 0.08 | 0.420 | 2.5 (2.5%) | 2.4 (2.4%) | -0.04 | 0.587 |
| High HDI | 2.3 (5.8%) | 2.4 (6.1%) | 0.19 | 0.138 | 3.0 (3.4%) | 2.9 (3.4%) | -0.04 | 0.787 |
| Very high HDI | 1.6 (7.1%) | 1.6 (7.4%) | -0.07 | 0.156 | 1.8 (2.2%) | 1.7 (2.0%) | -0.08 | 0.479 |
| **Income group** | | | | | | | | |
| Low income | 5.7 (17.2%) | 5.6 (16.5%) | -0.12 | 0.291 | 8.8 (10.0%) | 8.8 (10.3%) | -0.16 | 0.429 |
| Lower middle income | 2.3 (5.0%) | 2.3 (4.5%) | 0.10 | 0.197 | 3.2 (3.3%) | 3.1 (3.2%) | -0.07 | 0.335 |
| Upper middle income | 2.4 (6.1%) | 2.5 (6.2%) | 0.11 | 0.382 | 3.0 (3.5%) | 2.9 (3.5%) | -0.04 | 0.772 |
| High income | 1.1 (5.5%) | 1.1 (6.0%) | -0.03 | 0.426 | 1.6 (1.9%) | 1.5 (1.8%) | -0.03 | 0.782 |
| **Global** | 2.4 (6.2%) | 2.5 (6.1%) | 0.11 | 0.072 | 3.2 (3.6%) | 3.2 (3.6%) | 0.02 | 0.840 |

**b**

| | Annual person days exposed to SFAP (×10$^8$) | | | | Annual days exposed to SFAP per person | | | | Annual persons exposed to SFAP (×10$^6$) | | | |
|---|---|---|---|---|---|---|---|---|---|---|---|---|
| | 2000-09 | 2010-19 | Change per decade | P for trend | 2000-09 | 2010-19 | Change per decade | P for trend | 2000-09 | 2010-19 | Change per decade | P for trend |
| **Continent** | | | | | | | | | | | | |
| Africa | 310.8 | 379.7 | 60.3 | <0.001 | 34.2 | 32.5 | -2.5 | 0.029 | 512.4 | 596.4 | 83.5 | <0.001 |
| Asia | 178.4 | 195.5 | 15.9 | 0.593 | 4.5 | 4.4 | -0.1 | 0.895 | 873.7 | 803.1 | -19.2 | 0.777 |
| Europe | 9.8 | 6.3 | -4.1 | 0.169 | 1.4 | 0.9 | -0.6 | 0.158 | 128.0 | 97.6 | -15.4 | 0.576 |
| North America | 36.9 | 50.1 | 14.9 | 0.042 | 7.2 | 8.9 | 1.9 | 0.153 | 197.2 | 319.2 | 109.1 | 0.001 |
| Oceania | 2.1 | 1.9 | 0.2 | 0.828 | 7.2 | 5.4 | -0.8 | 0.739 | 17.8 | 17.8 | 1.8 | 0.456 |
| South America | 94.2 | 94.7 | -0.9 | 0.938 | 25.5 | 23.1 | -2.7 | 0.359 | 308.2 | 342.5 | 30.4 | 0.096 |
| **HDI group** | | | | | | | | | | | | |
| Low HDI | 221.2 | 273.5 | 44.8 | <0.001 | 36.0 | 33.7 | -3.4 | 0.012 | 349.5 | 404.4 | 55.8 | <0.001 |
| Medium HDI | 120.5 | 138.8 | 18.1 | 0.001 | 6.8 | 6.7 | -0.1 | 0.811 | 311.2 | 341.9 | 49.1 | 0.127 |
| High HDI | 217.0 | 239.4 | 22.6 | 0.365 | 8.3 | 8.4 | 0.1 | 0.907 | 929.4 | 942.2 | 46.4 | 0.307 |
| Very high HDI | 72.9 | 76.4 | 1.1 | 0.843 | 5.1 | 5.0 | -0.3 | 0.518 | 432.7 | 482.2 | 45.9 | 0.107 |
| **Income group** | | | | | | | | | | | | |
| Low income | 178.0 | 226.8 | 42.4 | <0.001 | 39.1 | 38.1 | -2.2 | 0.089 | 230.2 | 277.3 | 47.9 | <0.001 |
| Lower middle income | 187.7 | 217.0 | 26.9 | 0.010 | 8.0 | 8.0 | -0.2 | 0.657 | 593.6 | 620.6 | 60.1 | 0.165 |
| Upper middle income | 245.0 | 256.5 | 11.0 | 0.647 | 9.5 | 9.2 | -0.3 | 0.707 | 963.7 | 976.8 | 42.2 | 0.275 |
| High income | 21.4 | 28.0 | 6.1 | 0.150 | 1.9 | 2.3 | 0.4 | 0.305 | 249.5 | 301.2 | 39.7 | 0.135 |
| **Global** | 632.1 | 728.2 | 86.3 | 0.010 | 9.7 | 9.9 | 0.1 | 0.856 | 2037.2 | 2176.6 | 190.1 | 0.007 |

**a**, Table showing the global annual population-weighted average landscape fire-sourced PM$_{2.5}$ and O$_3$ during 2000–2019, by continent, human development index (HDI) group and income group. The units of PM$_{2.5}$ and O$_3$ concentrations are both µg/m³. **b**, Table showing the population exposure to substantial fire-sourced air pollution (SFAP) during 2000–2019, by continent, HDI group and income group.