## [Peer Review File · Nature]

Manuscript Title: Global population exposure to landscape fire air pollution from 2000-2019

Reviewer Comments & Author Rebuttals

Reviewer Reports on the Initial Version:

Referees' comments:

Referee #1 (Remarks to the Author):

This is a nice paper on an important topic. Wildfires have grown more frequent in much of the world and we lack a complete picture of how changing wildfire risk is affecting air pollution. Relative to similar work that uses transport models to estimate global wildfire PM, this paper usefully combines a transport model with a ML calibration to make sure that predictions match total PM at stations where measurements are available. The model leans pretty heavily on a specific transport model and a specific emissions inventory to partition total PM into wildfire- and non-wildfire PM, and I have some questions about how they validate the wildfire estimates. My comments are mainly on the paper's methodology, and I believe that getting these methods points nailed down is critical for publishing in a very high visibility place like Nature. I enjoyed the opportunity to read this very nice work.

1. It's not clear whether the wildfire validation is done using held out stations in the spatial CV or done using the model that has been fit to the stations that they're evaluating. I think it must be the latter since the performance is very high, meaning this validation is just showing they can accurately predict stations that they trained on. I think the appropriate validation here would be to fully hold out any wildfire stations they want to evaluate on when training the ML models (equations 1 and 2), and then show the model can re-predict those stations. Otherwise I think the performance in these tasks could be substantially overstated. Figs S8-31 should be redone using stations held out from ML model training.

Another option, which I propose they also do, is to validate against other recent estimates of wildfire PM which have been shown to have high performance, e.g. Childs et al ES&T 2022 estimates in the US. That paper has either gridded predictions of wildfire PM or, better yet, ground-based station estimates of wildfire PM that are available in their repo and could be used as comprehensive validation data. The most impressive test here would be to not use any US stations at all in model training — thus mimicking what the authors have to do in places like Africa where there are essentially no stations — and then seeing how well the model can prediction stations in the US and predict wildfire events. For a journal like Nature, I think this is not too much to ask — this is the only way to demonstrate that the model both predicts wildfire smoke accurately and is able to do so in a location where the model was not trained.

2. The other reason validation statistics can be inflated in the daily estimates is if a lot of variation in PM levels comes from seasonal differences or spatial differences in average PM, and the model is accurately reproducing these seasonal averages/ spatial differences but not getting the daily variation correct. The way to check this is, in the spatial CV, calculate the r^2 separately for each station but include month-of-year intercepts to take out the seasonal pattern in the analysis of each station. The r^2 is then computed on the residuals once month intercepts have been taken out (often called the "within" r^2 in statistics packages, e.g. the `fixest` package in R), and the authors can then report the distribution of r^2 across stations to give the reader a sense of the distribution of performance.

3. A third and related validation point again speaks to the fact that there are very few monitors in

SSA, South America, South East Asia, where wildfire PM2.5 accounts for the majority of the worldwide smoke exposure by their estimation. While the authors of course cannot do anything about the low station count, they could try additional experiments to see whether models trained on one continent predict well in another continent with few stations. So could a model trained in North America make accurate predictions in Europe without training on Europe? Again the worry is that the model is being pushed hard to make predictions in locations where it has seen very little data and might be doing a bad job without us knowing it, and so we should do everything we can to evaluate whether the model can make accurate predictions over very large areas without having been trained there. The only way to do this is to hold out a region with a bunch of stations, retrain the model without those stations, and then evaluate.

4. The partitioning between wildfire and other sources of PM relies on GEOS-Chem which in turn relies heavily on the emissions inventories the model is fed. GFED is of course a widely-used source, but existing work in the US shows that feeding different emissions inventories — all of which are widely in use — into the same transport model gives very different concentration estimates — often up to 20x different in a given region. See Koplitz, Nolte et al 2018 Atmospheric Environment. This suggests that we might be pretty worried about estimates just using one inventory as input. I see a few options here. The first is if the correct out-of-sample performance metrics performed above - if these still look good then the authors maybe have a strong argument that this Koplitz-type result is not a problem for them (and/or that GFED is the best inventory). The second and more painful is to also run this with a different inventory and see whether results change. A third is to try to make the argument that GFED is fine — but again I don't think the existing lit is their friend in that regard, but maybe there are more recent papers that support the use of GFED relative to other options.

5. For an outlet like Nature, I would encourage the authors to think hard about the few visuals that most compellingly communicate the message they want the reader to take home from the paper, and then pare down their large set of main text visuals to just those. Right now the visuals are tiny, hard to read, overwhelming in number, and not always necessary to support the narrative. Fig 1 is good, Figs 2-5 have a lot of repetitive information, not all of which is useful in the narrative, and Fig 6 looks like it was copy-pasted from the Lancet. I think the goal of the figures here is not to just spit out every analysis and output that was done, but to focus on the key part of the story - I would encourage them to do that.

Referee #2 (Remarks to the Author):

This manuscript looks at human exposure to wildfire smoke, trends over 2000-2019, and disparities between countries, income, etc. While this is an interesting study, I have significant concerns with aspects of the study, particularly the use of a single emissions inventory (when different inventories have shown different global trends during this time period). Hence, I cannot recommend publishing this paper without major revisions.

Major issues

Significant unstated and understated issues in the estimates: There are a number of issues with the estimates that are either not discussed or discussed very little.

(1) The ML is (necessarily) tuning the total PM2.5 and O3 and not the wildfire specific PM2.5 and O3. Hence, it fully relies on GEOS-Chem for the smoke vs. non-smoke split, so any issues with this split (e.g., from #2 and #3 below) are not corrected for, nor is it clear how well the ML would correct for those issues.

(2) GEOS-Chem w/ GFED does not include any fire-specific plume rise scheme. Depending on the

model setup, the emissions are either entirely into the boundary layer or a globally fixed split between the boundary layer and lower/mid free troposphere. This leads to known model biases in regions downwind of fires that mostly receive lofted smoke rather than smoke at the surface (e.g., the US Northern Great Plains in <https://doi.org/10.1021/acs.est.8b05430>). Even if the ML somewhat picks up on the biases when smoke is present, the smoke vs. non-smoke split will be wrong.

(3) This is my biggest issue with the manuscript. The trends in wildfire-specific PM_{2.5} are going to be driven almost entirely by the choice of emission inventory. The different emission inventories have quite different trends, and this is highlighted well in Carter et al, 2020: <https://doi.org/10.5194/acp-20-2073-2020>. Please see Figure 5 of this paper. During the 2004-2016 window, two inventories (FINN and QFED) appear to have decreasing emissions trends globally while GFED does not (GFAS perhaps has a small decreasing trend or no trend). Hence, I would expect the results in the current manuscript, especially the trends, to depend greatly on the choice of inventory. Hence, it seems like some other inventories might not have the same increase in exposure per person as GFED. Further, GFED is not clearly the best inventory in the Carter paper (though only evaluated over North America). The limitation of inventory choice is mentioned just very briefly in the 2nd to last paragraph, but this issue seems too big for publishing without testing the sensitivity to the inventories.

(4) The ML inputs are a dump of variables that do not all have a physical connection to affecting PM_{2.5} or O₃, yet some key variables that relate to how wildfires affect these variables are left out. For example, the altitude of the 0.25x0.25 deg box relative to the mean of the 2x2.5 deg box might relate to smoke pooling in low-lying areas overnight, and the distance from nearest fire and direction of the fire relative to the wind direction might help catch for smoke spikes. It seems like a lot of variables were thrown at the ML in an attempt to fit things against all monitors globally, but not a lot of thought was given to the factors that might help get the smoke impacts right.

Figures: The figures are poor quality, both in readability (small unreadable fonts, pale yellow very hard to see) and context (e.g., 9 panels figures where 7 of the panels are 20 year time series is just impossible to digest).

Specific comments

Title and throughout: Not all of the fires in GFED are wildfires. Should we be calling this exposure to wildfire smoke when it includes other types of fires? Seems misleading.

L37: I was confused by why O₃ was being presented in $\mu\text{g m}^{-3}$ in the abstract before I noticed an explanation in the methods. I bet this will confuse people who are used to thinking of O₃ in ppb. Not sure what's best.

L42-43: The comparison to WHO guidelines here also confused me. It gets explained in the paper as being total PM/O₃ > guideline + fraction smoke over a threshold, but here I wasn't sure what it meant. Was smoke PM/O₃ alone higher than the guideline? Was the total simply over the guideline. We don't actually learn this until L153

L70-72: I was confused as to why Brazil was getting specific attention here when smoke health impact assessments have been done on different countries as well but not mentioned. Then I realized it was just the authors citing themselves but not others. Not the best look.

L79: "is" should be "are"

L123-125: What are the ranges here? It seems like interannual variability? To me, the spatial variability within the countries/regions is more important. It definitely looks like there are parts of

North America where the contribution is <9.2%. Why would we want to spatially average a whole continent when some people are exposed to much more smoke than others while giving temporal variability? It would seem like the variability in people's mean exposure is more important to show than the interannual variability.

L141-143: Again, what is the variability here? It seems like temporal variability, but the variabilities between countries/populations seem more important.

L179: What does "population-averaged days" mean here? The average number of days of exposure per person? Please state it more explicitly.

Figure 6 and associated discussion: There is huge interannual variability in fires in any given region. Comparing 2000 to 2010 to 2019 is insufficient for looking at how exposure in different countries is changing since any given year could be anomalously better or worse than the surrounding years. There is little that we should be taking from the changes between the 3 compared years. If you wanted to stick with a similar analysis, I would compare the first 10 years averaged to the final 10 years averaged to look at how exposure might be changing across countries while removing some of the interannual variability.

Figure 1: Please add some sign of significance of the trends in the maps in panels E and F.

Figure 5: Is this for PM2.5 or O3?

L334-335: And forests have a lot more fuel that can burn than savannas/grasslands.

L392: "several catastrophic wildfire events". The recent bad years in North America generally have had 10s of very large wildfires in each year, so "several" does not seem like the right word.

L437: Smoke can travel 1000s of km and still have significant AQ impacts. E.g., <https://go.nasa.gov/3V83obX>

L454: "we have much higher accuracy" Did you quantify this? You can actually evaluate wildfire specific PM2.5 since monitors do not measure it, so how do you know?

L466: "increases in CO are generally restricted to the immediate wildfire areas". CO has a longer atmospheric lifetime than PM2.5 and O3, and it will move with the plume the exact same way, so this statement is very strange. Do you mean, "the region where CO is at dangerous levels is generally restricted to the immediate wildfire areas"?

L585-590: How good is GEOS-Chem alone before the ML?

L605: I don't think you can actually evaluate wildfire-specific PM2.5 and O3 against obs since the obs measure the total. These correlation coefficients seem misleading/incorrect.

L605: Also, in these comparisons, how was the ML trained and evaluated? When the ML was trained, were some monitors left out by other nearby monitors were included with the model being trained on the smoke time period? If yes, or something similar, it seems like this evaluation would do nothing to show the confidence in how well the ML does in locations away from monitors.

Referee #3 (Remarks to the Author):

This is an interesting study in which the authors use a combination of chemical transport model simulations and machine learning to estimate global daily exposure to PM2.5 and ozone from wildfires. The authors present some relevant and novel metrics for this exposure, shown in ways

that haven't been done previously to characterize the exposure. They find that wildfire smoke is a significant contributor to global air pollution exposure (4-7%), that wildfire smoke is much greater in certain low-income regions, and that exposure to PM_{2.5} is increasing through time. These findings are presented comprehensively in a way that I expect many researchers will be interested in.

But there are also weaknesses that work against the novelty of the paper. The paper is presented as an extension of the methods they applied in the authors' previous papers (l. 86). That is, the combination of a CTM with machine learning is not new here but was presented in their previous papers. Here the novelty is the global application, which makes the results comprehensive and interesting. There have also been previous global-scale wildfire smoke exposure and health impact studies, as the authors discuss in the conclusions, where it seems the advantage of this paper is in the machine learning to bias correct.

The finding that wildfire smoke is a few percent of total global exposure is not surprising. The details provided on the location of greater exposure and trends are worthwhile and interesting. Still the paper leaves questions about what to do with the information. There is rather little discussion of health consequences. Unlike anthropogenic or industrial emissions, wildfire smoke is not easily controlled. In places like the Western US, past fire suppression has been counter-productive. Perhaps only in low-income countries where wildfire is related with forest clearing does the policy-relevance become more apparent. This doesn't get much discussion.

Overall, I think the paper is interesting, is likely to be technically sound, and will likely be widely cited by the global air pollution public health community. But to me the factors that reduce its novelty that I mentioned above make it questionable whether it should appear in a high-impact journal such as *Nature*, and it may be more appropriate for a *Nature* specialty journal or another journal of similar stature (*Lancet Planetary Health*).

I say that it is likely to be technically sound because I am not an expert in machine learning and the machine learning methods are a bit of a black box. I think that a machine learning expert would need to read previous papers and the supplement carefully to understand what was done, and even with that, there may be possibilities of spurious (non-physical) relationships estimated by the machine learning. Testing the machine learning results with a small number of fires (6) in the USA and Australia seems to be a small sample.

The paper is well-written, but seems long to me for a *Nature* paper.

More specific comments:

If I understand their methods correctly, the only input data to the machine learning about where and when fires occur, and their magnitude, comes from the fire emissions database (GFED) which is input to GEOS-chem. That is state-of-the-art, but it should be mentioned that fire emissions are difficult to estimate from space (and in general), GFED may have significant errors in estimated emissions for individual fires, as well as possibly biases more broadly. Where there are no measurements to bias-correct (low-income nations), does then the model essentially use the GEOS-chem modeled estimates? Or use some machine learning corrections based on relationships found where monitors exist (US and Europe)? GEOS-chem also runs at a coarse resolution that is not great for estimating population exposure. Perhaps more discussion of these limitations is warranted.

The authors should also define "wildfire". I understand that wildfires would not include domestic wood burning. But would it include controlled burns for agriculture? Anthropogenic fires for land clearing, which are set by people and therefore not quite "wildfire"? I think these fires would be in the GFED emissions, and so I think they're included here.

l. 65-77 – The authors leave out discussion of other studies that have estimated global wildfire

smoke exposure, which they later mention in the discussion.

Fig. 3 – I think this would be better as a plot with 4 lines, rather than bars.

l. 330 – I'm not sure the discussion of CO₂ emissions is very relevant here.

l. 340-354 – Good points to mention, but I don't think this long discussion is necessary.

l. 401 – The language here about the climate change influence differs from that presented earlier, where "we postulate"... and "more studies are warranted" (l. 356).

l. 447-451 – Here references 28-34 are mentioned for the first time. The introduction emphasized the novelty of this study by comparing it with studies of fire exposure. But in fact there were several studies of wildfire smoke exposure that the authors didn't mention until now.

l. 465-467 – CO has a longer atmospheric lifetime than PM_{2.5}, O₃, or their precursors, so I'm not sure that this statement that CO is restricted to immediate fire areas is correct.

l. 479 – "world's first". It seems that this claim is not supported, given references 33 and 34.

l. 487 – the authors appear to have made a significant effort to gather air quality data. In the case of ozone, they could have used the TOAR database which did this work to gather data from many countries. For PM_{2.5}, the Global Burden of Disease group gathered data and I think (but am not sure) that they are making their database publicly accessible. Using data from these other efforts might include some measurements from some low-income nations that the authors may have missed. For ozone, TOAR also does a quality control step that is missing from databases like OpenAQ.

l. 591 – The method of estimating wildfire contribution as a modeled fraction seems simple (though I don't have a better alternative at hand) and may be prone to errors. For example, let's say the model has the smoke plume going in the wrong direction. Within that modeled smoke plume, the machine learning might adjust the PM_{2.5} down based on a low PM_{2.5} measurement, but it would still say the fraction from wildfire smoke would be high. Other errors like this might result, and in many cases there won't be a measurement to correct the error. I don't know a better method, but is this worthy of discussion?

Author Rebuttals to Initial Comments:

Referees' comments:

Referee #1 (Remarks to the Author):

This is a nice paper on an important topic. Wildfires have grown more frequent in much of the world and we lack a complete picture of how changing wildfire risk is affecting air pollution. Relative to similar work that uses transport models to estimate global wildfire PM, this paper usefully combines a transport model with a ML calibration to make sure that predictions match total PM at stations where measurements are available. The model leans pretty heavily on a specific transport model and a specific emissions inventory to partition total PM into wildfire- and non-wildfire PM, and I have some questions about how they validate the wildfire estimates. My comments are mainly on the paper's methodology, and I believe that getting these methods points nailed down is critical for publishing in a very high visibility place like Nature. I enjoyed the opportunity to read this very nice work.

[Response] We really appreciate all your encouraging comments and constructive comments. We have responded to these comments in detail below. Please note that the page and line numbers mentioned below refer to the marked-up manuscript with track changes.

1. It's not clear whether the wildfire validation is done using held out stations in the spatial CV or done using the model that has been fit to the stations that they're evaluating. I think it must be the latter since the performance is very high, meaning this validation is just showing they can accurately predict stations that they trained on. I think the appropriate validation here would be to fully hold out any wildfire stations they want to evaluate on when training the ML models (equations 1 and 2), and then show the model can re-predict those stations. Otherwise I think the performance in these tasks could be substantially overstated. Figs S8-31 should be redone using stations held out from ML model training.

[Response] Thanks for pointing out this issue which we did not realize before. Yes, we indeed used stations in the training set. In this revision, we have redone the original Figs S8-31 using stations held out from ML model training. Specifically, for each wildfire event, we selected the five nearest stations. Among the five stations, we chose the station that experienced the most significant increase in observed $PM_{2.5}$ (or O_3) during the wildfire event compared with the pre-wildfire period (i.e., the most affected station). Then we trained an ML model without the data of all the five nearest stations (i.e., all five nearby stations left out), the ML model was then validated in the most affected station. Please see Table S4 and Figs S12-S15, pages 18-23 of the Supplementary materials.

2. Another option, which I propose they also do, is to validate against other recent estimates of wildfire PM which have been shown to have high performance, e.g. Childs et al ES&T 2022 estimates in the US. That paper has either gridded predictions of wildfire PM or, better yet, ground-based station estimates of wildfire PM that are available in their repo and could be used as comprehensive validation data. The most impressive test here would be to not use any US stations at all in model training — thus mimicking what the authors have to do in places like Africa where there are essentially no stations — and then seeing how well the model can prediction stations in the US and predict wildfire events. For a journal like Nature, I think this is not too much to ask — this is the only way to demonstrate that the model both predicts wildfire smoke accurately and is able to do so in a location where the model was not trained.

[Response] As suggested, we have compared our estimated fire-sourced $PM_{2.5}$ with the smoke $PM_{2.5}$ estimated by Childs et al, and the PurpleAir station observed smoke $PM_{2.5}$ (also provided by Childs et al). Because most of the training stations used by Childs et al had also been our training stations, directly validating our estimated fire-sourced $PM_{2.5}$ against the observed smoke $PM_{2.5}$ by Childs et al's training stations may have overfitting issues (i.e., may overestimate our accuracy). To avoid this problem, we chose the station observed smoke $PM_{2.5}$ of PurpleAir stations (a kind of low-cost sensor; have been calibrated against reference grade stations by Childs et al before calculating smoke $PM_{2.5}$) as our validation target. We have added the validation results in the supplementary material and Methods section (lines 931-936, page 33 of the **marked-up manuscript):**

“The results showed that our estimated fire-sourced PM_{2.5} were highly correlated with the estimated smoke PM_{2.5} by Childs et al (Pearson correlation coefficient r : 0.88) (Fig S10). When validated against the PurpleAir station observed smoke PM_{2.5}, our estimates’ accuracy ($R^2 = 0.51$, RMSE = 11.76 $\mu\text{g}/\text{m}^3$) was close to the accuracy of Childs et al’s estimates ($R^2 = 0.66$, RMSE = 10.46 $\mu\text{g}/\text{m}^3$), both much higher than the accuracy of the fire-sourced PM_{2.5} from raw GEOS-Chem outputs ($R^2 = 0.18$, RMSE = 22.96 $\mu\text{g}/\text{m}^3$) (Fig S11).”

It could be too conservative and misleading to test our estimates based on model trained without any US stations, as US stations make a large proportion of our model training data (~36% of observations), please see our detailed response to this issue in the response to your fourth question.

3. The other reason validation statistics can be inflated in the daily estimates is if a lot of variation in PM levels comes from seasonal differences or spatial differences in average PM, and the model is accurately reproducing these seasonal averages/ spatial differences but not getting the daily variation correct. The way to check this is, in the spatial CV, calculate the r^2 separately for each station but include month-of-year intercepts to take out the seasonal pattern in the analysis of each station. The r^2 is then computed on the residuals once month intercepts have been taken out (often called the “within” r^2 in statistics packages, e.g. the *fixest* package in R), and the authors can then report the distribution of r^2 across stations to give the reader a sense of the distribution of performance.

[Response] We followed your insightful suggestions and have reported the within R^2 (Table S3) as well as the station-specific R^2 for the 10-fold spatial CV (Fig S9). We added some descriptions of these results in the Methods section (page 31):

“The model also showed similar high accuracy in the spatial 10-fold CV for both PM_{2.5} ($R^2 = 0.89$, RMSE = 9.24 $\mu\text{g}/\text{m}^3$) and O₃ ($R^2 = 0.80$, RMSE = 19.64 $\mu\text{g}/\text{m}^3$) (Fig S8), suggesting good spatial generalization abilities of the trained random forest models.

We calculated station-specific R^2 based on the 10-fold spatial CV. The median station performance was comparable to overall model performance (median station-specific R^2 : 0.80 for PM_{2.5}, and 0.72 for O₃), with 90% of station-specific R^2 above 0.38 for PM_{2.5} and above 0.53 for O₃. There were notable spatial variations of the station-specific model performance (Fig S9). Although most stations showed good performance, some PM_{2.5} stations in middle and southwestern US, Hawaii islands, southern Europe, Africa, and western and inland Australia, and some O₃ stations in Chile, South Africa and New Zealand showed poor performance.

We also estimated the within- R^2 of the 10-fold spatial CV which was calculated by regressing station observations on model estimates while controlling for the station and year fixed effects. As calculated by the *fixest* R package, the within- R^2 of spatial CV was 0.81 and 0.74 for PM_{2.5} and O₃ respectively (Table S3). This suggests that our random forest models can predict on average 81% and 74% local temporal variations of all-source PM_{2.5} and O₃, respectively, not just variations in average PM_{2.5} and O₃ across locations and years.”

4. A third and related validation point again speaks to the fact that there are very few monitors in SSA, South America, South East Asia, where wildfire PM_{2.5} accounts for the majority of the worldwide smoke exposure by their estimation. While the authors of course cannot do anything about the low station count, they could try additional experiments to see whether models trained on one continent predict well in another continent with few stations. So could a model trained in North America make accurate predictions in Europe without training on Europe? Again the worry is that the model is being pushed hard to make predictions in locations where it has seen very little data and might be doing a bad job without us knowing it, and so we should do everything we can to evaluate whether the model can make accurate predictions over very large areas without having been trained there. The only way to do this is to hold out a region with a bunch of stations, retrain the model without those stations, and then evaluate.

[Response] We agree with your concern that the trained model may not perform well in some regions (e.g., Sub-Saharan Africa, South America and Southeast Asia, and Oceania) with few or no stations. We made some additional efforts to address and evaluate this issue.

First, we made a more intensive search of monitored PM_{2.5} and O₃ data for those areas, and have added 204 new PM_{2.5} stations in Africa (South Africa, Algeria, Nigeria), South America (Chile, Argentina), Southeast Asia (Indonesia, Philippines, Singapore, Vietnam) and New Zealand, and 146 new O₃ stations in South Africa, Chile, and New Zealand. We have added these new data sources in the Methods (page 29) and updated the description of the station data in Table S1-S2 and Fig S4. With those new training stations, we have updated the trained ML model and all model estimates of fire-sourced PM_{2.5} and O₃, and thus all tables and figures accordingly. Although the exposure assessment results did not change substantially, we believe that the ML models with more training stations in those regions tend to be more robust than the previous version.

Second, as suggested, we performed a cross-continent validation (i.e., for each continent, using model trained in other continents to predict PM_{2.5} or O₃ in this continent), the results were shown below in Table R1. It seems that the performance of the model in some continents were too low, and even lower than the raw GEOS-Chem outputs (meaning that the model trained completely in other continents could even add more bias to the GEOS-Chem rather than add more accuracy). However, we don't think this is a reasonable test of our model performance because it does not mimic the real situation of our model training by assuming the testing continent was completely excluded in model training.

Table R1. Cross-continent validation of the machine learning model estimating all-source daily PM_{2.5} and O₃

Continent	PM _{2.5}				O ₃			
	No. of stations	No. of observations	R ²	RMSE	No. of stations	No. of observations	R ²	RMSE
Africa	208	259855	0.27	18.08	165	315284	0.26	27.47
Asia	1823	2857198	0.23	32.47	1831	2865348	0.34	32.54
Europe	1129	2446285	0.23	14.41	2393	10043651	0.59	21.34
North America	2203	3456399	0.02	29.03	2247	7459422	0.57	22.13
Oceania	193	384091	0.06	27.28	98	263885	0.20	29.36
South America	105	124352	0.04	52.80	117	150245	0.33	38.13
Global average	5661	9528180	0.11	27.39	6851	21097835	0.53	23.79

Taking Africa as an example, the global ML model training actually include 208 African PM_{2.5} stations, rather than completely no station. Therefore, what we want to test is whether the model trained in other continents' stations together with limited number of African stations (not zero African stations) can predict PM_{2.5} in African areas without training data (e.g., Central Africa). For this situation, the cross-continent CV is too conservative and might be misleading as it assumed there was completely no stations in Africa. Instead of performing cross-continent CV, we performed a spatial cluster-based CV to test the models' prediction ability in large remote areas with essentially no training stations.

We have added detailed descriptions of this CV procedure in Methods (lines 859-870, page 32):

"We tested the spatial generalization ability of the models further using a spatial cluster-based CV approach. Specifically, we conducted a K-Means cluster analysis based on the Euclidean distances between stations based on their longitude and latitude, and the optimal number of spatial clusters was determined by selecting the minimum sum of squares distances within groups. As a result, we identified 75 spatial clusters for PM_{2.5} stations and 99 spatial clusters for O₃ stations across the globe. Then, we used each cluster as a testing dataset and the remaining clusters as the training dataset to train and test our random forest model 75 and 99 times for PM_{2.5} and O₃, respectively. Compared with spatial 10-fold CV where the nearby stations could be allocated to training and testing dataset simultaneously, the spatial cluster-based CV increases the difficulty of the prediction task, but is a more realistic test of the models' prediction ability in large remote areas with essentially no training stations (e.g., many areas in Africa, South America, see Fig S4)"

The spatial cluster-cluster based CV results were presented in Table S3 and described in Methods (lines

896-902, page 32):

“As expected, the model performance of spatial cluster-based CV (PM_{2.5}: R² = 0.69, RMSE = 14.79 µg/m³; O₃: R² = 0.67, RMSE = 18.14 µg/m³) was lower than spatial 10-fold CV, but were still much higher than the performance of the raw GEOS-Chem outputs (PM_{2.5}: R² = 0.48, RMSE = 31.00 µg/m³; O₃: R² = 0.47, RMSE = 46.81 µg/m³) across the globe and in all continents. This suggests that our models can predict the daily all-source PM_{2.5} and O₃ in large remote areas with no training data with an accuracy much higher than the raw GEOS-Chem outputs alone.”

Table S3. Global and continent-specific spatial 10-fold cross-validation (CV) and cluster-based CV of the machine learning model estimating all-source daily PM_{2.5} and O₃

Continent	No. of stations	No. of observations	Spatial 10-fold CV			Cluster-based CV			Performance of raw GEOS-Chem outputs	
			R ²	RMSE	within-R ²	R ²	RMSE	No. of clusters	R ²	RMSE
PM _{2.5} Africa	208	259855	0.53	12.22	0.36	0.34	15.78	11	0.09	22.68
Asia	1823	2857198	0.87	13.68	0.85	0.56	23.34	17	0.31	51.93
Europe	1129	2446285	0.77	6.11	0.74	0.53	8.33	13	0.32	14.12
North America	2203	3456399	0.72	4.06	0.67	0.39	5.68	16	0.22	9.03
Oceania	193	384091	0.51	4.97	0.45	0.19	7.07	11	0.11	13.72
South America	105	124352	0.60	25.50	0.48	0.38	36.96	11	0.0004	60.80
Global	5661	9528180	0.89	9.24	0.81	0.69	14.79	75	0.48	31.00
O ₃ Africa	165	315284	0.43	25.29	0.43	0.31	23.11	18	0.28	46.39
Asia	1831	2865348	0.77	23.35	0.71	0.53	26.40	19	0.31	49.04
Europe	2393	10043651	0.79	18.71	0.74	0.70	16.37	19	0.53	45.38
North America	2247	7459422	0.82	19.20	0.76	0.68	16.15	19	0.47	48.36
Oceania	98	263885	0.70	14.19	0.70	0.49	12.80	11	0.29	34.87
South America	117	150245	0.63	20.14	0.51	0.32	25.36	13	0.07	38.42
Global	6851	21097835	0.80	19.64	0.74	0.67	18.14	99	0.47	46.81

Notes: some spatial cluster could be in two continents; this is why the global total cluster number is smaller than the sum of continent-specific cluster number. The unit of the root mean squared error (RMSE) is µg/m³ for both PM_{2.5} and O₃. The continent-specific CV results was calculated by extracting the each continent’s CV data from the global spatial 10-fold CV or global cluster-based CV.

5. The partitioning between wildfire and other sources of PM relies on GEOS-Chem which in turn relies heavily on the emissions inventories the model is fed. GFED is of course a widely-used source, but existing work in the US shows that feeding different emissions inventories — all of which are widely in use — into the same transport model gives very different concentration estimates — often up to 20x different in a given region. See Koplitz, Nolte et al 2018 Atmospheric Environment. This suggests that we might be pretty worried about estimates just using one inventory as input. I see a few options here. The first is if the correct out-of-sample performance metrics performed above - if these still look good then the authors maybe have a strong argument that this Koplitz-type result is not a problem for them (and/or that GFED is the best inventory). The second and more painful is to also run this with a different inventory and see whether results change. A third is to try to make the argument that GFED is fine — but again I don’t think the existing lit is their friend in that regard, but maybe there are more recent papers that support the use of GFED relative to other options.

[Response] We agree that the uncertainty of fire emission inventories could be a major concern. Following your advice, we have addressed this concern as follows.

First, the out-of-sample performance metrics all look good and support the validity of the GFED-based estimates, particularly our GFED-based estimates of fire-sourced PM_{2.5} showed good agreement with Childs et al’s smoke PM_{2.5} (see our response to your second question).

Second and third, we collected data of other three widely used global fire emission inventories, including

Fire Inventory from NCAR version 1.6 (FINN1.6, available for 2002-2017), Quick Fire Emission Dataset version 2.5 (QFED2.5, available for 2000-2017), and the Global Fire Assimilation System version 1.2 (GFAS1.2, available for 2003-2018), and used them to run the GEOS-Chem simulations as well as the machine learning calibrations just like our primary analyses. We summarized the results and conclusions of comparing different inventories in the Discussion as follows (lines 643-656, page 27):

“Studies suggest that GEOS-Chem simulations based on different fire emission inventories may generate quite different estimates of fire-sourced PM_{2.5} in North America. However, according to our validation results (Table S5-S6), the GFED4.1s was the best inventory among four widely used inventories considering both accuracy (i.e., agreement with ground station observations and the Childs et al’s smoke PM_{2.5}) and data availability, and it is also currently the most widely used one. Our results also suggest that the estimates of all-source and fire-sourced PM_{2.5} and O₃ based on three alternative inventories were mostly highly consistent with GFED-based estimates, and the consistency improved after machine learning calibration (Fig S16-S19). Furthermore, even based on the FINN (the inventory showed the least agreement with GFED-based estimates of fire-sourced PM_{2.5}), the generated estimates of fire-sourced PM_{2.5} and O₃ showed very similar spatial distribution, temporal trends and seasonal patterns with GFED-based estimates (Fig S20-S23). Therefore, our assessment of population exposure to fire-sourced air pollution was robust against the choice of fire emission inventories.”

More details about the procedures of comparing different fire emission inventories and performing sensitivity analyses using FINN were presented in the last section of the Methods (lines 996-1043, pages 34-35).

6. For an outlet like Nature, I would encourage the authors to think hard about the few visuals that most compellingly communicate the message they want the reader to take home from the paper, and then pare down their large set of main text visuals to just those. Right now the visuals are tiny, hard to read, overwhelming in number, and not always necessary to support the narrative. Fig 1 is good, Figs 2-5 have a lot of repetitive information, not all of which is useful in the narrative, and Fig 6 looks like it was copy-pasted from the Lancet. I think the goal of the figures here is not to just spit out every analysis and output that was done, but to focus on the key part of the story - I would encourage them to do that.

[Response] Based on yours and other referees’ suggestions, we have significantly improved our figures, including: 1) increased the font size; 2) changed the daily times in Fig 2 to yearly time series to make the trend clearer; 3) increased the size of trend lines to make them more visible; 4) moved Fig4 to supplement and replaced it with a Table 2 with clearer presentation of the information; 5) removed Fig 5 whose information has been convey by other figures and tables; 6) simplified Fig 6 (now the Fig 4) as describing changes from 2000-09 average to 2010-19 average (rather than changes between 2000, 2010, and 2019). Please see the updated figures.

Referee #2 (Remarks to the Author):

This manuscript looks at human exposure to wildfire smoke, trends over 2000-2019, and disparities between countries, income, etc. While this is an interesting study, I have significant concerns with aspects of the study, particularly the use of a single emissions inventory (when different inventories have shown different global trends during this time period). Hence, I cannot recommend publishing this paper without major revisions.

[Response] Thank you so much for all your insightful comments and suggestions. **Please note that the page and line numbers mentioned below refer to the marked-up manuscript with track changes.**

Major issues

Significant unstated and understated issues in the estimates: There are a number of issues with the estimates that are either not discussed or discussed very little.

(1) The ML is (necessarily) tuning the total PM_{2.5} and O₃ and not the wildfire specific PM_{2.5} and O₃. Hence, it fully relies on GEOS-Chem for the smoke vs. non-smoke split, so any issues with this split (e.g., from #2 and #3

below) are not corrected for, nor is it clear how well the ML would correct for those issues. [Response] We agree that relying on the GEOS-Chem for the fire vs. non-fire split may have some limitations, we have discussed or addressed these limitations carefully, please see our response to your following two questions.

Although the ML is tuning the total $PM_{2.5}$ and O_3 , we indeed found some evidence that the ML can also add more accuracy to the fire-sourced $PM_{2.5}$ compared with raw GEOS-Chem outputs, when validated against the station observed smoke $PM_{2.5}$ provided by Childs et al. Please see our response to the second question of Referee #1. This suggests that the ML did a good job in correcting the errors of both total and fire-sourced $PM_{2.5}$ from GEOS-Chem.

Here is the likely reason why ML trained to predict total $PM_{2.5}$ can also improve the accuracy of fire-sourced $PM_{2.5}$. During wildfire events, the proportion of fire-sourced $PM_{2.5}$ among all-sources were often high (e.g., often >50% during wildfire days, see Fig R1 for an example based on our 10 selected wildfire events used for validation). In this case, the fire-sourced $PM_{2.5}$ is the main contributor of the total $PM_{2.5}$, thus correcting or calibrating the total $PM_{2.5}$ is close to or even equal to (in an extreme 100% fire contribution) correcting the fire-sourced $PM_{2.5}$.

Fig R1. The proportion of fire-sourced $PM_{2.5}$ among all sources during the days of 10 selected wildfire events.

Notes: Frequency refer to number of days during wildfire events. Details of these wildfire events can be found in Table S4 in supplementary materials.

(2) GEOS-Chem w/ GFED does not include any fire-specific plume rise scheme. Depending on the model setup, the emissions are either entirely into the boundary layer or a globally fixed split between the boundary layer and lower/mid free troposphere. This leads to known model biases in regions downwind of fires that mostly receive lofted smoke rather than smoke at the surface (e.g., the US Northern Great Plains in <https://doi.org/10.1021/acs.est.8b05430>). Even if the ML somewhat picks up on the biases when smoke is present, the smoke vs. non-smoke split will be wrong.

[Response] We have added detailed discussion of this limitation in Discussion section (lines 658-668, pages 27-28 of the **marked-up manuscript**):

“Second, our GEOS-Chem simulations did not account for plume rise and assumed that all fire emissions were emitted at the surface, because there are large uncertainties in the fire plume height data ^{1,2}. A recent study found that including the fire plume rise did not always improve the accuracy of simulated $PM_{2.5}$ and O_3 ³. GEOS-Chem simulations without considering plume rise can overestimates the contribution of fire emissions to surface $PM_{2.5}$ and O_3 in fire source regions while underestimating the impacts of fire emissions in regions downwind from fire source ^{1,3}. Given that fire source regions (e.g., wildlands or agricultural lands) tend to have smaller population densities than other regions, our GEOS-Chem approach

is likely to cause an underestimation of global population exposure to fire-sourced air pollution. Further studies are warranted to quantify and correct the bias caused by omitting plume rise.”

(3) This is my biggest issue with the manuscript. The trends in wildfire-specific PM_{2.5} are going to be driven almost entirely by the choice of emission inventory. The different emission inventories have quite different trends, and this is highlighted well in Carter et al, 2020: <https://doi.org/10.5194/acp-20-2073-2020> Please see Figure 5 of this paper. During the 2004-2016 window, two inventories (FINN and QFED) appear to have decreasing emissions trends globally while GFED does not (GFAS perhaps has a small decreasing trend or no trend). Hence, I would expect the results in the current manuscript, especially the trends, to depend greatly on the choice of inventory. Hence, it seems like some other inventories might not have the same increase in exposure per person as GFED. Further, GFED is not clearly the best inventory in the Carter paper (though only evaluated over North America). The limitation of inventory choice is mentioned just very briefly in the 2nd to last paragraph, but this issue seems too big for publishing without testing the sensitivity to the inventories.

[Response] We agree with your concern and have added more detailed discussion on this issue, as well as sensitivity analyses based on FINN, QFED and GFAS. The results suggested that our model estimates and population exposure assessment (spatial and temporal variations) were robust against the choice of inventories, and the GFED was the best inventory for our study considering accuracy and data availability. Please see our detailed response to the 5th question of Referee #1.

(4) The ML inputs are a dump of variables that do not all have a physical connection to affecting PM_{2.5} or O₃, yet some key variables that relate to how wildfires affect these variables are left out. For example, the altitude of the 0.25x0.25 deg box relative to the mean of the 2x2.5 deg box might relate to smoke pooling in low-lying areas overnight, and the distance from nearest fire and direction of the fire relative to the wind direction might help catch for smoke spikes. It seems like a lot of variables were thrown at the ML in an attempt to fit things against all monitors globally, but not a lot of thought was given to the factors that might help get the smoke impacts right.

[Response] As we presented in main text, the ML input predictors are all important and related to the PM_{2.5} and O₃, the PM_{2.5_chem_total} and O_{3_chem_total} accounted for the impacts of different emissions (including fire emissions) on air quality under different meteorological conditions (including wind directions and speeds), T_{mean} to UV variables further accounted for the impacts of meteorological conditions, Year to Lat variables accounted for temporal and spatial variations.

$$PM_{2.5_station} = f(PM_{2.5_chem_total}, T_{mean}, T_{max}, T_{min}, TV, RH, Wind_u, Wind_v, Precip, Pressure, UV, Year, Month, DOW, DOY, Lon, Lat) \quad (1)$$

$$O_{3_station} = f(O_{3_chem_total}, T_{mean}, T_{max}, T_{min}, TV, RH, Wind_u, Wind_v, Precip, Pressure, UV, Year, Month, DOW, DOY, Lon, Lat) \quad (2)$$

According to your suggestion, we tried to add the following variables as additional predictors to the ML model, including:

(1) two altitude variables: the absolute altitude of the 0.25°×0.25° grid box, and the altitude of the 0.25x0.25 deg box relative to the mean of the 2°×2.5° box (i.e., the difference between the small box and the large box)

(2) eleven fire related variables: we used the MODIS/TERRA daily active fire intensity (fires counts per unit area) data at 0.25°×0.25° spatial resolution (product no: MOD14A1). Based on this product and the wind direction information retrieved from ERA5 (i.e., wind_u and wind_v variable used in our primary model), we linked the following variables:

- Fire intensity in the 0.25°×0.25° grid box
- The mean fire intensity of all 0.25°×0.25° grid boxes within 1° (about 110km) in the **upstream of the wind direction (UWD)**
- The maximum fire intensity among all 0.25°×0.25° grid boxes within 1° (about 110km) in the UWD
- The mean fire intensity of all 0.25°×0.25° grid boxes within 2° (about 220km) in the UWD
- The maximum fire intensity among all 0.25°×0.25° grid boxes within 2° (about 110km) in the UWD
- The mean fire intensity of all 0.25°×0.25° grid boxes within 5° (about 550km) in the UWD
- The maximum fire intensity among all 0.25°×0.25° grid boxes within 5° (about 550km) in the UWD

- The mean fire intensity of all $0.25^\circ \times 0.25^\circ$ grid boxes within 1° (about 1100km) in the UWD
- The maximum fire intensity among all $0.25^\circ \times 0.25^\circ$ grid boxes within 1° (about 1100km) in the UWD
- The distance to the nearest $0.25^\circ \times 0.25^\circ$ grid box with fires in the UWD
- The fire intensity in the nearest $0.25^\circ \times 0.25^\circ$ grid box with fires in the UWD

The UWD defined as the $\frac{1}{4}$ round buffer in the reverse direction of the wind direction in the targeted $0.25^\circ \times 0.25^\circ$ grid box, as illustrated in the following Fig R2. This is based on the hypothesis that only fires in the UWD have the potential to affect the $PM_{2.5}$ and O_3 concentration in the targeted grid box.

Fig R2. The definition of the upstream of the wind direction (UWD), taking 1° buffer as an example.

However, after including these variables, the model performance did not improve as evaluated by spatial 10-fold cross-validation, see Table R1 below. **The most likely reason why these variables added little predictive value to the ML is that their contribution had already been accounted for by the GEOS-Chem outputs (the most important predictor in our ML model).** Therefore, we don't think it is necessary to include these variables into our final ML model. Moreover, the MODIS/Terra fire intensity data are only available from Feb 2000, including these variables will make us unable to generate full time-series ML estimates during 2000-2019.

Table R2. Spatial 10-fold cross-validation results of different models with or without altitude and MODIS fire information

Model	$PM_{2.5}$		O_3	
	RMSE	R^2	RMSE	R^2
Primary model	8.707	0.903	19.585	0.798
Primary model + altitude	8.743	0.899	19.615	0.799
Primary model + fire	8.830	0.895	19.673	0.784
Primary model+ altitude + fire	8.840	0.892	19.620	0.790

Notes: primary model refers to the model used in our main text; altitude refers to two altitude variables, including absolute altitude and relative altitude; fire refers to 11 fire-related variables based on MODIS/TERRA daily active fire intensity (fires counts per unit area) data at $0.25^\circ \times 0.25^\circ$ spatial resolution. Because the MODIS/TERRA fire information was available since Feb 2000, the training and validation dataset used in this table is a bit smaller than the dataset used in our main text, this is why the R^2 and RMSE were a bit different from the ones we reported in main text.

5. Figures: The figures are poor quality, both in readability (small unreadable fonts, pale yellow very hard to see) and context (e.g., 9 panels figures where 7 of the panels are 20 year time series is just impossible to digest). **[Response] Thanks for the suggestion. We have improved the figures significantly according to yours and other referees' suggestions, including: 1) increased the font size; 2) changed the daily times in Fig 2 to yearly time series to make the trend clearer; 3) increased the size of trend lines to make them more visible.**

Please see the updated figures.

Specific comments

Title and throughout: Not all of the fires in GFED are wildfires. Should we be calling this exposure to wildfire smoke when it includes other types of fires? Seems misleading.

[Response] Thanks for pointing this out, we agree with your concern. A wildfire is often defined as uncontrolled or unplanned fire that burns in wildland vegetation ⁴, thus the human planned prescribed or controlled burns (i.e., fires set by experts under less dangerous weather conditions in order to reduce fuels or favour biodiversity) and agriculture fires generally cannot be called “wildfires”. It is more appropriate to call the fires in GFED as “landscape fires” which is a term applied to any fire burning in natural and cultural landscapes, e.g. natural and planted forest, shrub, grass, pastures, agricultural lands and peri-urban areas ⁵. Therefore, we have replaced the term “wildfire” as “landscape fire” in the title and throughout the manuscript when appropriate. We also added some definitions and clarifications about the relationship and difference between “wildfires” and “landscape fires” in the first paragraph of the Introduction.

L37: I was confused by why O₃ was being presented in $\mu\text{g m}^{-3}$ in the abstract before I noticed an explanation in the methods. I bet this will confuse people who are used to thinking of O₃ in ppb. Not sure what's best.

[Response] As we mentioned in methods, “we unified all units of PM_{2.5} and O₃ as $\mu\text{g}/\text{m}^3$ in consistent with the latest WHO air quality guidelines 2021”, thus the $\mu\text{g}/\text{m}^3$ is better than ppb at least for our analyses. I understand ppb is more widely used in US, but $\mu\text{g}/\text{m}^3$ is more suitable for our global study given the WHO guideline and the fact that some countries station O₃ was recorded as $\mu\text{g}/\text{m}^3$. To avoid any potential confusion, we have clearly labeled or noted the unit of O₃ as $\mu\text{g}/\text{m}^3$ in all tables and figures.

L42-43: The comparison to WHO guidelines here also confused me. It gets explained in the paper as being total PM/O₃ > guideline + fraction smoke over a threshold, but here I wasn't sure what it meant. Was smoke PM/O₃ alone higher than the guideline? Was the total simply over the guideline. We don't actually learn this until L153

[Response] Nature has very tight word limit on the abstract (within 150 words), it is not possible to explain this in abstract. To avoid confusion, we have deleted “comparing with WHO guideline 2021” in the abstract and will let readers to find the definition of substantial fire-sourced air pollution in the manuscript.

L70-72: I was confused as to why Brazil was getting specific attention here when smoke health impact assessments have been done on different countries as well but not mentioned. Then I realized it was just the authors citing themselves but not others. Not the best look.

[Response] We have deleted the citation to the Brazil study here, please see lines 79-80, page 3.

L79: “is” should be “are”

[Response] Done.

L123-125: What are the ranges here? It seems like interannual variability? To me, the spatial variability within the countries/regions is more important. It definitely looks like there are parts of North America where the contribution is <9.2%. Why would we want to spatially average a whole continent when some people are exposed to much more smoke than others while giving temporal variability? It would seem like the variability in people's mean exposure is more important to show than the interannual variability.

[Response] We agree that the spatial variability of the fire contribution could be more important. Therefore, we have added a FigS1 to map the spatial variability of the landscape fire contribution to PM_{2.5} and O₃, and have revised the descriptions as follows (lines 184-191, page 5):

“The proportions of fire-sourced PM_{2.5} and O₃ among all sources showed similar spatial distribution for the 2000-2009 and 2010-2019 (Fig S1). The highest landscape fire contribution to PM_{2.5} was observed in Central Africa (up to 70%), followed by South America (~40%), Northern Australia (~40%), Southeast Asia (~30%), Western US and Canada (~20% in 2000-2009, increased to ~30% in 2010-2019), and Northeast Asia (~20%).”

The highest landscape fire contribution to O₃ was also observed in Central Africa (up to 46%), followed by South America (~30%), Northern Australia (up to 20%), and Southeast Asia (up to 20%).”

Fig. S1. Proportions of fire-sourced PM_{2.5} and O₃ among all sources during 2000-2009 and 2010-2019 across the globe.

L141-143: Again, what is the variability here? It seems like temporal variability, but the variabilities between countries/populations seem more important.

[Response] Yes, it is the temporal variability here. We agree that variabilities between countries/populations could be important, but this information has been reported in Figure 1 and Figure 4, we don't need to report it repeatedly here. Instead, here we aimed to give a summary of the overall disparity in the exposure levels between different HDI and income groups (i.e., an overall description of the socioeconomic disparity), the only variability is from temporal fluctuations. We have revised the sentence as follows to make it clear that we reported temporal variability (lines 194-203, page 6):

“There were quite consistent socioeconomic disparities in the annual average fire-sourced PM_{2.5} and O₃ concentrations (Fig 3, Table 1). Countries with low human development index (HDI) and low income had the highest exposure to fire-sourced air pollution, while countries with very high HDI and high income had the lowest exposure. The annual population-weighted average fire-sourced PM_{2.5} concentrations in low HDI countries were 2.9- to 4.2-fold of those in very high HDI countries in different years during 2000-2019. These ratios for annual fire-sourced O₃ (low HDI versus very high HDI) were 4.1 to 7.8 in different years. Similarly, annual fire-sourced PM_{2.5} and O₃ in low income countries were 4.5- to 6.2-fold, and 3.9- to 8.1-fold of those in high income countries in different years, respectively.”

L179: What does “population-averaged days” mean here? The average number of days of exposure per person? Please state it more explicitly.

[Response] Yes, to make it clearer, we have revised it as follows:

“Africa had the highest average days exposed to SFAP per person per year……”

Figure 6 and associated discussion: There is huge interannual variability in fires in any given region. Comparing

2000 to 2010 to 2019 is insufficient for looking at how exposure in different countries is changing since any given year could be anomalously better or worse than the surrounding years. There is little that we should be taking from the changes between the 3 compared years. If you wanted to stick with a similar analysis, I would compare the first 10 years averaged to the final 10 years averaged to look at how exposure might be changing across countries while removing some of the interannual variability.

[Response] Great suggestion, we have updated this figure in the form of comparing the first 10-year average (2000-2009) to the last 10-year average (2010-2019), rather than comparing three individual years (2000,2010, 2019). The Table 1 and Table 2 were also updated in a similar way (reporting 2000-09 and 2010-19 average rather than only 2000, 2010 and 2019).

Figure 1: Please add some sign of significance of the trends in the maps in panels E and F.

[Response] Done, please see the updated Fig 1.

Figure 5: Is this for PM_{2.5} or O₃?

[Response] The exposure metrics in this figure has synthesized information of both PM_{2.5} and O₃ exposure, as stated clearly in the definition of substantial fire-sourced air pollution. However, considering all referees' comments, we decided to delete this figure for simplicity, given that its core information has been conveyed by other tables and figures.

L334-335: And forests have a lot more fuel that can burn than savannas/grasslands.

[Response] We have mentioned this point in our discussion (line 463, page 24), thanks for the suggestion.

L392: "several catastrophic wildfire events". The recent bad years in North America generally have had 10s of very large wildfires in each year, so "several" does not seem like the right word.

[Response] We have deleted "several" here.

L437: Smoke can travel 1000s of km and still have significant AQ impacts. E.g., <https://go.nasa.gov/3V83obX>

[Response] We have revised this sentence as:

"The former can often travel hundreds (sometimes even thousands) of kilometres and affect much larger populations"

L454: "we have much higher accuracy" Did you quantify this? You can actually evaluate wildfire specific PM_{2.5} since monitors do not measure it, so how do you know?

[Response] This is proved by our validation against the PurpleAir station observed smoke PM_{2.5} by Childs et al, please see our response to your first question.

L466: "increases in CO are generally restricted to the immediate wildfire areas". CO has a longer atmospheric lifetime than PM_{2.5} and O₃, and it will move with the plume the exact same way, so this statement is very strange. Do you mean, "the region where CO is at dangerous levels is generally restricted to the immediate wildfire areas"?

[Response] We agree that CO has longer lifetime and can also travel with fire smoke. However, Tao et al (Ref 13 in our manuscript)'s simulation study in US has concluded that "the fire impacts on CO were generally confined to the fire source areas". This phenomenon could be explained by the photochemical loss of CO (i.e., photochemical oxidation of CO and hydrocarbons in the presence of nitrogen oxides produces O₃) during long-distance transport of biomass plumes⁶. We have added this explanation in the Discussion of CO (lines 631-635, page 27) to make this point clearer.

L585-590: How good is GEOS-Chem alone before the ML?

[Response] We have added this information in Table S3 and Fig S11 and Table S6, they all suggest that the ML made significant improvement of the accuracy of all-source PM_{2.5} and O₃, and fire-sourced PM_{2.5} compared with the GEOS-Chem alone.

For example, in Table S3, the performance of GEOS-Chem all-source PM_{2.5} before ML ($R^2 = 0.48$ and RMSE = 31 $\mu\text{g}/\text{m}^3$) is lower than the ML-calibrated performance (spatial 10-fold CV: $R^2 = 0.89$ and RMSE = 9.24 $\mu\text{g}/\text{m}^3$). In Fig S11, when validated against smoke PM_{2.5} observed by PurpleAir station, the performance of our ML estimated fire-sourced PM_{2.5} ($R^2 = 0.512$, RMSE = 11.76 $\mu\text{g}/\text{m}^3$) is also much better than the GEOS-Chem fire-sourced PM_{2.5} ($R^2 = 0.176$, RMSE= 22.96 $\mu\text{g}/\text{m}^3$).

L605: I don't think you can actually evaluate wildfire-specific PM_{2.5} and O₃ against obs since the obs measure the total. These correlation coefficients seem misleading/incorrect.

[Response] To avoid confusion as you mentioned, we have deleted the correlation coefficients between estimated fire-sourced PM_{2.5}/O₃ and observed all-source PM_{2.5}/O₃ in these figures, please see the Fig S12-S13.

L605: Also, in these comparisons, how was the ML trained and evaluated? When the ML was trained, were some monitors left out by other nearby monitors were included with the model being trained on the smoke time period? If yes, or something similar, it seems like this evaluation would do nothing to show the confidence in how well the ML does in locations away from monitors.

[Response] Following your and Referee #1's advice, we have used out-of-sample model training and validation approach. Specifically, for each wildfire event, we selected five nearest stations. Among the five stations, we chose the station experienced most significant increase in observed PM_{2.5}(or O₃) during wildfire event compared with pre-wildfire period (i.e., the most affected station). Then we trained a ML model without the data of all the five nearest stations (i.e., all nearby stations left out), the ML model was then validated in the most affected station. Please see our detailed response to Referee #1's first question.

Referee #3 (Remarks to the Author):

This is an interesting study in which the authors use a combination of chemical transport model simulations and machine learning to estimate global daily exposure to PM_{2.5} and ozone from wildfires. The authors present some relevant and novel metrics for this exposure, shown in ways that haven't been done previously to characterize the exposure. They find that wildfire smoke is a significant contributor to global air pollution exposure (4-7%), that wildfire smoke is much greater in certain low-income regions, and that exposure to PM_{2.5} is increasing through time. These findings are presented comprehensively in a way that I expect many researchers will be interested in.

[Response] Thank you so much for all your insightful comments and suggestions. **Please note that the page and line numbers mentioned below refer to the marked-up manuscript with track changes.**

But there are also weaknesses that work against the novelty of the paper. The paper is presented as an extension of the methods they applied in the authors' previous papers (l. 86). That is, the combination of a CTM with machine learning is not new here but was presented in their previous papers. Here the **novelty** is the global application, which makes the results comprehensive and interesting. There have also been previous global-scale wildfire smoke exposure and health impact studies, as the authors discuss in the conclusions, where it seems the advantage of this paper is in the machine learning to bias correct.

[Response] Although our previous studies have used the combination of a chemical transport model (CTM) with machine learning (ML) to estimated fire-sourced PM_{2.5}, the present paper has contributed to many novel data and insights.

First, our previous studies only estimated fire-sourced PM_{2.5} for Brazil and 749 worldwide locations during 2000-2016, while the present study generated global gap-free daily fire-sourced PM_{2.5} and O₃ during 2000-

2019. The spatiotemporal coverage has been improved significantly and the new data addressed the critical gaps in many regions, particularly Africa, Southeast Asia, South America where landscape fires are frequent. The estimated fire-sourced O_3 data product is the world's first global fire-sourced O_3 estimation (previous global studies only estimated fire-sourced $PM_{2.5}$ using CTM, but not fire-sourced O_3 ; only few studies estimated fire-sourced O_3 in US using CTM).

Second, the present study performed comprehensive validations (e.g., 10-fold general and spatial CV, spatial cluster-based CV, validation against large wildfire events and the smoke $PM_{2.5}$ by Childs et al) of our estimates. Our previous studies only provided 10-fold general CV results for all-source $PM_{2.5}$. Overall, these new validation results suggest that our estimated all-source $PM_{2.5}$ and O_3 have good accuracy compared with ground observations (Fig S7-S9, Table S3), and the fire-sourced $PM_{2.5}$ and O_3 can well capture the impacts of wildfires (Table S4, Fig S12-S15), and showed good agreement with Childs et al (Fig S10-S11). These provide novel and essential information about the validity and accuracy of our estimates, which forms a critical basis for the global application of the data, including population exposure assessment in our main text and assessing various health impacts of landscape fire-sourced air pollution in future.

Third, our study compared performance of different fire emission inventories (GFED, FINN, GFAS, QFED, see our response to the 5th question of Referee #1) in the CTM+ML approach. This is new to literature, as previous studies either only use one inventory (mostly GFED) or only compare their performance in CTM (without ML calibration). Our results suggest that GFED is generally the best inventory considering data availability and accuracy (Table S5-S6), and we also found that **ML correction can improve the agreement between different inventories** (Fig S16-S19). These novel results provide important information for the selection for fire emission inventories in future studies and also further supports the robustness of our exposure assessment (e.g., consistent spatial temporal pattern using FINN-based estimates, Fig S20-S23).

Fourth, we provided a map of the different fire types (boreal forest fires; tropical forest fires; savanna, grassland, and shrubland fires; temperate forest fires; peatland fires; agricultural waste burning) based on the GFED (Fig S5-S6). This provides important information for readers to understand the different fire types' spatial distribution and relative contributions to fire-sourced air pollution across the globe.

Finally, as you agree, the global application of the estimated data generates comprehensive and interesting results, such as the geographical distribution, long-term trends, seasonal pattern and socioeconomic disparities. Most of these results are novel to literature. For example, the two previous global studies estimated global daily fire-sourced $PM_{2.5}$ for 1997-2006⁷ and 2016-2019⁸ using CTM (no ML correction). The first study mainly focused on mortality burden estimation⁷, and the second one only reported population exposure assessment results at annual scale (not daily scale)⁸, and their short temporal coverage determines that they cannot evaluate long-term trends. Moreover, as we presented in Table S3 and Fig S11 and Table S6, the accuracy of all-source $PM_{2.5}/O_3$ and fire-sourced $PM_{2.5}$ from raw CTM outputs were much lower than the ML-corrected estimates.

In summary, the present study has provided many important novelties, including novel comprehensive data, novel insights into the CTM+ML methodology, novel and by far the most comprehensive assessment of the population exposure to fire-sourced $PM_{2.5}$ and O_3 at multiple spatial-temporal levels (global/regional/national, yearly/monthly/daily). Therefore, our study provides the currently most accurate and comprehensive data for the policy makers and the public to manage and mitigate landscape fire-sourced air pollution at global scale. The generated database also formed a critical basis for many future applications, such as evaluating various health impacts of this environmental hazard⁹, and estimating corresponding attributable mortality, morbidity, and healthcare costs^{7,10}.

We have discussed these novelties in detail in Discussion section (lines 588-627, pages 26-27)

The finding that wildfire smoke is a few percent of total global exposure is not surprising. The details provided on the location of greater exposure and trends are worthwhile and interesting. Still the paper leaves questions about

what to do with the information. There is rather little discussion of health consequences. Unlike anthropogenic or industrial emissions, wildfire smoke is not easily controlled. In places like the Western US, past fire suppression has been counter-productive. Perhaps only in low-income countries where wildfire is related with forest clearing does the policy-relevance become more apparent. This doesn't get much discussion.

[Response] We have added one paragraph to discuss these points (health consequences, landscape fire control, public health implications and health protection strategies)(lines 423-447, page 23):

“Our assessment highlighted the severity and scale of the fire-sourced air pollution and a notable increasing trend in the population exposure. Short-term exposure to fire-sourced air pollution has been linked to many adverse health consequences, including increased mortality, and exacerbations of respiratory and cardiovascular conditions that lead to hospital admissions and emergency department visits ^{9,11,12}. The large quantity (e.g., 72.8 billion person days exposed to SFAP per year during 2010-2019) and increasing trend of the population exposure to SFAP suggests that landscape fire air pollution is an increasing public health concern. Addressing this concern needs multi-sector efforts to reduce landscape fires and protect people from the adverse health impacts of landscape fire air pollution. Landscape fires can be partially prevented through effective evidence-based fire management, as well as appropriate planning and design of natural and urban landscapes ¹³. Policy change may help to prevent or reduce some landscape fires directly cause by human, such as agricultural wasting burning in Europe, India, Eastern China and US (Fig S6), and the fires deliberately set by humans to convert wildlands to agricultural or commercial lands (common in South America, South and Southeast Asia ^{5,14}). However, unplanned wildfires are more difficult to control, as evidenced by the fact that aggressive fire suppression actually contributed to the extreme wildfires in western US in recent decade due to fuel accumulation ¹⁵. Wildfires are also an essential component of Earth's ecosystem and can never be totally prevented ¹³. Therefore, a considerable proportion of human exposure to landscape fire-sourced air pollution seems to be unavoidable. This highlights the importance of health protection measures against the exposure. Unfortunately, the measures that individuals can take to protect themselves from landscape air pollution, such as relocation, staying indoors, using air purifiers with effective filters, and wearing N95 or P100 face masks, all have limitations and not friendly for people with limited resources ⁹. More efforts are urgently required to develop cost-effective public health measures to mitigate the health impacts of landscape fire-sourced air pollution.”

Overall, I think the paper is interesting, is likely to be technically sound, and will likely be widely cited by the global air pollution public health community. But to me the factors that reduce its novelty that I mentioned above make it questionable whether it should appear in a high-impact journal such as Nature, and it may be more appropriate for a Nature specialty journal or another journal of similar stature (Lancet Planetary Health). I say that it is likely to be technically sound because I am not an expert in machine learning and the machine learning methods are a bit of a black box. I think that an machine learning expert would need to read previous papers and the supplement carefully to understand what was done, and even with that, there may be possibilities of spurious (non-physical) relationships estimated by the machine learning. Testing the machine learning results with a small number of fires (6) in the USA and Australia seems to be a small sample.

[Response] As we mentioned in the response to your first question, the novelty of this paper is not just the CTM+ML approach, but many other novel data and insights (including methodology insights). As wildfires are increasing with climate change and our assessment highlighted that more than 2 billion people are exposed to substantial fire-sourced air pollution each year, the novel data generated by this study definitely have important social, policy and public health implications. Publishing this paper in a high-impact journal such as Nature will attract more research, media, public and policy attention to this important and emerging topic, which will accelerate the mitigation of landscape fires, climate change, and climate injustice.

In terms of the technical side, both the Referee #1 and #2 are experts of ML and CTM. They have provided many insightful comments and suggestions, and we have addressed them very well with many additional analyses (Fig S9-S23, Table S3-S6) and Discussions, please see our detailed response to the first five questions of Referee #1 and the first four questions of Referee #2.

As for testing the ML estimates against wildfire events, to address your concern, we have added another

four wildfire events (including 2011 Texas Fire in US, 2017 Portugal Fire, 2016 Chile Fire, and the 2015 Western Cape Fires in South Africa, see Table S4) to test the results. The validation results of these added wildfire events are similar or even better than the previous six wildfire events, suggesting the robustness of our estimates. Moreover, we also validated our estimated fire-sourced $PM_{2.5}$ against the smoke $PM_{2.5}$ by Childs et al and found high agreement ($r=0.88$) and similar accuracy (Fig S10-S11), as we summarized in Discussion (lines 592-595, page 26):

“Our estimated fire-sourced $PM_{2.5}$ showed high agreement (Pearson correlation coefficient $r = 0.88$) with the estimated smoke $PM_{2.5}$ by Childs et al, and proved similar accuracy as Childs et al when further validated against PurpleAir station observed smoke $PM_{2.5}$.”

This provides more straightforward evidence about the accuracy of our estimates.

More specific comments:

If I understand their methods correctly, the only input data to the machine learning about where and when fires occur, and their magnitude, comes from the fire emissions database (GFED) which is input to GEOS-chem. That is state-of-the-art, but it should be mentioned that fire emissions are difficult to estimate from space (and in general), GFED may have significant errors in estimated emissions for individual fires, as well as possibly biases more broadly. Where there are no measurements to bias-correct (low-income nations), does then the model essentially use the GEOS-chem modeled estimates? Or use some machine learning corrections based on relationships found where monitors exist (US and Europe)? GEOS-chem also runs at a coarse resolution that is not great for estimating population exposure. Perhaps more discussion of these limitations is warranted.

[Response] We have added one paragraph to discuss the **limitations of GFED** (lines 640-656, page 27):

“First, the GFED4.1s used for GEOS-Chem simulations has some uncertainties and limitations, such as uncertainties in the emission factor and the estimation of burned areas based on satellite images¹⁶. Studies suggest that GEOS-Chem simulations based on different fire emission inventories may generate quite different estimates of fire-sourced $PM_{2.5}$ in North America^{17,18}. However, according to our validation results (Table S5-S6), the GFED4.1s was the best inventory among four widely used inventories considering both accuracy (i.e., agreement with ground station observations and the Childs et al’s smoke $PM_{2.5}$) and data availability, and it is also currently the most widely used one¹⁸. Our results also suggest that the estimates of all-source and fire-sourced $PM_{2.5}$ and O_3 based on three alternative inventories were mostly highly consistent with GFED-based estimates, and the consistency improved after machine learning calibration (Fig S16-S19). Furthermore, even based on the FINN (the inventory showed least agreement with GFED-based estimates of fire-sourced $PM_{2.5}$), the generated estimates of fire-sourced $PM_{2.5}$ and O_3 showed very similar spatial distribution, temporal trends and seasonal patterns with GFED-based estimates (Fig S20-S23). Therefore, our assessment of population exposure to fire-sourced air pollution was robust against change of fire emission inventories.”

Overall, our sensitivity analyses using alternative fire emission inventories (Fig S16-23) and validation against the smoke $PM_{2.5}$ by Childs et al (which used satellite-based smoke plume approach and not relying on fire emission inventories) suggested that **the bias or uncertainties of GFED and the GEOS-Chem have been largely reduced by the ML correction and would not likely to bias our results substantially.**

Regarding the problem that many low-income nations have low station counts for ML correction, this is about the spatial generalization ability of the ML. We have addressed this issue in response to the 4th question of Referee #1. Briefly, we added 204 new $PM_{2.5}$ stations and 146 O_3 stations in Africa, South America, Southeast Asia and Oceania where stations counts were low. We also performed spatial cluster-based CV to test the spatial generalization ability of the ML (lines 859-870 and lines 896-902, page 32 in Methods). We have added one paragraph to discuss this limitation (lines 685-693, pages 28):

“Finally, the machine learning models were trained against station observations dominated by several regions (Europe, US, China), which may not apply to regions with few or no stations. However, according to our spatial-cluster cross-validation which mimics this situation, our models showed good accuracy in predicting observations far away from the training stations (overall R^2 : 0.69 for $PM_{2.5}$ and 0.67 for O_3), and the accuracy was still much higher than the raw GEOS-Chem outputs even in continents (Africa, South America, and Oceania) with small number of stations (Table S3), suggesting that our trained machine model can also add accuracy to the GEOS-Chem in regions with limited or no training stations.”

Regarding the coarse resolution issues, we have added one paragraph to discuss this limitation in detail (lines 670-683, page 28):

“Third, the GEOS-Chem was run at a coarse spatial resolution ($2.0^{\circ}\times 2.5^{\circ}$) which may cause errors in population exposure assessment at high spatial resolution. However, we have performed downscaling of the GEOS-Chem and added higher-resolution meteorological data as additional predictors in the machine learning model. Our validation against PurpleAir station observed smoke $PM_{2.5}$ suggested that our estimated fire-sourced $PM_{2.5}$ can explain about 10% spatial variations of the observed smoke $PM_{2.5}$ within the large $2.0^{\circ}\times 2.5^{\circ}$ grid box (Fig S11), which was a major improvement compared with the raw GEOS-Chem outputs. Moreover, there was almost no correlation between grid-specific population counts and the annual fire-sourced $PM_{2.5}$ ($r=-0.02$) and O_3 ($r=0.001$) concentrations in our data, suggesting that the bias in concentration caused by coarse-resolution of GEOS-Chem tend to be distributed to $0.25^{\circ}\times 0.25^{\circ}$ grid boxes with high or low population counts randomly and cause random errors rather than systematic errors of population exposure assessment. Nevertheless, cautions should be taken if our data are used to perform individual-level exposure assessment in epidemiological studies.”

The authors should also define “wildfire”. I understand that wildfires would not include domestic wood burning. But would it include controlled burns for agriculture? Anthropogenic fires for land clearing, which are set by people and therefore not quite “wildfire”? I think these fires would be in the GFED emissions, and so I think they’re included here.

[Response] We agree that the GFED emissions should include agriculture burns and other anthropogenic fires. We therefore changed the term “wildfire” to “landscape fire” throughout the paper and gave a definition of landscape fires and wildfires in the introduction. The Referee #2 also raised this issue, please see our detailed response to the Referee #2’s 6th question.

I. 65-77 – The authors leave out discussion of other studies that have estimated global wildfire smoke exposure, which they later mention in the discussion.

[Response] We have now added more detailed descriptions of these studies in the Introduction part and discussed the limitations of those previous studies to highlight the novelty and importance of our study (lines 104-114, page 4):

“Two early studies attempted to address the data gap at global scale using chemical transport models, and they estimated global daily fire-sourced $PM_{2.5}$ for 1997-2006⁷ and 2016-2019⁸, respectively. However, the accuracy of chemical transport model outputs could be problematic without calibration against observations of air quality monitoring stations¹⁹, and these two global studies cannot assess the long-term trend of fire-sourced $PM_{2.5}$ given their short study periods. Furthermore, no previous study has estimated global landscape fire-sourced O_3 , and this important fire-related pollutant has only been estimated for US using chemical transport models^{20,21}. Last but not the least, all these previous studies mainly focused on data generation or health impact assessment, little attention has been paid to a comprehensive assessment of the global human exposure to daily fire-sourced air pollution.”

Fig. 3 – I think this would be better as a plot with 4 lines, rather than bars.

[Response] As suggested, we have revised this fig as line graphs, please see the updated Fig 3.

I. 330 – I’m not sure the discussion of CO₂ emissions is very relevant here.

[Response] We have deleted the discussion of CO₂ emission here.

I. 340-354 – Good points to mention, but I don’t think this long discussion is necessary.

[Response] We have significantly shortened this paragraph by about 30%, please see lines 473-492, page 24.

l. 401 – The language here about the climate change influence differs from that presented earlier, where “we postulate”... and “more studies are warranted” (l. 356).

[Response] To avoid confusion and for simplicity of the Discussion, we have deleted the paragraph “we postulate ...”

l. 447-451 – Here references 28-34 are mentioned for the first time. The introduction emphasized the novelty of this study by comparing it with studies of fire exposure. But in fact there were several studies of wildfire smoke exposure that the authors didn't mention until now.

[Re] We have now added more detailed descriptions of these studies in the Introduction part and discussed the limitations of those previous studies to highlight the novelty and importance of our study (lines 93-114, pages 3-4).

l. 465-467 – CO has a longer atmospheric lifetime than PM2.5, O3, or their precursors, so I'm not sure that this statement that CO is restricted to immediate fire areas is correct.

[Response] We agree that CO has longer lifetime and can also travel with fire smoke. However, Tao et al (Ref 30 in our manuscript)'s simulation study in US has concluded that “the fire impacts on CO were generally confined to the fire source areas”. This phenomenon could be explained by the photochemical loss of CO (i.e., photochemical oxidation of CO and hydrocarbons in the presence of nitrogen oxides produces O3) during long-distance transport of biomass plumes⁶. We have added this explanation in the Discussion of CO (lines 633-635, page 27) to make this point clearer.

l. 479 – “world's first”. It seems that this claim is not supported, given references 33 and 34.

[Response] We have removed the “world's first”.

l. 591 – The method of estimating wildfire contribution as a modeled fraction seems simple (though I don't have a better alternative at hand) and may be prone to errors. For example, let's say the model has the smoke plume going in the wrong direction. Within that modeled smoke plume, the machine learning might adjust the PM2.5 down based on a low PM2.5 measurement, but it would still say the fraction from wildfire smoke would be high. Other errors like this might result, and in many cases there won't be a measurement to correct the error. I don't know a better method, but is this worthy of discussion?

[Response] We totally agree that the using GEOS-Chem to determine the fire contribution has limitations. It is also true that there is no other better alternative approach currently. One novel and important contribution of this study is that we demonstrated that the **GEOS-Chem+ML approach could largely decrease the uncertainty and errors in all-source and fire-sourced air pollution estimated by GEOS-Chem alone.**

Following your advice and other two Referees' advice, we have added three paragraphs to discuss the limitations of GEOS-Chem, including the uncertainty of fire emission inventories, not considering plume rise, and coarse spatial resolution. Please see lines 639-683, pages 27-28.

References

- 1 Ke, Z. M., Wang, Y. H., Zou, Y. F., Song, Y. J. & Liu, Y. Q. Global Wildfire Plume-Rise Data Set and Parameterizations for Climate Model Applications. *Journal of Geophysical Research-Atmospheres* **126**, e2020JD033085, doi:ARTN e2020JD03308510.1029/2020JD033085 (2021).
- 2 Sofiev, M., Ermakova, T. & Vankevich, R. Evaluation of the smoke-injection height from wild-land fires using remote-sensing data. *Atmospheric Chemistry and Physics* **12**, 1995-2006, doi:10.5194/acp-12-1995-2012 (2012).
- 3 Tang, W. *et al.* Effects of Fire Diurnal Variation and Plume Rise on U.S. Air Quality During FIREX-AQ and WE-CAN Based on the Multi-Scale Infrastructure for Chemistry and Aerosols (MUSICAv0). *Journal of*

- 4 *Geophysical Research: Atmospheres* **127**, e2022JD036650, doi:10.1029/2022jd036650 (2022).
National Geographic Society. *Wildfires*, <<https://education.nationalgeographic.org/resource/wildfires/>> (2022).
- 5 Global Fire Monitoring Center (GFMC) & the Editorial Board. *Landscape Fires*,
<<https://besafenet.net/hazards/landscape-fires/>> (2023).
- 6 Mauzerall, D. L. *et al.* Photochemistry in biomass burning plumes and implications for tropospheric ozone
over the tropical South Atlantic. *Journal of Geophysical Research: Atmospheres* **103**, 8401-8423,
doi:10.1029/97JD02612 (1998).
- 7 Johnston, F. H. *et al.* Estimated global mortality attributable to smoke from landscape fires. *Environ Health
Perspect* **120**, 695-701, doi:10.1289/ehp.1104422 (2012).
- 8 Roberts, G. & Wooster, M. J. Global impact of landscape fire emissions on surface level PM_{2.5}
concentrations, air quality exposure and population mortality. *Atmos Environ* **252**, 118210, doi:ARTN
11821010.1016/j.atmosenv.2021.118210 (2021).
- 9 Xu, R. *et al.* Wildfires, Global Climate Change, and Human Health. *N Engl J Med* **383**, 2173-2181,
doi:10.1056/NEJMSr2028985 (2020).
- 10 Kollanus, V. *et al.* Mortality due to Vegetation Fire-Originated PM_{2.5} Exposure in Europe-Assessment for
the Years 2005 and 2008. *Environ Health Perspect* **125**, 30-37, doi:10.1289/EHP194 (2017).
- 11 Ye, T. *et al.* Risk and burden of hospital admissions associated with wildfire-related PM_{2.5} in Brazil, 2000-
15: a nationwide time-series study. *Lancet Planet Health* **5**, e599-e607, doi:10.1016/S2542-
5196(21)00173-X (2021).
- 12 Yu, P., Xu, R., Abramson, M. J., Li, S. & Guo, Y. Bushfires in Australia: a serious health emergency under
climate change. *Lancet Planet Health* **4**, e7-e8, doi:10.1016/S2542-5196(19)30267-0 (2020).
- 13 Bowman, D. M. J. S. *et al.* Vegetation fires in the Anthropocene. *Nat Rev Earth Env* **1**, 500-515,
doi:10.1038/s43017-020-0085-3 (2020).
- 14 Vadrevu, K. P. *et al.* Trends in Vegetation fires in South and Southeast Asian Countries. *Sci Rep-Uk* **9**,
7422, doi:10.1038/s41598-019-43940-x (2019).
- 15 Calkin, D. E., Thompson, M. P. & Finney, M. A. Negative consequences of positive feedbacks in US
wildfire management. *Forest Ecosystems* **2**, 9, doi:10.1186/s40663-015-0033-8 (2015).
- 16 van der Werf, G. R. *et al.* Global fire emissions estimates during 1997–2016. *Earth Syst. Sci. Data* **9**, 697-
720, doi:10.5194/essd-9-697-2017 (2017).
- 17 Koplitz, S. N., Nolte, C. G., Pouliot, G. A., Vukovich, J. M. & Beidler, J. Influence of uncertainties in burned
area estimates on modeled wildland fire PM_{2.5} and ozone pollution in the contiguous U.S. *Atmos
Environ (1994)* **191**, 328-339, doi:10.1016/j.atmosenv.2018.08.020 (2018).
- 18 Carter, T. S. *et al.* How emissions uncertainty influences the distribution and radiative impacts of smoke
from fires in North America. *Atmospheric Chemistry and Physics* **20**, 2073-2097, doi:10.5194/acp-20-
2073-2020 (2020).
- 19 Jiang, X., Eum, Y. & Yoo, E. H. The impact of fire-specific PM_{2.5} calibration on health effect analyses. *Sci
Total Environ*, 159548, doi:10.1016/j.scitotenv.2022.159548 (2022).
- 20 Tao, Z., He, H., Sun, C., Tong, D. & Liang, X.-Z. Impact of Fire Emissions on U.S. Air Quality from 1997
to 2016—A Modeling Study in the Satellite Era. *Remote Sens* **12**, doi:10.3390/rs12060913 (2020).
- 21 Koplitz, S. N., Nolte, C. G., Pouliot, G. A., Vukovich, J. M. & Beidler, J. Influence of uncertainties in burned
area estimates on modeled wildland fire PM_{2.5} and ozone pollution in the contiguous U.S. *Atmos Environ*
191, 328-339, doi:10.1016/j.atmosenv.2018.08.020 (2018).

Reviewer Reports on the First Revision:

Referees' comments:

Referee #1 (Remarks to the Author):

Thanks very much to the authors for their careful and thorough response to my comments. I think the manuscript is much improved, and have only a few remaining minor comments that should be straightforward to address.

All of the paper's main conclusions rest on the accuracy of the model. I think that accuracy needs to be as fully and transparently explained in the main text, and the single paragraph devoted to model performance (lines 130-136) substantially expanded, even if at the cost of reducing other text and/or moving display items to the SI (I think Tables 1-2 and Fig 3 could easily be SI material).

That expanded paragraph should include the following, in my view:

- In most locations, the authors are not able to validate their smoke predictions. They are only able to validate them perhaps in the US, where other studies have tried to construct smoke PM estimates from ground data. They need to say this explicitly. e.g.: "In most regions of the world, we are only able to evaluate our model performance in predicting variation in total PM, and are not able to directly evaluate our predicted smoke PM."

- They need to be careful with how they compare their results to US-specific efforts. The statement: "demonstrated similar accuracy with Childs et al in validation against station measured smoke PM" is not correct; the accuracy is substantially lower, e.g. Fig S11. That's fine, and the authors are trying to do something harder by building a global model rather than a country model as in Childs et al, but the statement should be accurate. I would amend that statement and say something like: "Performance on pollution monitors in the US that were not used in model training was somewhat lower than a recent US-specific analysis, perhaps as a result of our attempt to build a globally-generalizable model".

- The fire-specific comparisons in figs s12-13 are welcome, but again a bit hard to directly evaluate since smoke PM is not directly observed in the ground data. They should just say this explicitly: e.g. "To further validate our data during known wildfire events, we compared model-predicted time-series of total and smoke PM against ground-station-measured total PM, training on only one monitor near the fire and evaluating on other nearby monitors. Model predictions tended to generally track daily variation in total PM (average $r^2=XX$), although tended to substantially understate PM concentrations during extreme PM periods." The authors should compute the average r^2 across the fires in fig s12 and report it in parentheses.

- Their response to my Q4 about doing cross-continent validation is I think reasonable given that they were able to track down some more stations, and the cluster CV appropriate (new Table S3). However, the fact that the global r^2 is higher than all the regional r^2 's suggest that again, lots of the predictive power is coming from the model predicting across-location differences in average PM rather than within-location variation. So users can evaluate this latter capability, I would encourage the authors to report within- R^2 for the cluster-based CV as well (just add a column, just as in the spatial 10-fold CV). They should report these values in the main text. e.g. "As a final test of our model's ability to generalize to regions where monitoring data are unavailable, we clustered globally available stations into 75 contiguous clusters, and used leave-one-out CV to evaluate model performance on each cluster as it was held out from model training. As expected, performance was somewhat lower than our globally pooled model, with our model explaining XX% of the temporal variation in total PM in clusters where it wasn't trained". Where XX is the r^2 from the cluster CV that you could add to table s3.

The wording above is just a suggestion, but the authors need to be exceedingly careful to state what is and what is not able to be validated, and what the validation results actually say.

Referee #2 (Remarks to the Author):

Thanks for the thorough and thoughtful revisions. I recommend publication.

Referee #3 (Remarks to the Author):

The authors have made substantial changes to the manuscript in response to my comments and those of the other two reviewers. I find that the authors have responded adequately to my major comments, in particular by better motivating the study in the Introduction based on discussion of the current literature. Some claims in the earlier paper that emphasized the “world’s first” novelty are now explained more fully in relation to previous studies.

Overall, I find that the study is high-quality. It is novel in important ways, especially in the production of a daily global dataset over many years, and analysis of that dataset for fire-related concentrations and population exposure.

I will reiterate my earlier comment that the main findings that fires generate 3-6% of global air pollution exposure is not itself surprising, but that the detail provided among different countries and world regions is valuable. I expect that the paper will be referenced widely in the coming years. But the decision about whether the novelty is sufficient to warrant publication in Nature lies with the editor.

Some comments

1. The paper is well-written in general, but the English could be improved in some places. I also thought that the Discussion section is now very long, and longer than I would expect for a Nature paper. Perhaps it meets length limits. But that long discussion of uncertainties etc. seems to take away from the impactfulness of the results highlighted in the first few paragraphs of the Discussion. For similar papers, I would often put the long discussion of uncertainties into a Discussion section, followed by a Conclusions section that highlights the main policy-relevant findings. Perhaps that organization would be an improvement.

2. I remain somewhat unclear on how the machine learning generates fire air pollution data in regions far from monitors. The authors seem to sufficiently detail their machine learning methods, and to test and evaluate it in several ways, but I’m looking for a more intuitive explanation. Of the inputs to the machine learning, only the GEOS-chem model output has information on where and when fires occurred based on its emissions inputs (if I understand correctly). I would think that the model must put a huge weight on the GEOS-chem estimates in regions with few monitors, because meteorological parameters and other inputs used would not know whether a fire occurred on a specific day or not. If it is true that GEOS-chem is used heavily far from monitors, then perhaps the authors should be more humble in presenting the uncertainties of their dataset far from monitors, which may be substantial.

l. 36. The “billion person days” emphasized in the abstract is hard to understand without more context. How about instead using indicators you present later in the study? Or might you relate this with exposure to air pollution from other sources? Then this sentence and the next emphasize the upward trend, but from the paper that increasing trend is due mainly to population growth rather than an increase in fires. Is it justified to emphasize the increasing trend here?

I. 58-60. I don't find this so notable, and I think it detracts from the main message of the paragraph.

I. 390. I don't think it's correct to say that most LMICs are hot and dry. Congo and Brazil are in rainforests.

Author Rebuttals to First Revision:

Referees' comments:

Referee #1 (Remarks to the Author):

Thanks very much to the authors for their careful and thorough response to my comments. I think the manuscript is much improved, and have only a few remaining minor comments that should be straightforward to address.

[Re] Thank you so much for all your comments and suggestions which have improved our study significantly.

All of the paper's main conclusions rest on the accuracy of the model. I think that accuracy needs to be as fully and transparently explained in the main text, and the single paragraph devoted to model performance (lines 130-136) substantially expanded, even if at the cost of reducing other text and/or moving display items to the SI (I think Tables 1-2 and Fig 3 could easily be SI material).

[Re] We have expanded this paragraph as you suggested, please see details below. We also moved Tables 1-2 to the Extended Data as suggested by the editor.

That expanded paragraph should include the following, in my view:

- In most locations, the authors are not able to validate their smoke predictions. They are only able to validate them perhaps in the US, where other studies have tried to construct smoke PM estimates from ground data. They need to say this explicitly. e.g.: "In most regions of the world, we are only able to evaluate our model performance in predicting variation in total PM, and are not able to directly evaluate our predicted smoke PM."

[Re] Suggestion accepted.

- They need to be careful with how they compare their results to US-specific efforts. The statement: "demonstrated similar accuracy with Childs et al in validation against station measured smoke PM" is not correct; the accuracy is substantially lower, e.g. Fig S11. That's fine, and the authors are trying to do something harder by building a global model rather than a country model as in Childs et al, but the statement should be accurate. I would amend that statement and say something like: "Performance on pollution monitors in the US that were not used in model training was somewhat lower than a recent US-specific analysis, perhaps as a result of our attempt to build a globally-generalizable model".

[Re] Suggestion accepted.

- The fire-specific comparisons in figs s12-13 are welcome, but again a bit hard to directly evaluate since smoke PM is not directly observed in the ground data. They should just say this explicitly: e.g. "To further validate our data during known wildfire events, we compared model-predicted time-series of total and smoke PM against ground-station-measured total PM, training on only one monitor near the fire and evaluating on other nearby monitors. Model predictions tended to generally track daily variation in total PM (average $r^2=XX$), although tended to substantially understate PM concentrations during extreme PM periods." The authors should compute the average r^2 across the fires in fig s12 and report it in parentheses.

[Re] We have computed the average R^2 across the fires and accept your suggested words.

- Their response to my Q4 about doing cross-continent validation is I think reasonable given that they were able to track down some more stations, and the cluster CV appropriate (new Table S3). However, the fact that the global r^2 is higher than all the regional r^2 's suggest that again, lots of the predictive power is coming from the model predicting across-location differences in average PM rather than within-location variation. So users can evaluate this latter capability, I would encourage the authors to report within- R^2 for the cluster-based CV as well

(just add a column, just as in the spatial 10-fold CV). They should report these values in the main text. e.g. “As a final test of our model’s ability to generalize to regions where monitoring data are unavailable, we clustered globally available stations into 75 contiguous clusters, and used leave-one-out CV to evaluate model performance on each cluster as it was held out from model training. As expected, performance was somewhat lower than our globally pooled model, with our model explaining XX% of the temporal variation in total PM in clusters where it wasn’t trained”. Where XX is the r^2 from the cluster CV that you could add to table S3.

[Re] We have added within- R^2 in the Table S3 (now the Extended Data Figure 4a) and reported it in the main text.

The wording above is just a suggestion, but the authors need to be exceedingly careful to state what is and what is not able to be validated, and what the validation results actually say.

[Re] We have accepted all your suggestions, but made some changes in content and sequence to make the text more logical, accurate and easy flowing. We also think it is important to mention that our accuracy, even lower than Childs et al’s US-specific smoke $PM_{2.5}$ estimates, were still much higher than the raw GEOS-Chem outputs. This is our major advance compared with existing global studies that solely relied on uncalibrated GEOS-Chem to estimate fire-sourced $PM_{2.5}$.

We have expanded first paragraph of the Results section significantly as follows:

“As detailed in the Methods, Extended Data and Supplementary Information, we validated our estimated all-source and fire-sourced $PM_{2.5}$ and O_3 in several ways.

The spatial 10-fold cross-validation (CV) (i.e., by dividing all stations into 10 approximately equal subsets, then performing cross-validation on each subset as it was held out from model training) demonstrated our machine learning models’ high accuracy in estimating both all-source daily average $PM_{2.5}$ ($R^2 = 0.89$, root mean squared error [RMSE] = $9.24 \mu\text{g}/\text{m}^3$) and all-source daily maximum 8-h O_3 ($R^2 = 0.80$, RMSE = $19.24 \mu\text{g}/\text{m}^3$) in new locations not in the training data. As a further test of our model’s ability to generalize to regions far from available training stations, we clustered globally available $PM_{2.5}$ and O_3 stations into 75 and 99 contiguous clusters, respectively, and used leave-one-out CV to evaluate model performance on each cluster as it was held out from model training. As expected, performance was lower than the spatial 10-fold CV. In clusters where the model wasn’t trained, the model estimates explained 69% and 67% of the overall variations in all-source $PM_{2.5}$ and O_3 , respectively; as well as 41% and 52% of local temporal daily variations (i.e., after excluding variations across stations and between years) of all-source $PM_{2.5}$ and O_3 , respectively. This performance, however, was still much higher than the performance of the uncalibrated raw GEOS-Chem outputs, suggesting that our models can predict the daily all-source $PM_{2.5}$ and O_3 in large remote areas with no training data with an accuracy much higher than the raw GEOS-Chem outputs alone.

Notably, in most regions of the world, we are only able to evaluate our model performance in predicting variation in all-source but not fire-sourced $PM_{2.5}$ and O_3 . We made two additional efforts to validate our estimated fire-sourced $PM_{2.5}$ and O_3 in some regions.

First, under a straightforward hypothesis that the station observed $PM_{2.5}$ and O_3 during wildfire events should be mainly contributed by wildfire smoke, we chose ten large wildfire events in Australia, US, Chile, Portugal, and South Africa to validate our estimated all-source and fire-sourced $PM_{2.5}$ and O_3 . For each wildfire event, we chose the most affected monitoring station (i.e., the nearby station showing largest increase in observed concentrations during the wildfire event compared to the pre-wildfire period) as the validation target. During the wildfire event and up to 60 days before and after the event, the observed daily all-source $PM_{2.5}$ or O_3 from the most affected station showed good agreement with our estimated daily all-source $PM_{2.5}$ ($R^2=0.64$ on average across events) and O_3 ($R^2=0.78$) based on model trained in stations excluding all nearby stations. Although our estimates tended to substantially understate $PM_{2.5}$ concentrations during some extreme $PM_{2.5}$ periods. Furthermore, we observed expected increase in the

estimated concentrations and proportions (among all sources) of fire-sourced PM_{2.5} and O₃ during the selected wildfire events compare to pre-wildfire period, suggesting that our models can reasonably capture the wildfires' impacts on the daily PM_{2.5} and O₃.

Second, we compared our estimated fire-sourced PM_{2.5} with the smoke PM_{2.5} (i.e., PM_{2.5} concentrations attributable to fire smoke overhead that detected by satellite images) estimated by Childs et al in contiguous US²⁰, and found a high agreement (Pearson correlation coefficient $r = 0.88$). When further validated against the smoke PM_{2.5} observed by 2147 PurpleAir stations that were neither in ours nor in Childs et al's training data: our estimated fire-sourced PM_{2.5} ($R^2 = 0.51$, RMSE = 11.76 $\mu\text{g}/\text{m}^3$) showed lower accuracy than Childs et al's estimated smoke PM_{2.5} ($R^2 = 0.66$, RMSE = 10.46 $\mu\text{g}/\text{m}^3$), perhaps as a result of our attempts to build a globally-generalizable model. However, our performance was still much higher than the accuracy of the fire-sourced PM_{2.5} from raw GEOS-Chem outputs ($R^2 = 0.18$, RMSE = 22.96 $\mu\text{g}/\text{m}^3$).

Based on our validated data, the global population exposures to fire-sourced PM_{2.5} and O₃ were described as follows.”

Referee #2 (Remarks to the Author):

Thanks for the thorough and thoughtful revisions. I recommend publication.

[Re] Thank you so much for your helpful comments and suggestions.

Referee #3 (Remarks to the Author):

The authors have made substantial changes to the manuscript in response to my comments and those of the other two reviewers. I find that the authors have responded adequately to my major comments, in particular by better motivating the study in the Introduction based on discussion of the current literature. Some claims in the earlier paper that emphasized the “world's first” novelty are now explained more fully in relation to previous studies.

Overall, I find that the study is high-quality. It is novel in important ways, especially in the production of a daily global dataset over many years, and analysis of that dataset for fire-related concentrations and population exposure.

I will reiterate my earlier comment that the main findings that fires generate 3-6% of global air pollution exposure is not itself surprising, but that the detail provided among different countries and world regions is valuable. I expect that the paper will be referenced widely in the coming years. But the decision about whether the novelty is sufficient to warrant publication in Nature lies with the editor.

[Re] We appreciate all your insightful comments and suggestions.

Some comments

1. The paper is well-written in general, but the English could be improved in some places. I also thought that the Discussion section is now very long, and longer than I would expect for a Nature paper. Perhaps it meets length limits. But that long discussion of uncertainties etc. seems to take away from the impactfulness of the results highlighted in the first few paragraphs of the Discussion. For similar papers, I would not often put the long discussion of uncertainties into a Discussion section, followed by a Conclusions section that highlights the main policy-relevant findings. Perhaps that organization would be an improvement.

[Re] We agree that the Discussion might be too long. Following your and the editor's advice, we have moved the long discussion of the uncertainties (nearly 600 words) to the Methods section, and instead just briefly mentioned these uncertainties in the Discussion section. This change enabled us to comply

with the word limit of the main text (5000 words).

2. I remain somewhat unclear on how the machine learning generates fire air pollution data in regions far from monitors. The authors seem to sufficiently detail their machine learning methods, and to test and evaluate it in several ways, but I'm looking for a more intuitive explanation. Of the inputs to the machine learning, only the GEOS-chem model output has information on where and when fires occurred based on its emissions inputs (if I understand correctly). I would think that the model must put a huge weight on the GEOS-chem estimates in regions with few monitors, because meteorological parameters and other inputs used would not know whether a fire occurred on a specific day or not. If it is true that GEOS-chem is used heavily far from monitors, then perhaps the authors should be more humble in presenting the uncertainties of their dataset far from monitors, which may be substantial.

[Re] Your understanding that GEOS-Chem is used heavily in our machine learning model is right. However, no matter whether the location is close or far away from the monitoring stations, the machine learning models trained in existing stations were used to perform bias correction of the GEOS-Chem outputs. We have added a more intuitive explanation of this process in the Methods:

“In step three, the daily total (all-source) $PM_{2.5}$ ($PM_{2.5_est_total}$) and O_3 ($O_{3_est_total}$) for each $0.25^\circ \times 0.25^\circ$ grid (regardless of whether the grid was close to or far away from the training stations) across global lands were estimated using the trained random forest models (i.e., machine learning calibration or bias correction algorithm found in where training stations existed) and global seamless predictor data. Then the final estimated fire-sourced $PM_{2.5}$ ($PM_{2.5_est_fire}$) and O_3 ($O_{2.5_est_fire}$) were calculated as follows ^{11,12,70}:

$$PM_{2.5_est_fire} = PM_{2.5_est_total} \times (PM_{2.5_chem_fire}/PM_{2.5_chem_total}) \quad (3)$$

$$O_{2.5_est_fire} = O_{3_est_total} \times (O_{3_chem_fire}/O_{3_chem_total}) \quad (4)$$

The $PM_{2.5_chem_fire}$ and $O_{3_chem_fire}$ refer to the downscaled fire-sourced $PM_{2.5}$ and O_3 from GEOS-Chem.”

Our spatial cross-validation (CV) and spatial cluster-based CV mimic the situations that the machine learning calibration models trained in known stations were applied to new locations (including locations far away from the training stations in the cluster CV). The results suggested that ML corrections still worked well in locations far away from the training stations (see the Extended Data Figure 4), and had much higher accuracy than the raw GEOS-Chem outputs (which meant in those locations the GEOS-Chem was not the only contributor of the estimates).

On the other hand, we have made it clear that the ML corrections were only applied to calibrate the all-source $PM_{2.5}$ or O_3 (i.e., to generate the $PM_{2.5_est_total}$ and $O_{3_est_total}$). While the proportions of fire contributions had to rely only on the GEOS-Chem (and still, this process did not differ by locations close to or far away from the training stations). We acknowledge that this was a potential limitation, as there were no observation to validate the accuracy of the GEOS-Chem estimated fire contributions. However, our sensitivity analyses using alternative fire emission inventories, and the validation against large wildfire events and Child et al's smoke $PM_{2.5}$ in US, suggested that the our GEOS-Chem simulations based on GFED4.1s could reasonably capture the impacts of landscape fires on $PM_{2.5}$ and O_3 (i.e., the GEOS-Chem estimated fire contributions tended to be reasonable at an acceptable accuracy), see Extended Data Figures 4-9.

I. 36. The “billion person days” emphasized in the abstract is hard to understand without more context. How about instead using indicators you present later in the study? Or might you relate this with exposure to air pollution from other sources? Then this sentence and the next emphasize the upward trend, but from the paper that increasing trend is due mainly to population growth rather than an increase in fires. Is it justified to

emphasize the increasing trend here?

[Re] We agree that the term “person days” might be difficult to understand for general readers (for whom the summary paragraph is designed) without detailed explanation. Following your and the editor’s suggestion, we have instead reported the average exposed days here. The sentence was revised as:

“During 2010-2019, there were 2.18 billion persons exposed to at least one day of substantial LFS air pollution per year, with each person in the world having on average 9.9 days of exposure per year. These two metrics increased by 6.8% and 2.1% compared to 2000-2009, respectively.”

We believe that it is reasonable to mention the increasing trend here for two reasons:

1) The population average exposure levels (i.e., mean population-weighted average fire-sourced $PM_{2.5}$ concentrations, average exposed days per person per year) also increased slightly during the period and contributed to the increase in total exposed persons and person days.

2) The increase in total exposed persons and person days have their unique public health significance in addition to the average exposure metrics, e.g., for understanding absolute scale of the fire air pollution problem, or for further estimating the total disease burden related to landscape-related air pollution. Therefore, emphasizing their increases (even mainly contributed by population growth) is still meaningful from the perspectives of public health.

I. 58-60. I don’t find this so notable, and I think it detracts from the main message of the paragraph.

[Re] We have deleted this paragraph due to the word limit required by the editor. Deleting this paragraph did not affect the flow of the whole paper.

I. 390. I don’t think it’s correct to say that most LMICs are hot and dry. Congo and Brazil are in rainforests.

[Re] We have revised the “most” to “many”.